

# Non-perturbative topological string theory on compact Calabi-Yau 3-folds

**Jie Gu**[1⋆]**, Amir-Kian Kashani-Poor**[2†]**, Albrecht Klemm**[3,4‡] **and Marcos Mariño**[5∘]

**1** School of Physics and Shing-Tung Yau Center, Southeast University, Nanjing 210096, China
**2** LPENS, ENS, Université PSL, CNRS, Sorbonne Université,
Université Paris Cité, F-75005, France
**3** Bethe Center for Theoretical Physics, Universität Bonn, D-53115, Germany
**4** Hausdorff Center for Mathematics, Universität Bonn, D-53115, Germany
**5** Département de Physique Théorique et Section de Mathématiques,
Université de Genève, Genève, CH-1211 Switzerland

⋆ eij.ug.phys@gmail.com, † amir-kian.kashani-poor@ens.fr,
‡ aklemm@th.physik.uni-bonn.de, ∘ Marcos.Marino@unige.ch

## Abstract

We obtain analytic and numerical results for the non-perturbative amplitudes of topological string theory on arbitrary, compact Calabi–Yau manifolds. Our approach is based on the theory of resurgence and extends previous special results to the more general case. In particular, we obtain explicit trans-series solutions of the holomorphic anomaly equations. Our results predict the all orders, large genus asymptotics of the topological string free energies, which we test in detail against high genus perturbative series obtained recently in the compact case. We also provide additional evidence that the Stokes constants appearing in the resurgent structure are closely related to integer BPS invariants.



# 1 Introduction

String theory was originally formulated as a purely perturbative theory, but there are many indications that a complete description of it will also involve non-perturbative corrections in the string coupling constant. One of these indications is the factorial growth of the genus expansion, discovered in [1] in the case of the bosonic string and generalized afterwards to other situations [2]. It has been proposed that D-brane amplitudes supply the non-perturbative corrections connected to the factorial growth of perturbation theory [3], although a detailed verification of this statement has only been made in non-critical string theory (see e.g. [4]).

The connection between the factorial divergence of perturbative series and the existence of non-perturbative sectors is a universal feature of quantum theories, pointed out by Dyson [5] and first tested quantitatively by Bender and Wu, in the context of quantum mechanics [6]. This connection is the basis of the modern theory of resurgence, which provides a far-reaching mathematical framework to understand the emergence of non-perturbative sectors (see [7–10] for recent introductions to resurgence). In this theory, one can extract from the Borel singularities of the perturbative expansion a collection of new formal power series which characterize the non-perturbative sectors, as well as a set of analytic invariants called Stokes constants. In quantum field theory, these sectors correspond to instantons and renormalons. In some cases, Stokes constants turn out to be closely related to BPS invariants [11–16], making the theory of resurgence a powerful tool in mathematical physics.

Topological string theory on Calabi–Yau 3-folds provides a very rich testing ground for our understanding of string theory, and it has many applications both in mathematics and in quantum field theory. In view of this, there have been many efforts to understand its non-perturbative aspects. In [17–19] it was proposed to use the theory of resurgence as a rigorous tool to understand the emergence of non-perturbative sectors in topological string theory from its perturbative series. One expects to find in this way non-perturbative amplitudes, exponentially small in the string coupling constant, as well as a set of Stokes constants potentially related to BPS invariants of the Calabi–Yau manifold.

A significant advance in this program was made in [20, 21], inspired by Écalle's theory of ODEs. In this theory, the non-perturbative sectors associated to the perturbative solution to an ODE can be found by considering formal "trans-series" solutions, i.e. formal power series with exponentially small prefactors. Guided by this principle, [20, 21] proposed to find the non-perturbative sectors of the topological string by considering trans-series solutions to the holomorphic anomaly equations of [22]. [21] developed this framework in the case of a local Calabi–Yau example and provided concrete evidence that these solutions have the expected properties. In particular, they explain quantitatively the factorial divergence of the perturbative series. In a more recent development, [23] found explicit, closed formulae for these trans-series solutions in the case of local Calabi–Yau manifolds with one modulus. They showed in particular that these solutions are generalizations of the eigenvalue tunneling amplitudes characterizing instanton sectors in matrix models.

So far the results obtained with the trans-series approach have focused on local Calabi–Yau manifolds with one modulus, and it was not clear whether one can extend it to the compact, generic case. In this paper we show that this can be done, and we present explicit, closed formulae for multi-instanton trans-series, for general Calabi–Yau manifolds. Interestingly, they are natural generalizations of the formulae of [23] in the local case, once the theory is formulated in the big moduli space. In particular, the non-perturbative amplitudes take the form of generalized eigenvalue tunneling. As already pointed out in [23] in the local case, this is suggestive of some underlying "bit" model of the Calabi–Yau manifold, in which the local coordinates of the moduli space are quantized in units of the string coupling constant. Such a picture is not completely unexpected in the local case, due to various large $N$ dualities, but it is more striking in the compact case.

A defining property of the trans-series obtained in the theory of resurgence is that they should provide precise asymptotic formulae for the perturbative series, and this can be regarded as a test of the theory, as already pointed out in this context in [21]. Armed with the explicit trans-series solutions for the compact case, we verify this property in detail in the case of compact, one-parameter Calabi–Yau manifolds. For this we rely on the important progress in the calculation of perturbative topological string series started by [24], and recently developed in [25]. We find that our trans-series describe with very high precision the asymptotic behavior of the topological string free energies at large genus. These are new results on the generating functionals of Gromov–Witten invariants of compact Calabi–Yau manifolds.

The formal trans-series solutions obtained in this paper provide explicit descriptions of the non-perturbative amplitudes associated to a given Borel singularity. A complete description of this resurgent structure includes the list of actual Borel singularities and their associated Stokes constants. On this front, our results are partial. The leading Borel singularities at MUM and conifold points can be shown to be given by periods over distinguished integral cycles of the mirror Calabi–Yau. Based on general principles and explicit computations, we conjecture that the correspondence between Borel singularities and *integral* periods holds generally.[1] In addition, we identify a family of Borel singularities associated to A-periods, whose Stokes

---

[1] The conjecture that the singularities correspond to periods was stated in the local case in [26], and confirmed in [20, 21]. The precise integrality condition and the extension to the compact case are new.

constants are genus zero Gopakumar–Vafa invariants. This verifies the general expectations that these constants are related to BPS counting, although much more work is needed in this direction.

We find that the propagators of the topological string on compact Calabi–Yau 3-folds are related to the periods on compact Calabi–Yau 4-folds, making the perturbative topological string theory expansion reminiscent of the evaluation of perturbative quantum field theory amplitudes in terms of master integrals [27], which are periods on (degenerate) higher dimensional Calabi–Yau varieties. It would be interesting to understand if the non-perturbative behavior of these quantum field theory amplitudes is likewise related to integral periods that are singled out by their behavior at the Landau singularities.

This paper is organized as follows. In section 2, we provide a detailed review of the special geometry of general Calabi–Yau 3-folds, which is crucial both for the original theory of [22] and for the non-perturbative generalization developed in this paper. We also review aspects of one-parameter compact Calabi–Yau 3-folds which will serve as testing grounds for our theory. Section 3 reviews the holomorphic anomaly equations of [22], while section 4 recalls some basic ingredients of the theory of resurgence. In section 5, we find trans-series solutions to the holomorphic anomaly equations for general Calabi–Yau 3-folds. We first do some warm-up calculations in the compact, one-modulus case, and then develop an operator formalism which allows us to write down exact multi-instanton amplitudes. This generalizes in particular the results of [23] to the most general case. The results of section 5 are based on boundary conditions which fix holomorphic ambiguities in the non-perturbative amplitudes, and we explain these boundary conditions in detail at the end of the section. In section 6, we present numerical evidence for our results from compact, one-modulus Calabi–Yau 3-folds. This evidence covers the structure of Borel singularities, as well as the large genus asymptotics controlled by instanton amplitudes. Section 7 contains some conclusions and prospects for future work.

## 2 Period geometry for compact Calabi–Yau 3-folds

In this section, we shall describe aspects of the period geometry of complex families of compact Calabi–Yau 3-folds $W$ that are central to computing the topological string partition function, or equivalently, its free energy. After discussing such geometries in general in subsection 2.1, we will specialize to the case of hypergeometric one-complex parameter Calabi–Yau manifolds in the ensuing subsection. These models will serve as instructive examples for the period geometry and will provide the principal testing ground for our non-perturbative predictions.

The topological string partition function $Z^{(0)}$ is related to the genus $g$ topological string free energies $F_g(z)$ via

$$Z^{(0)}(g_s, z) = \exp(F^{(0)}(g_s, z)) = \exp\left( \sum_{g=0}^{\infty} g_s^{2g-2} F_g(z) \right). \tag{1}$$

We have attached a superscript $^{(0)}$ to the conventional topological string free energy and partition function to distinguish them from the non-perturbative sectors of the theory which we shall be studying below. The RHS of (1) is a perturbative, asymptotic series in the topological string coupling constant $g_s$ and depends on the complex structure moduli $z$ of $W$. According to a theorem of Tian [28] and Todorov [29], one has $h^{2,1}(W) = r$ unobstructed complex moduli parameterizing the complex structure moduli space $\mathcal{M}_{cs}(W)$. A property of $F^{(0)}(g_s, z)$ which is absent for the topological string on local Calabi–Yau spaces and many other familiar asymptotic series is that $g_s$ and the $F_g(z)$ are non-trivial sections of powers of the Kähler line bundle $\mathcal{L}$ over $\overline{\mathcal{M}_{cs}(W)}$, the canonical compactification of the complex structure moduli space $\mathcal{M}_{cs}(W)$. We shall review this structure below.

The computation of $F^{(0)}(g_s, z)$ via the holomorphic anomaly equations will be the topic of Section 3. A principal feature of the setup is that by the construction of the involved quantities from the solutions of a Picard-Fuchs differential ideal, all $F_g(z)$ have good (real) analytic properties in $z$. In particular, convergent series expressions for them exist in a neighborhood of any point of $\overline{\mathcal{M}_{cs}(W)}$.

## 2.1 Differential properties of the Calabi–Yau 3-fold periods

The defining property of a complex analytic Calabi–Yau 3-fold $W$ is the existence of a unique $(1,1)$ Kähler form $\omega$ in $H^{1,1}(W, \mathbb{R})$ and a holomorphic $(3,0)$-form $\Omega(z)$, unique up to normalization, spanning $H^{3,0}(W, \mathbb{Q})$. In the case of complete intersections, the $(3,0)$-form is explicitly given by Griffiths residuum expressions. Choosing a fixed basis $\gamma_i$ in $H^3(W, \mathbb{C})$, we can expand $\Omega$ as

$$\Omega = \sum_{i=1}^{b_3(W)} f_i(z)\gamma_i, \tag{2}$$

with coefficients given by periods

$$f_i(z) = \int_{\Gamma_i} \Omega(z). \tag{3}$$

Here, the $\Gamma_i$ furnish a basis of $H_3(W, \mathbb{C})$ dual to $\{\gamma_i\}$. There is a symplectic intersection pairing

$$\Sigma : H_3(W, \mathbb{Z}) \times H_3(W, \mathbb{Z}) \to \mathbb{Z}, \tag{4}$$

on the integral homology $H_3(W, \mathbb{Z})$ of $W$, which allows defining a dual *integral* symplectic basis also in $H^3(W, \mathbb{C})$, as we shall discuss below. The $b_3(W)$ periods span a vector space that is identified with the solution space of a linear Picard-Fuchs differential ideal,

$$\mathfrak{L}(\theta_z, z) f_i(z) = 0, \tag{5}$$

where $\theta_z = z\frac{d}{dz}$ are logarithmic derivatives. In the case of one-parameter families (where $b_3(W) = 2r + 2 = 4$), this ideal is generated by a fourth order Picard-Fuchs operator $L^{(4)}$, given below for the hypergeometric families in equation (74). In general, the ideal is generated by many linear differential operators in the complex moduli $z$,

$$\mathfrak{L}(\theta_z, z) = \{L_i^{(k)}(\theta_z, z) \,|\, i = 1, \ldots, |\mathfrak{L}|, \, k = 2, 3, 4\}. \tag{6}$$

### 2.1.1 Special geometry

The moduli space $\mathcal{M}_{cs}(W)$ of complex structures of a Calabi–Yau manifold $W$ is a special Kähler manifold [30,31]. Here, we will review the basic aspects of this geometric structure, which underlies the holomorphic anomaly equations of [22] and their non-perturbative generalization.

Special geometry derives entirely from two bilinear relations: one *real* bilinear involving the holomorphic $(3,0)$ form $\Omega(z)$ and the complex conjugate $(0,3)$ form $\bar{\Omega}(\bar{z})$,

$$e^{-K} = i \int_W \Omega \wedge \bar{\Omega} > 0, \tag{7}$$

and the other a *complex* bilinear relation involving $\Omega(z)$ and its derivatives with regard to the complex moduli,

$$\int_W \Omega \wedge \partial_{z_{I_1}} \ldots \partial_{z_{I_k}} \Omega = \begin{cases} 0, & \text{if } k < 3, \\ C_I(z) \in \mathbb{Q}[z], & \text{if } k = 3. \end{cases} \tag{8}$$

With regard to the Hodge decomposition on $H^3(W, \mathbb{C})$, the $\wedge$ pairing has the property that

$$\int_W \alpha_{mn} \wedge \beta_{pq} = 0, \quad \text{unless} \quad m + p = n + q = 3, \tag{9}$$

for $\alpha_{mn} \in H^{m,n}(W)$ and $\beta_{pq} \in H^{p,q}(W)$. Griffiths transversality implies

$$\underline{\partial}_I \Omega = \partial_{z_{I_1}} \dots \partial_{z_{I_k}} \Omega \in \bigoplus_{p=0}^{\min(k,3)} H^{3-p,p}(W), \tag{10}$$

for each index set $I$ with $I_i \in \{1, \dots, r\}$ and $|I| = k$. This together with (9) implies the zero in (8). The derivatives of order $k \leq 3$ redundantly span $H^3(W, \mathbb{Q})$; beyond $k = 3$, no new classes arise. The redundancies are encoded in $\mathfrak{L}$, which can be derived by systematically[2] finding all cohomological relations among the derivatives $\underline{\partial}_I \Omega$ that generate $\mathfrak{L}$. The $C_I(z)$ occurring in (8) are *rational functions* labeled, taking the symmetry in the indices into account, by $\frac{1}{3!} \prod_{m=0}^2 (r+m)$ 3-tuples $I$. They are called three point functions or Yukawa couplings. Note that (8) can be expressed as bilinears in the periods (3). Hence, by (5), $\mathfrak{L}$ imposes differential relations on the $C_I(z)$; they are determined by these up to one multiplicative normalization. For instance, in the case of the one-parameter hypergeometric differential equation (74) which we shall introduce below, one can easily check that

$$C_{111} = \frac{(2\pi i)^3 \kappa}{z^3(1 - \mu^{-1}z)}, \tag{11}$$

where the normalization is in accord with the one chosen below in (79).

Note that the variables $z$ are globally defined on $\mathcal{M}_{cs}(W)$. All of the metric data which follows from the Kähler potential (7), such as the Weil–Petersson metric on $\mathcal{M}_{cs}(W)$,

$$G_{i\bar{j}} = \partial_{z_i} \bar{\partial}_{\bar{z}_j} K(z), \tag{12}$$

as well as the associated covariant derivatives

$$D_i F_j = (\partial_i + nK_i)F_j - \Gamma_{ij}^k F_k, \qquad F_j dz^j \in \Gamma(\mathcal{L}^n \otimes (T^*\mathcal{M}_{cs})^{1,0}), \tag{13}$$

$$D_i F^j = (\partial_i + nK_i)F^j + \Gamma_{ik}^j F^k, \qquad F^j \partial_j \in \Gamma(\mathcal{L}^n \otimes (T\mathcal{M}_{cs})^{1,0}), \tag{14}$$

$$D_0 F = nF, \qquad F \in \Gamma(\mathcal{L}^n), \tag{15}$$

with regard to the Weil–Petersson connection on the holomorphic (co)tangent bundles of $\mathcal{M}_{cs}$,

$$\Gamma_{ij}^k = G^{k\bar{l}} \partial_{\bar{l}} \partial_i \partial_j K, \tag{16}$$

and the connection on the Kähler line bundle $\mathcal{L}^n$,

$$nK_i = n\partial_i K, \tag{17}$$

are equally globally defined quantities. We recall that the Kähler line bundle $\mathcal{L}$ is trivial, hence defined via its global section $\Omega \in \Gamma(\mathcal{L})$ (likewise, for $\overline{\mathcal{L}}$ and $\bar{\Omega} \in \Gamma(\overline{\mathcal{L}})$). By (7), a rescaling

$$\Omega \to \Omega e^{f(z)}, \tag{18}$$

of $\Omega$ induces a Kähler gauge transformation

$$K \to K - f(z) - \overline{f(z)}. \tag{19}$$

---

[2]Using the Griffiths reduction formula and Gröbner basis calculus.

As follows from its definition in (8), $C_{ijk}(z)$ is a section of $\mathcal{L}^2 \otimes \text{Sym}^3((T^*\mathcal{M}_{cs})^{1,0})$.[3] The Kähler gauge transformation extends to the sections of $\mathcal{L}^n$ in the obvious way. The quantities just introduced have anti-holomorphic counterparts $\overline{D}_{\bar{0}}, \overline{D}_{\bar{j}}$, as well as

$$\Gamma^{\bar{i}}_{\bar{j}\bar{k}} = G^{i\bar{i}}\partial_i\partial_{\bar{j}}\partial_{\bar{k}}K\,. \tag{20}$$

We recall that Christoffel symbols with mixed holomorphic and anti-holomorphic indices vanish. Let us note that the covariant derivatives appearing in (13) have the properties that

$$D_\iota G_{i\bar{j}} = 0\,, \quad D_i e^{nK} = 0\,, \quad D_0 e^{nK} = -n e^{nK}\,, \quad \text{for} \quad \iota = 0, 1, \ldots, r\,, \quad i = 1, \ldots, r\,. \tag{21}$$

Throughout this paper, we shall have Greek indices $\iota, \alpha, \ldots$ run over the range $0, 1, \ldots, r$, while Latin indices $i, k, \ldots$ are meant to run over $1, \ldots, r$.

Using (7), (8) and (9), one concludes that

$$D_i \Xi_0 = \Xi_i\,, \quad D_i \Xi_j = \mathrm{i}e^K C_{ijk} G^{k\bar{k}}\bar{\Xi}_{\bar{k}}\,, \quad D_i \bar{\Xi}_{\bar{k}} = G_{i\bar{k}}\bar{\Xi}_{\bar{0}}\,, \quad D_i \bar{\Xi}_{\bar{0}} = 0\,, \tag{22}$$

where $\Xi_0 := \Omega$ and $\bar{\Xi}_{\bar{0}} := \bar{\Omega}$ span $H^{3,0}(W)$, $H^{0,3}(W)$, and $\Xi_i$ and $\bar{\Xi}_{\bar{k}}$ span $H^{2,1}(W)$, $H^{1,2}(W)$, respectively. Since (22) are cohomological relations, they can be integrated over closed cycles $\Gamma$ to yield relations between periods. Introducing

$$\chi_\alpha^\Gamma := \int_\Gamma \Xi_\alpha\,, \quad \bar{\chi}_{\bar{\alpha}}^\Gamma := \int_\Gamma \bar{\Xi}_{\bar{\alpha}}\,, \quad \alpha, \bar{\alpha} = 0, \ldots, r\,, \tag{23}$$

and combining these into a $b_3$ dimensional vector

$$\vec{\chi}_\Gamma = (\chi_0, \chi_i, \bar{\chi}_{\bar{i}}, \bar{\chi}_{\bar{0}})^T\,, \tag{24}$$

we can express the relations (22) and their conjugates as

$$(D_i + A_i)\vec{\chi}_\Gamma = 0\,, \quad (\overline{D}_{\bar{i}} + \bar{A}_{\bar{i}})\vec{\chi}_\Gamma = 0\,. \tag{25}$$

Here, $A_i$ and $\bar{A}_{\bar{i}}$ are $b_3 \times b_3$ upper diagonal and lower diagonal matrices respectively, whose explicit form follows from (22). These two sets of identities imply a relation between the Riemann tensor and the Yukawa couplings,

$$-R_{i\bar{j}}{}^k{}_l = \partial_{\bar{j}}\Gamma^k_{il} = \delta^k_l G_{\bar{j}i} + \delta^k_i G_{\bar{j}l} - C_{\bar{j}}^{km} C_{ilm}\,, \tag{26}$$

easily obtained from the second of the following relations:

$$[D_i, \bar{D}_{\bar{j}}]\Xi_0 = -G_{i\bar{j}}\Xi_0\,, \qquad [D_i, \bar{D}_{\bar{j}}]\Xi_k = -G_{i\bar{j}}\Xi_k - R_{i\bar{j}}{}^p{}_k \Xi_p\,. \tag{27}$$

In (26),

$$C_{\bar{j}}^{kl} := e^{2K}\overline{C}_{\bar{j}\bar{k}\bar{l}}G^{k\bar{k}}G^{l\bar{l}} \in \Gamma\left(\mathcal{L}^{-2} \otimes (T^*\mathcal{M}_{cs})^{0,1} \otimes \text{Sym}^2((T\mathcal{M}_{cs})^{1,0})\right)\,, \tag{28}$$

is a section of the indicated bundle. Note that by holomorphicity of the Yukawa coupling and by (21), it is covariantly constant,

$$D_i C_{\bar{j}}^{kl} = 0\,. \tag{29}$$

By using (27), we find that (26) is equivalent to the integrability condition

$$[D_i + A_i, \overline{D}_{\bar{j}} + \bar{A}_{\bar{j}}] = 0\,. \tag{30}$$

---

[3]More precisely, $C_{ijk}dz^i \otimes dz^j \otimes dz^k$ is such a section. We will leave the inclusion when necessary of differentials and derivative operators in such statements to the reader in the following.

It is convenient to introduce a fixed topological integral symplectic basis $\{A_I, B^I | I = 0, \ldots, r\}$ for $H_3(W, \mathbb{Z})$, such that

$$A_I \cap A_J = B^I \cap B^J = 0, \quad A_I \cap B^J = -B^J \cap A_I = \delta_I^J. \tag{31}$$

Such a basis is determined only up to a symplectic transformation $\Gamma$ in $\mathrm{Sp}(b_3(M), \mathbb{Z})$. Given a choice of basis, we can define a dual integral symplectic basis in cohomology, $\{\alpha_I, \beta^I | I = 0, \ldots, r\}$, defined via the non-vanishing pairings

$$\int_{A_I} \alpha_J = \delta_J^I, \qquad \int_{B^I} \beta^J = \delta_I^J. \tag{32}$$

Let us introduce the rank $b_3(W)$ period vector $\vec{\Pi}$ as the first column of the period matrix[4] $\Pi$

$$\vec{\Pi} = \begin{pmatrix} \int_{B^I} \Omega \\ \int_{A_I} \Omega \end{pmatrix} = \begin{pmatrix} P_I \\ X^I \end{pmatrix}, \quad \Pi = \begin{pmatrix} \int_{B^I} \Xi^T \\ \int_{A_I} \Xi^T \end{pmatrix} = \begin{pmatrix} D_\alpha P_I & D_{\tilde{\alpha}} \bar{P}_I \\ D_\alpha X^I & D_{\tilde{\alpha}} \bar{X}^I \end{pmatrix} = \begin{pmatrix} \rho_{I\alpha} & \bar{\rho}_{I\tilde{\alpha}} \\ \chi_\alpha^I & \bar{\chi}_{\tilde{\alpha}}^I \end{pmatrix}. \tag{33}$$

In the integral symplectic basis (31), the intersection pairing (4) is represented by the $b_3(W) \times b_3(W)$ matrix

$$\Sigma = \begin{pmatrix} \mathbf{0} & \mathbf{1} \\ -\mathbf{1} & \mathbf{0} \end{pmatrix}, \tag{34}$$

and symplectic transformations $\Gamma$ are defined by the property $\Gamma^T \Sigma \Gamma = \Sigma$. In terms of $\Sigma$, we can write the bilinears in (7), (8) as

$$\int_W \Omega \wedge \bar{\Omega} = \vec{\Pi}^\dagger \cdot \Sigma \cdot \vec{\Pi}, \qquad \int_W \Omega \wedge \underline{\partial}_I \Omega = -\vec{\Pi}^T \cdot \Sigma \cdot \underline{\partial}_I \vec{\Pi}. \tag{35}$$

Let $P_*$ be a point in $\mathcal{M}_{cs}(W)$ and $\delta$ local coordinates near this point, chosen in such a way that $\delta(z)$ are rational functions in the original variables $z$, and such that $\delta = 0$ specifies $P_*$. For any complete basis of solutions $\vec{\Pi}_*^T(\delta)$ with $\mathfrak{L}(\delta)\vec{\Pi}_*^T(\delta) = 0$ near $P_*$, the Griffiths bilinear relation (8) determines an intersection matrix $\Sigma^*$ associated to this basis by expanding

$$\Pi_*^T(\delta) \cdot \Sigma^* \cdot \underline{\partial}_{I_3}^* \Pi_*(\delta) = -C_{I_3}^*(\delta), \tag{36}$$

in $\delta$. Here, $\underline{\partial}_{I_3}^*$ is a triple derivative in the coordinates $\delta$ and $C_{I_3}^*(\delta)$ is the tensor transform of $C_{I_3}(z)$ to the $\delta$ coordinates.

In physics applications, $H_3(W, \mathbb{Z})$ is identified with the electric-magnetic charge lattice, and $Z_\Gamma(z) = \int_\Gamma \Omega(z)$ with the central charge of special Lagrangian D-branes wrapping a cycle $\Gamma \in H_3(W, \mathbb{Z})$. The integrality of the charge lattice is natural due to the geometric origin of charged objects as wrapped branes; it is also required by the Dirac–Zwanziger quantization condition for electric and magnetic charges. By convention, electric and magnetic charges are associated to the A- and B-periods respectively. From the symplectic or Hamiltonian point of view, the choice of basis (31) for $H_3(W, \mathbb{Z})$ defines a polarization of $H^3(W, \mathbb{K})$ for $\mathbb{K} = \mathbb{C}$, $\mathbb{R}$ or $\mathbb{Q}$. The generating set $\{\alpha_I, \beta^I\}$ for the lattice $H^3(W, \mathbb{Z})$ also provides a basis of $H^3(W, \mathbb{K})$ over the respective field $\mathbb{K}$.

### 2.1.2 Special geometry in special coordinates

The local Torelli theorem states that a sufficiently small domain $U_* \subset \mathcal{M}_{cs}(W)$ can be identified with an open set in $\mathbb{P}^r$ using the period map $z \mapsto (X_*^0(z) : \ldots : X_*^r(z)) \in \mathbb{P}^r$, see (33), and parametrized by inhomogeneous or affine coordinates

$$t_*^i(z) = X_*^i / X_*^0. \tag{37}$$

---

[4]Here, we define the canonical order $\alpha = 0, 1, \ldots, r$ and $\tilde{\alpha} = 1, \ldots, r, 0$.

A priori, there is no preferred choice of the 3-cycles $A_I^*$ yielding the homogeneous coordinates $X_*^I$; in particular, they do not need to furnish an integral basis of $H_3(X)$. At generic points of $\mathcal{M}_{cs}(W)$, this choice does not matter much. Near special points in moduli space, however, one wants to make a choice so that the period map and the $t_*^i$ have the simplest possible branch behavior over $U_*$.

By singling out $(1+r)$ non-intersecting A-cycles and considering the associated periods $X_*^I$ as *homogeneous coordinates* on $\mathcal{M}_{cs}(W)$, we are choosing a *projective frame*. An *affine frame* is obtained by singling out one cycle $A_0^*$ among the A-cycle, whose associated period we divide by to define the *affine coordinates* $t_*^i = \int_{A_i^*} \Omega / \int_{A_0^*} \Omega = X_*^i/X_*^0$.

The relations (8) allow us to define a prepotential $F^*(X_*)$ via

$$2F_*(X_*) = X_*^I P_I^*(X_*),\qquad(38)$$

such that

$$P_I^*(X_*) = \frac{\partial}{\partial X_*^I} F_*(X_*).\qquad(39)$$

Here and in the following, we will use Einstein summation conventions also on repeated homogeneous indices $I, J$, etc. Equation (38) is Euler's relation. It implies that $F(X_*)$ is a homogeneous function of degree two in the $X_*^I$, and hence a section of $\mathcal{L}^2$, consistent with the fact that the $P_I^*$ are sections of $\mathcal{L}$, hence must be homogeneous functions of the $X_*^I$ of degree 1. Traditionally, the $P_I^*$ are denoted $F_I$, but the symplectic structure on $H^3(W,\mathbb{R})$ suggests thinking of the $X_*^I$ as coordinates, the $P_I^*$ as momenta, and $\exp(F_*(X))$ as a state, thus motivating our notation. The star as a superindex is intended as a reminder that a particular polarization – a symplectic frame in phase space – has been chosen, organizing periods into coordinates and momenta.

Introducing the inhomogeneous prepotential $\mathcal{F}_*(t_*) \in \Gamma(\mathcal{L}^0)$ by

$$F_*(X_*) = \mathcal{F}_*(t_*)(X_*^0)^2 \in \Gamma(\mathcal{L}^2),\qquad(40)$$

the period vector $\vec{\Pi}_*$ can be written as

$$\vec{\Pi}_* = X_*^0\left(2\mathcal{F}_*(t_*) - t^i \partial_{t_*^i}\mathcal{F}_*(t_*), \partial_{t_*^i}\mathcal{F}_*(t_*), 1, t^i\right).\qquad(41)$$

The third derivative of the prepotential yields the three point functions in terms of the normalized holomorphic 3-form,

$$C_{t_*^i t_*^j t_*^k} := \int \frac{\Omega}{X_*^0} \wedge \partial_{t_*^i}\partial_{t_*^j}\partial_{t_*^k}\frac{\Omega}{X_*^0} = \partial_{t_*^i}\partial_{t_*^j}\partial_{t_*^k}\mathcal{F}_*(t_*).\qquad(42)$$

One of the special points in $\mathcal{M}_{cs}(W)$ is the point of maximal unipotent monodromy (m). We will choose our complex coordinates in such a way that the MUM point occurs at $z = 0$. This point is characterized by the property that the symbol $\mathfrak{S}(\mathfrak{L}) = \{L_i^{(k)}(\theta_z, z)|_{z=0}, i = 1,\ldots,|\mathfrak{L}|, k = 2,3,4\}$ is maximally degenerate, i.e. has a $(2r+2)$-fold degenerate zero in the formal variables $\theta_{z_i}$. Let us assume for simplicity that all local exponents are zero. We then have a unique holomorphic solution $X_m^0$, and $r$ single logarithmic solutions $X_m^a$, $a = 1,\ldots,r$. Introducing inhomogeneous coordinates $t_m^a = X_m^a/X_m^0$, the period vector reads

$$\vec{\Pi} = \begin{pmatrix} P_0^m \\ P_a^m \\ X_m^0 \\ X_m^a \end{pmatrix} = \begin{pmatrix} \int_{B^0}\Omega \\ \int_{B^a}\Omega \\ \int_{A_0}\Omega \\ \int_{A_a}\Omega \end{pmatrix} = X_m^0\begin{pmatrix} 2\mathcal{F}_m(t_m) - t_m^i\partial_{t_m^i}\mathcal{F}_m(t_m) \\ \partial_{t_m^a}\mathcal{F}_m(t_m) \\ 1 \\ t_m^a \end{pmatrix}.\qquad(43)$$

If the Calabi–Yau manifold $W$ has a known mirror manifold $M$, the MUM point corresponds to the so-called large radius point of the mirror. Mirror symmetry identifies the Kähler parameters

$$t^a = \int_{\mathcal{C}^a} (b + i\omega) = B^a + iA^a\,, \tag{44}$$

with

$$t_{\mathrm{m}}^a(z) = 1/(2\pi i)\log(z_a) + \mathcal{O}(z)\,. \tag{45}$$

This is the so-called mirror map (see (79) for the one parameter case). In (44), $A^a$ is the area of a complex curve $\mathcal{C}^a$ on the boundary of the the Mori cone of $M$. With this identification of variables, the instanton corrected genus zero prepotential $\mathcal{F}(t)$ of $M$ is identified with $\mathcal{F}_{\mathrm{m}}(t_{\mathrm{m}})$. We will denote by

$$\kappa_{ijk} = D_i \cdot D_j \cdot D_k\,, \qquad \gamma_i = \frac{c_2(TM) \cdot D_i}{24}\,, \qquad \chi(M)\,, \tag{46}$$

the intersection numbers among the divisor classes $D_i$ in $M$, the normalized components of the second Chern class of the tangent bundle of $M$, and the Euler number $\chi$ of $M$, respectively. In terms of this topological data, the instanton corrected prepotential near the large radius point has the structure

$$\mathcal{F}_{\mathrm{m}}(t_{\mathrm{m}}) = (X_{\mathrm{m}}^0)^{-2}\mathcal{F}_0(X_{\mathrm{m}}) = -\frac{\kappa_{ijk}t^i t^j t^k}{3!} + \frac{\sigma_{ij}t^i t^j}{2} + \gamma_j t^j - \frac{1}{(2\pi \mathrm{i})^3}\sum_{\beta \in H_2(M,\mathbb{Z})} n_{0,\beta}\mathrm{Li}_3(Q_\beta)\,. \tag{47}$$

To lighten the notation, we have dropped the subindex $\mathrm{m}$ on $t$ and $Q$ on the RHS and also in (48) below. We have $\sigma_{ij} = (\kappa_{iij} \bmod 2)$, and the $n_{g,\beta} \in \mathbb{Z}$ are BPS indices, which count stable sheaves with one complex dimensional support. We have introduced the world-sheet instanton counting parameter

$$Q_\beta = \mathrm{e}^{2\pi \mathrm{i}\beta \cdot t}\,, \quad \text{with} \quad \beta \cdot t = \sum_{a=1}^{h_2(M)} \beta_a t^a\,. \tag{48}$$

Note that $n_{0,0} = \chi(M)/2 \in \mathbb{Z}$, such that the $\beta = 0$ contribution to the sum in (47) yields the constant contribution $\zeta(3)\chi/2$ to the prepotential.[5] The cycles $\{A_I, B^I\}$ which lead to (43) with $\mathcal{F}_{\mathrm{m}}(t_{\mathrm{m}})$ given by (47) are integral. Mirror symmetry thus provides a preferred period vector $\vec{\Pi}$ based on a symplectic basis of $H_3(W,\mathbb{Z})$.

Another set of special loci in the complex structure moduli space are the so-called conifold loci. These are loci in $\mathcal{M}_{cs}(W)$ where $W$ develops a node due to a shrinking three-sphere $\nu = \mathbb{S}^3$ (or, more generally, a shrinking lens space $\mathbb{S}^3/N$ for some $N \in \mathbb{N}$). This singles out a vanishing period $\Pi_{\mathbb{S}^3}$ which can be normalized to be part of an integral symplectic basis. We can also define an affine variable $t_{\mathfrak{c}} = \Pi_{\mathbb{S}^3}/\Pi_{\Gamma_{\mathfrak{c}}} = X_\nu/X_{\mathfrak{c}}^0$ at this point, where, for the application to the gap condition which we shall introduce below, $X_{\mathfrak{c}}^0$ should be a period which stays finite at the conifold locus, and $\Gamma_{\mathfrak{c}}$ should not involve the symplectically dual cycle to the shrinking cycle; see Section 3.2 for a more detailed discussion. Note that the general form of the transition matrix in the hypergeometric one-modulus cases (86) makes it possible to choose $\Gamma_{\mathfrak{c}}$ to be part of the same integral symplectic basis as $\nu$. Hence, the affine variable in the conifold frame can be chosen to be a quotient of two integral symplectic periods, in analogy with the affine coordinate in the maximal unipotent monodromy frame.

---

[5]Due to the negative sign in front of the enumerative contribution to (47) proportional to $\mathrm{Li}_3$, we will have the negative of (47) contribute to the total perturbative prepotential $F^{(0)}$. Also, in the following, whenever a formula for $\mathcal{F}_g$ for general $g$ appears, the $g = 0$ specialization will refer to the negative of (47).

Note that we can write the Gauss-Manin connection in terms of the periods

$$(V^0, V^b, V_b, V_0)^T = (1, t_*^b, \partial_{t_*^b}(2\mathcal{F}_* - t_*^a \partial_{t_*^a} \mathcal{F}_*), 2\mathcal{F}_* - t_*^a \partial_{t_*^a} \mathcal{F}_*)^T, \tag{49}$$

of the normalized holomorphic 3-form $\frac{\Omega}{X_*^0}$ as

$$\partial_{t_*^a} \begin{pmatrix} V^0 \\ V^b \\ V_b \\ V_0 \end{pmatrix} = \begin{pmatrix} 0 & 0 & 0 & 0 \\ \delta_a^b & 0 & 0 & 0 \\ 0 & -C_{t_*^a t_*^b t_*^c} & 0 & 0 \\ 0 & 0 & \delta_{ac} & 0 \end{pmatrix} \begin{pmatrix} V^0 \\ V^c \\ V_c \\ V_0 \end{pmatrix}. \tag{50}$$

This can be seen as the holomorphic version of the first equation from (22). This is compatible with the holomorphic limit of the metric data

$$e^{-K} \to e^{-\mathcal{K}} = X_*^0, \qquad K_i \to \mathcal{K}_i = -\partial_i \log(X_*^0), \qquad \Gamma_{jk}^i \to \Upsilon_{jk}^i = \frac{\partial z^i}{\partial t_*^a} \frac{\partial^2 t_*^a}{\partial z^j \partial z^k}, \tag{51}$$

that can be taken once a frame has been chosen together with associated affine coordinates. In the following, we will use calligraphic symbols instead of straight symbols to indicate holomorphic limits. In particular, we will introduce the higher genus topological string amplitudes $F_g$ and their holomorphic limits $\mathcal{F}_g(X_*)$ below. We will sometimes drop the $*$ in our notation, since the holomorphic limit of an anholomorphic quantity always depends on the choice of a frame. When we wish to emphasize the dependence on the frame, we include the periods defining the frame in parentheses, e.g. $\mathcal{F}_g^{(k)}(X_{\mathfrak{m}}^0, X_{\mathfrak{m}}^1)$ will indicate, in a one-parameter model, the holomorphic limit of the amplitude $F_g^{(k)}(X_{\mathfrak{m}})$ in the MUM frame. We will also require notation for the Kähler gauge invariant quantity $\mathcal{F}_g(t_*) = (X_*^0)^{2g-2} \mathcal{F}_g(X_*)$. To avoid a proliferation of symbols, we somewhat inelegantly distinguish between the section of $\Gamma(\mathcal{L}^{2g+2})$ and its image in $\Gamma(\mathcal{L}^0)$ by indicating the argument as $X_*$ or $t_*$, respectively. We have already used this notation at genus 0 above.

We will end this section by recalling some relation between projective and affine coordinates, following [32]. For ease of presentation, we will drop the star from our notation and assume that a frame has been fixed. Two choices of frame related by a symplectic transformation $\Gamma \in \mathrm{Sp}(b_3, \mathbb{R})$ yield the same expression for (7) and (8) in terms of the associated periods, as follows immediately from (35). Writing

$$\Gamma = \begin{pmatrix} \mathfrak{A} & \mathfrak{B} \\ \mathfrak{C} & \mathfrak{D} \end{pmatrix}, \tag{52}$$

the periods and their covariant derivatives introduced in (33) transform as

$$\begin{aligned} \rho_{I\alpha} &\mapsto \rho_{I\alpha}^\Gamma = \mathfrak{A}_I^J \rho_{J\alpha} + \mathfrak{B}_{IJ} \chi_\alpha^J, \\ \chi_\alpha^I &\mapsto \chi_{\alpha\,\Gamma}^I = \mathfrak{C}^{IJ} \rho_{J\alpha} + \mathfrak{D}_J^I \chi_\alpha^J. \end{aligned} \tag{53}$$

Since $F(X)$ is homogeneous of degree 2, we have

$$\tau_{IJ} := \frac{\partial P_I}{\partial X^J}, \quad X^I \tau_{IJ} = P_J, \quad C_{IJK} := \frac{\partial}{\partial X^K} \tau_{IJ}, \quad X^I C_{IJK} = 0, \tag{54}$$

and the symmetric $(1+r) \times (1+r)$ matrix $\tau_{IJ}$ transforms as

$$\tau^\Gamma = (\mathfrak{A}\tau + \mathfrak{B})(\mathfrak{C}\tau + \mathfrak{D})^{-1}. \tag{55}$$

As pointed out in [32], the inverse of the matrix $\text{Im}\,\tau_{IJ}$, which we will denote by $(\text{Im}\,\tau)^{-1,IJ}$, transforms as

$$(\text{Im}\,\tau_\Gamma)^{-1,IJ} = (\mathfrak{C}\tau + \mathfrak{D})^I{}_K (\mathfrak{C}\tau + \mathfrak{D})^J{}_L (\text{Im}\,\tau)^{-1,KL} - 2i\mathfrak{C}^{IK}(\mathfrak{C}\tau + \mathfrak{D})^J{}_K. \tag{56}$$

As we have just seen, quantities written in terms of the projective coordinates $X^I$ on $\mathcal{M}_{cs}(W)$ transform straightforwardly under the symplectic group. When we write these functions as transcendental functions of global coordinates $z$ on moduli space by substituting explicit expressions for the periods $X^I$, their transformation properties become obscured. However, they undergo monodromy upon analytic continuation around singular points of $\mathcal{M}_{cs}(W)$; the monodromy group will generically be a subgroup of the symplectic group, but will act in accordance with the symplectic action determined before substitution.

As pointed out in [32], where the special geometry formalism was developed in detail both in projective and in affine coordinates, the matrix $\chi^I_\alpha$ appearing in the period matrix (33) and its inverse

$$\chi^I_\alpha \chi^\alpha_J = \delta^I_J, \qquad \chi^I_\alpha \chi^\beta_I = \delta^\beta_\alpha, \tag{57}$$

play an important role in relating projective and affine quantities; in a sense, the two can be used to raise and lower the $I, J \ldots = 0, \ldots, r$ indices while simultaneously converting them to the $\alpha, \beta, \ldots = 0, \ldots, r$ indices. Note that the theorem of Tian and Todorov in combination with the local Torelli theorem imply that the inverse matrix exists outside the discriminant locus of $\mathfrak{L}$. The second relation in (54) implies that $\rho_{I\alpha} = \tau_{IJ}\chi^J_\alpha$. Due to (57), the $\chi^\alpha_I$ transform inversely to $\chi^I_\alpha$ under a symplectic transformation $\Gamma$.

### 2.1.3 The propagators and their transformations

To solve the holomorphic anomaly equations, [22] introduces a set of an-holomorphic sections $S^{ij}, S^i, S$ of $\mathcal{L}^{-2} \otimes \text{Sym}^2(T\mathcal{M}^{1,0}_{cs})$, $\mathcal{L}^{-2} \times T\mathcal{M}^{1,0}_{cs}$ and $\mathcal{L}^{-2}$ respectively, called propagators. They are defined by

$$\partial_{\bar{i}} S^{jk} = C^{jk}_{\bar{i}}, \quad \partial_{\bar{j}} S^k = G_{i\bar{j}} S^{ik}, \quad \partial_{\bar{j}} S = G_{i\bar{j}} S^i, \tag{58}$$

where $C^{jk}_{\bar{i}}$ was introduced in (28). In this section, we will list some properties of the propagators that follow from special geometry, and discuss their holomorphic limit upon the choice of local special coordinates.

The first equation in (58) is integrated by using (26): one observes that all terms contributing to $C^{km}_{\bar{j}} C_{ilm}$ are $\bar{\partial}_{\bar{j}}$-derivatives. Therefore, as long as the matrix $[C_{(i)}]_{lm}$ is invertible for one fixed index $i$, one finds

$$S^{km} = C^{(i)kl}\left( \delta^m_l K_{(i)} + \delta^m_{(i)} K_l - \Gamma^m_{(i)l} + q^m_{(i)l} \right), \tag{59}$$

where $q^m_{(i)l}$ is the holomorphic propagator ambiguity. While the inversion is not necessarily possible over all indices $(i)$, it is easy to see that if $\mathfrak{L}$ is complete and determines the $C_{klm}$ through (8), the inversion is possible at least over one index. If it is possible over more than one index, then the $q^m_{il}$ can always be chosen as rational functions of $z$ so that all $i = 1, \ldots, r$ equations (59) are compatible. We can therefore drop the brackets which specify the special index $i$.

One can integrate (26) with regard to $z_{\bar{j}}$ and show that Christoffel symbols can be written in terms of propagators and $K_i$, up to a holomorphic ambiguity. Furthermore, applying $\bar{\partial}_{\bar{i}}$ to the covariant derivatives $D_i S^{ab}, D_i S^a, D_j S, D_i K_j$ and using (26), (58) and (29), one can express the results as $\bar{\partial}_{\bar{i}}$ derivatives of second order polynomials in the propagators and $K_i$. Integrating, one gets expressions for the covariant derivatives up to additional holomorphic ambiguities.

The $K_i$ dependence in the equations for the derivatives of the propagators can be absorbed in a redefinition [33]

$$\tilde{S}^i = S^i - S^{ij}K_j\,,$$
$$\tilde{S} = S - S^i K_i + \frac{1}{2}S^{ij}K_i K_j\,. \tag{60}$$

This leads to

$$\partial_i S^{jk} = C_{imn}S^{mj}S^{nk} + \delta_i^j\tilde{S}^k + \delta_i^k\tilde{S}^j - q_{im}^j S^{mk} - q_{im}^k S^{mj} + q_i^{jk}\,,$$
$$\partial_i\tilde{S}^j = C_{imn}S^{mj}\tilde{S}^n + 2\delta_i^j\tilde{S} - q_{im}^j\tilde{S}^m - q_{ik}S^{kj} + q_i^j\,,$$
$$\partial_i\tilde{S} = \frac{1}{2}C_{imn}\tilde{S}^m\tilde{S}^n - q_{ij}\tilde{S}^j + q_i\,,$$
$$\partial_i K_j = K_i K_j - C_{ijn}S^{mn}K_m + q_{ij}^m K_m - C_{ijk}\tilde{S}^k + q_{ij}\,. \tag{61}$$

Again as a consequence of the rationality of the coefficients in $\mathfrak{L}$ and (8), the holomorphic ambiguities $q_{jk}^i, q_i^{jk}, q_{ij}, q_j^i$ and $q_i$ can be chosen non-uniquely as rational functions in $z_i$, and one observes that $\tilde{q}_{il}^m = q_{il}^m z_i z_l/z_m, \tilde{q}_k^{ij} = q_k^{ij}z_k/(z_i z_j)$ etc. are polynomials. It is computationally advantageous to choose the degree as low as possible.

Explicit expressions for $\tilde{S}^i$ and $\tilde{S}$ can be obtained by solving successively the first and the second equation in (61) for these propagators:

$$\tilde{S}^k = \frac{1}{2}\left(\partial_k S^{kk} - C_{klm}S^{kl}S^{km} + 2q_{kl}^k S^{lk} - q_k^{kk}\right)\,,$$
$$\tilde{S} = \frac{1}{2}\left(\partial_l\tilde{S}^l - C_{klm}\tilde{S}^k S^{lm} + q_{lm}^l\tilde{S}^m + q_{lm}S^{lm} - q_l^l\right)\,. \tag{62}$$

Note that there is no sum over the index $k$ on the RHS of the first and over the index $l$ in the second of these equations.

The holomorphic limits of the propagators are defined by invoking (51). The holomorphic limit $\mathcal{S}^{km}$ of $S^{km}$ thus is given by

$$\mathcal{S}^{km} = C^{(i)kl}\left(\delta_l^m\mathcal{K}_{(i)} + \delta_{(i)}^m\mathcal{K}_l - \Upsilon_{(i)l}^m + q_{(i)l}^m\right)\,. \tag{63}$$

It satisfies [34]

$$C_{ijm}\mathcal{S}^{mk} - q_{ij}^k = -\chi_I^k\partial_{ij}^2 X^I\,. \tag{64}$$

As proved in Lemma 3.7 in [34], the $\chi_I^k$ are holomorphic, hence independent of the $K_i$. It is thus not necessary to take the holomorphic limit on the RHS. The holomorphic limits of $\tilde{S}^k$ and $\tilde{S}$ follow from (63) and (62). Equation (63) and the last relation in (61) imply

$$C_{ijm}\tilde{\mathcal{S}}^m - q_{ij} = h_I\partial_{ij}^2 X^I\,, \tag{65}$$

with the functions $h_I$ (which are holomorphic and hence independent of the $K_i$, again by Lemma 3.7 in [34]) given by

$$h_I = \chi_I^0 + K_a\chi_I^a\,. \tag{66}$$

We next turn to the transformation behavior of the propagators under the symplectic group. The propagator ambiguities $q_{\cdot}^{\cdot}$ are rational functions in $z$ and therefore do not undergo monodromy upon analytic continuation around singular points of $\mathcal{M}_{cs}$. We hence take them to be frame-independent. From their occurrence in the covariant derivatives $D_i S^{jk}, D_i S^k, D_j S$ and $D_i K_j$, from which (61) follow, we can easily read off which bundles $\mathcal{L}^k\otimes\text{Sym}^m(T\mathcal{M}_{cs})^{1,0}\otimes\text{Sym}^n(T^*\mathcal{M}_{cs})^{1,0}$ they belong to. As all other quantities contributing to the propagator $S^{ij}$ given in (59) are metric, and as $\tilde{S}^k$ and $\tilde{S}$ can be expressed in terms of $S^{ij}$,

metric data, and propagator ambiguities, we can conclude that the propagators we have introduced are invariant under symplectic transformations. On the other hand, the holomorphic limits of these propagators do undergo monodromy, as they involve interesting transcendental functions inherited from the holomorphic limit of the metric data. This observation fits in nicely with the picture proposed in [32] regarding the transformation behavior of the propagators: it was pointed out there that the propagators obtained from the expression

$$\Delta^{IJ} = -(\text{Im}\,\tau)^{-1,IJ} + \mathcal{E}^{IJ}(X)\,, \tag{67}$$

via

$$S_{\mathcal{E}}^{\alpha\beta} = \begin{pmatrix} 2S_{\mathcal{E}} & -S_{\mathcal{E}}^i \\ -S_{\mathcal{E}}^i & S_{\mathcal{E}}^{ij} \end{pmatrix}^{\alpha\beta} = \chi_I^{\alpha}\Delta^{IJ}\chi_J^{\beta}\,, \tag{68}$$

with $\mathcal{E}^{IJ}(X)$ an arbitrary holomorphic function, satisfy (58). If $\mathcal{E}$ is taken to be invariant – we will denote such choices as $\epsilon$ – the behavior of these propagators under symplectic transformations follows easily from (56), (53) and (57):

$$S_{\epsilon,\Gamma}^{\alpha\beta} = S_{\epsilon}^{\alpha\beta} + \chi_I^{\alpha}[(\mathfrak{C}\tau + \mathfrak{D})^{-1}\mathfrak{C}]^{IJ}\chi_J^{\beta}\,. \tag{69}$$

As the monodromy invariance of the topological string amplitudes $F_g$ relies on the propagators being monodromy invariant, [32] proposed that in analogy to the behavior of the almost holomorphic form $\hat{E}_2$, the transformation of $(\text{Im}\,\tau)^{-1,IJ}$ should be cancelled by the transformation of the holomorphic contribution $\mathcal{E}^{IJ}$ to $\Delta^{IJ}$.

We now compare the triple $(S^{ij}, S^i, S)$ that we have introduced above to the triple that follows from $S_{\epsilon}^{\alpha\beta}$. As both triples solve (58), we can conclude that

$$S^{ij} - S_{\epsilon}^{ij} = h^{ij}\,, \tag{70}$$

$$S^i - S_{\epsilon}^i = K_j h^{ij} + h^i\,, \tag{71}$$

$$S - S_{\epsilon} = \frac{1}{2}K_i K_j h^{ij} + K_i h^i + h\,, \tag{72}$$

where the functions $h^{ij}$, $h^i$ and $h$ are holomorphic. Via the inverse of the map $\Delta^{IJ} \mapsto S_{\mathcal{E}}^{\alpha\beta}$ applied in (68), these differences map to a function $\mathcal{E}_S^{IJ}$ which, by holomorphicity of $\chi_I^k$ and $h_I$, is holomorphic. By invariance of the propagators $S^{\alpha\beta}$, $\mathcal{E}_S^{IJ}$ is an explicit realization of the holomorphic contribution to $\Delta^{IJ}$ which cancels the transformation behavior of $(\text{Im}\,\tau)^{-1,IJ}$. And indeed, the transformations of the holomorphic limits of the propagators $(S^{ij}, S^i, S)$ under the symplectic group can easily be calculated. The transformation of the propagators $\mathcal{S}^{ij}$ and $\tilde{\mathcal{S}}^i$ follow from (63) and (65), keeping in mind that $X^I C_{IJK} = 0$ and $C_{IJK}\chi_a^I\chi_b^J\chi_c^K = C_{abc}$. The transformation of $\tilde{\mathcal{S}}$ follows with somewhat more work from the holomorphic limit of its expression in (62); the identities $\chi_L^{\alpha}X^L = \delta_0^{\alpha}$ and $h_L\partial_l X^L = 0$ are useful in this calculation. The transformation properties that thus follow are

$$\begin{aligned} \mathcal{S}^{kl} &\to \mathcal{S}_{\Gamma}^{kl} = \mathcal{S}^{kl} - [(\mathfrak{C}\tau + \mathfrak{D})^{-1}\mathfrak{C}]^{IJ}\chi_I^k\chi_J^l\,, \\ \tilde{\mathcal{S}}^k &\to \tilde{\mathcal{S}}_{\Gamma}^k = \tilde{\mathcal{S}}^k + [(\mathfrak{C}\tau + \mathfrak{D})^{-1}\mathfrak{C}]^{IJ}\chi_I^k h_J\,, \\ \tilde{\mathcal{S}} &\to \tilde{\mathcal{S}}_{\Gamma} = \tilde{\mathcal{S}} - \frac{1}{2}[(\mathfrak{C}\tau + \mathfrak{D})^{-1}\mathfrak{C}]^{IJ}h_I h_J\,. \end{aligned} \tag{73}$$

Mapping (73) to the transformation behavior of the holomorphic untilded propagators by inverting (60), one discovers, as expected, the same transformation behavior as in (69), up to the opposite sign in front of the inhomogeneous term.

## 2.2 The class of hypergeometric one parameter Calabi–Yau 3-folds

Completeness, richness and simplicity make the class of hypergeometric models an ideal sample to study non-perturbative aspects in compact, one-parameter Calabi–Yau 3-folds. The behavior of the perturbative higher genus amplitudes at the various types of singularities which occur in these models is relatively well understood. In particular, a gap condition occurs at some conifold loci, which can be used as boundary condition to fix the holomorphic ambiguity and solve these theories to high genus [24]. Boundary conditions arising at the $z = 0$ MUM point have recently been better understood [25], as have the critical points at $1/z = 0$ in [35], see also [25]. This advance has made it possible to fix the holomorphic ambiguity to higher genus than achieved in [24]. One other useful aspect of these models is that they have four small cousins that describe one parameter families of local Calabi–Yau spaces; these are defined as the total space $\mathcal{O}(-K_{dP}) \to dP$ over del Pezzo surfaces with $dP = \{\mathbb{P}^2, \mathbb{P}^1 \times \mathbb{P}^1, dP_5, dP_6\}$. See [36] for enumerative and [37] for arithmetic aspects of these models, and [23] for a study of their resurgence structure, which we sometimes use as comparison below.

### 2.2.1 Introducing the class

Topological string theories and mirror symmetry on Calabi–Yau manifolds have been developed on one parameter families, starting with the mirror pair $(M, W) = (M_5, \widehat{M_5/\mathbb{Z}_5^3})$ of quintic hypersurfaces in $\mathbb{P}^4$. The quintic example was solved for genus zero in [38], for genus one in [39], for genus two in [22], and for higher but finite genus ($g \leq 53$)[6] in [24]. It was soon realized that in addition to the quintic example, there are twelve additional smooth complete intersection Calabi–Yau families $M$ in (weighted) projective spaces with one Kähler parameter. Their mirrors $W$ all have Picard-Fuchs differential operators of hypergeometric type [40–44], i.e. they are given by

$$L^{(4)} = \theta^4 - \mu^{-1} z \prod_{k=1}^{4} (\theta + a_k), \tag{74}$$

where $z$ parametrizes the complex structure moduli space $\mathcal{M}_{cs}(W) = \mathbb{P}^1 \setminus \{0, \mu, \infty\}$ of $W$. It was later proven [45], independently of geometric representations, that these are the only possible hypergeometric Calabi–Yau motives.[7]

Table 1 contains comprehensive local and global information regarding these hypergeometric motives. The local information at the special points $\{0, \mu, \infty\}$[8] is captured by the Riemann symbol, which for all hypergeometric models is of the form

$$\mathcal{P} \left\{ \begin{array}{ccc} 0 & \mu & \infty \\ \hline 0 & 0 & a_1 \\ 0 & 1 & a_2 \\ 0 & 1 & a_3 \\ 0 & 2 & a_4 \end{array} \right\}. \tag{75}$$

Recall that the columns of $\mathcal{P}$ list the rational local exponents of the four independent entries of the solution vector $f$, $Lf = 0$, at the three special points. According to a theorem of Landman [46], the principal properties of the monodromy matrix $M_*$ transporting $f$ along a path $\gamma_* \in \Pi_1(\mathcal{M}_{cs})$ around the special point $*$ are captured by two minimal integers $1 \leq k < \infty$

---

[6]See footnote 15.

[7]In [45], a fourteenth hypergeometric Calabi–Yau motive was found that does not correspond to a smooth Calabi–Yau family, and to which many geometrical considerations therefore do not apply.

[8]The systems are associated to the hypergeometric functions $_4F_3\left(\begin{smallmatrix} a_1, a_2, a_3, a_4 \\ 1,1,1,1 \end{smallmatrix}; \mu z\right)$ or closely related Meijer G function $G_{4,4}^{n,4}(\mu z)$, with $n = 1, \ldots, 4$ and the same indices, and have no apparent singularities.

Table 1: The manifolds $M$ are generically smooth, complete intersections of $r$ polynomials $P_j$ of degree $d_j$ in the weighted projective ambient spaces $\mathbb{P}^{3+r}(w_1, \ldots, w_{4+r})$. They are denoted as $X_{d_1,\ldots,d_r}(w_1, \ldots, w_{4+r})$. We list the triple intersection number $\kappa = D^3$ of $M$, with $D$ denoting the restriction of the hyperplane class of the ambient space to $M$, the intersection number of $D$ with the second Chern class of the tangent bundle $c_2(TM)$, the Euler number $\chi(M)$, the local exponents at $z = \infty$, the inverse of the location of the third singularity (thus giving all data required to specify the differential operator $L^{(4)}$ in (74)), and finally the degeneration type $dT_\infty$ of the mixed Hodge structure at $z = \infty$.

| # | $M$ | $\kappa$ | $c_2 \cdot D$ | $\chi(M)$ | $a_1, a_2, a_3, a_4$ | $\mu^{-1}$ | $dT_\infty$ |
|---|---|---|---|---|---|---|---|
| 1 | $X_5(1^5)$ | 5 | 50 | $-200$ | $\frac{1}{5}, \frac{2}{5}, \frac{3}{5}, \frac{4}{5}$ | $5^5$ | $O_5^{DG}$ |
| 2 | $X_6(1^4 2^1)$ | 3 | 42 | $-204$ | $\frac{1}{6}, \frac{1}{3}, \frac{2}{3}, \frac{5}{6}$ | $2^4 3^6$ | $O_6^{DG}$ |
| 3 | $X_8(1^4 4^1)$ | 2 | 44 | $-296$ | $\frac{1}{8}, \frac{3}{8}, \frac{5}{8}, \frac{7}{8}$ | $2^{16}$ | $O_8^{DG}$ |
| 4 | $X_{10}(1^3 2^1 5^1)$ | 1 | 34 | $-288$ | $\frac{1}{10}, \frac{3}{10}, \frac{7}{10}, \frac{9}{10}$ | $2^8 5^5$ | $O_{10}^{DG}$ |
| 5 | $X_{4,3}(1^5 2^1)$ | 6 | 48 | $-156$ | $\frac{1}{4}, \frac{1}{3}, \frac{2}{3}, \frac{3}{4}$ | $2^6 3^3$ | $O_{12}$ |
| 6 | $X_{6,4}(1^3 2^2 3^1)$ | 2 | 32 | $-156$ | $\frac{1}{6}, \frac{1}{4}, \frac{3}{4}, \frac{5}{6}$ | $2^{10} 3^3$ | $O_{24}$ |
| 7 | $X_{4,2}(1^6)$ | 8 | 56 | $-176$ | $\frac{1}{4}, \frac{1}{2}, \frac{1}{2}, \frac{3}{4}$ | $2^{10}$ | $C_4$ |
| 8 | $X_{6,2}(1^5 3^1)$ | 4 | 52 | $-256$ | $\frac{1}{6}, \frac{1}{2}, \frac{1}{2}, \frac{5}{6}$ | $2^8 3^3$ | $C_6$ |
| 9 | $X_{3,2,2}(1^7)$ | 12 | 60 | $-144$ | $\frac{1}{3}, \frac{1}{2}, \frac{1}{2}, \frac{2}{3}$ | $2^4 3^3$ | $C_6$ |
| 10 | $X_{3,3}(1^6)$ | 9 | 54 | $-144$ | $\frac{1}{3}, \frac{1}{3}, \frac{2}{3}, \frac{2}{3}$ | $3^6$ | $K_3$ |
| 11 | $X_{4,4}(1^4 2^2)$ | 4 | 40 | $-144$ | $\frac{1}{4}, \frac{1}{4}, \frac{3}{4}, \frac{3}{4}$ | $2^{12}$ | $K_4$ |
| 12 | $X_{6,6}(1^2 2^2 3^2)$ | 1 | 22 | $-120$ | $\frac{1}{6}, \frac{1}{6}, \frac{5}{6}, \frac{5}{6}$ | $2^8 3^6$ | $K_6$ |
| 13 | $X_{2,2,2,2}(1^8)$ | 16 | 64 | $-128$ | $\frac{1}{2}, \frac{1}{2}, \frac{1}{2}, \frac{1}{2}$ | $2^8$ | $M_2$ |
| 14 | $X_{12,2}^{n.s.}(1^4 4^1 6^1)$ | 1 | 46 | $-484$ | $\frac{1}{12}, \frac{5}{12}, \frac{7}{12}, \frac{11}{12}$ | $2^{12} 3^6$ | $O_{12}$ |

(the unipotency index) and $0 \le p \le \dim_{\mathbb{C}}(W)$ (the nilpotency index), such that

$$(M_*^k - 1)^{p+1} = 0. \tag{76}$$

In the one-parameter cases at hand, $k$ and $p$ are determined by the local exponents. For these models, the theory of degenerations of Hodge structures applied to the Calabi–Yau case implies, as reviewed in [47], that only three types of nilpotent degenerations can occur:

- $p = 3$ corresponds to MUM points, also called M-points. $M_*$ has one maximal rank Jordan block. These points occur if all local exponents are equal, i.e. $(a_1, a_2, a_3, a_4) = (a, a, a, a)$.

- $p = 1$ corresponds either to K-points, if $M_*$ has two $2 \times 2$ Jordan blocks, or to conifold or C-points, if $M_*$ has one $2 \times 2$ Jordan block. K-points occur iff two pairs of local exponents are equal, i.e. $(a_1, a_2, a_3, a_4) = (a, a, b, b)$, C-points iff two local exponents are equal and different from the others, i.e. $(a_1, a_2, a_3, a_4) = (a, b, b, c)$, $(a, b, c)$ pairwise unequal.

- At regular points, also called R-points, $p = 0$ and all local exponents are different, i.e. $(a_1, a_2, a_3, a_4) = (a, b, c, d)$, $(a, b, c, d)$ pairwise unequal.

We note that R- and C-points are at finite distance in the Weil–Petersson metric on $\overline{\mathcal{M}_{cs}(W)}$, while K- and M-points are at infinite distance [48]. Conjecturally, this implies that an infinite

number of stable BPS states become massless at the latter. The least common multiple of the denominators of the local exponents determines the unipotency index $k$. Points with $p = 0$ and $k > 1$ are referred to as orbifold points, or O-points.

From the form of the Riemann symbol (75), we can read off that all hypergeometric models have an M-point at $z = 0$ and a C-point at $z = \mu$. Furthermore, from Table 1, we see that all types of nilpotent points occur at $z = \infty$ in the hypergeometric cases; in the last column of the table, we have indicated the nature of the point at $z = \infty$, and included the unipotency index $k$ at this point as a subscript.[9] The superscript $DG$ (for Doron Gepner) indicates that a description of the string world-sheet theory as an exact rational $(\mathcal{N}, \overline{\mathcal{N}}) = (2, 2)$ superconformal field theory is known here.

The relation between the nilpotency index and the structure of the local exponents follows from the theory of differential equations with only regular singular points. An $n$-fold degenerate local exponent $a$ implies the existence of a power series solution $\varpi_0(z) = z^a \sum_{n \geq 0} c_n z^n$ and $n-1$ logarithmic solutions $\varpi_0(z) \log(z)^k + \ldots$, $k = 1, \ldots, n-1$ [49]. The logarithmic branch behavior of the latter is responsible for the nilpotent part of $M_*$. The fact that these theorems apply in the geometric setting follows immediately upon identifying the Picard-Fuchs system satisfied by the periods. Specifically geometric statements, e.g. the absence of the index structure $(a, b, b, b)$ in the geometric setting, are more difficult to prove. The Lefshetz monodromy theorem relates $\varpi_0(z)$ to periods over actual geometric (vanishing) cycles $V$ in $H_3(W, \mathbb{Z})$.

Calabi–Yau 3-fold systems described by differential operators such as (74) have another interesting property that follows from special geometry: the anti-symmetric square of the differential operator corresponds to a one-parameter Calabi–Yau 4-fold operator. Concretely, the $2 \times 2$ minors of the Wronskian $(W)_{ij} = \partial_z^i f_j$ are the solutions of a one parameter $5^{\text{th}}$ order Calabi–Yau 4-fold operator [47]. For the hypergeometric families, the latter can be written in closed form as

$$L^{(5)} = \theta^5 - \frac{z}{\mu}(2\theta + 1)\left[\theta^4 + 2\theta^3 + (5 - \alpha)\theta^2 + (4 - \alpha)\theta - (3 - \alpha - \gamma)\right]$$
$$+ \frac{z^2}{\mu^2}(\theta + 1)\prod_{k=2}^{3}(\theta + a_1 + a_k)(\theta + a_4 + a_k). \tag{77}$$

Here $\alpha = \sum_{i \leq j} a_i a_j$ and $\gamma = \prod_{i=1}^{4} a_i$. This operator is not hypergeometric and corresponds to a one-parameter Calabi–Yau 4-fold with M-point at $z = 0$. Note that the discriminant $\Delta^{(4)} = (1 - \mu^{-1}z)^2$ of the 4-fold is the square of the discriminant of the 3-fold. The existence of $L^{(5)}$ has interesting consequences for the properties of the holomorphic limit of propagators of the 3-fold: they can be expressed up to rational functions as ratios of the 4-fold periods, see equations (99) further below.

### 2.2.2 Global properties of the periods

Note that monodromy preserves the intersection form, so that in an integral symplectic basis, the monodromy matrices $M_* \in \text{Sp}(b_3(W), \mathbb{Z})$ generate (up to van Kampen relations) an irreducible (paramodular) subgroup $\Gamma_W \subset \text{Sp}(b_3(W), \mathbb{Z})$ of the corresponding Siegel modular group.[10] Consequently, one can find an integral symplectic basis by taking an arbitrary basis of solutions $f$ to the Picard-Fuchs system corresponding to the choice of cycles in $H_3(W, \mathbb{C})$, analytically continuing it, and considering linear combinations of these global solutions so that all

---

[9]Clearly, $k = 1$ can always be achieved by a local coordinate change. As the choice of $z$ is canonical however, the entry nevertheless tells us something about the global symmetries of the family.

[10]The subgroup is not necessarily of finite index. Both finite and infinite index arise for the hypergeometric families, see [50].

monodromies are simultaneously in $\text{Sp}(b_3(W), \mathbb{Z})$.[11] However, this procedure is cumbersome, and we prefer invoking homological mirror symmetry and the $\hat{\Gamma}$ class formalism to construct a distinguished integral symplectic basis at an M-point. We will review this procedure in the following for the example of one-parameter hypergeometric models, but the discussion is easily generalizable to multi-moduli cases, see e.g. the appendix of [51].

A $\mathbb{Q}$-basis $L_m$, $m = 0, 1, 2, 3$, of solutions to (74) at the M-point at $z = 0$ can be constructed using the definition

$$(2\pi i)^3 \sum_{k=0}^{\infty} \frac{\prod_{l=1}^{r} \Gamma(d_l(k+\epsilon)+1)}{\prod_{l=1}^{r+4} \Gamma(w_l(k+\epsilon)+1)} z^{k+\epsilon} =: \sum_{m=0}^{\infty} L_m(z)(2\pi i\epsilon)^m, \tag{78}$$

where the weights $w_l$ and the degrees $d_l$ are given in Table 1. In terms of the $L_m$, a canonical integral symplectic basis $\vec{\Pi}$ is given as

$$\vec{\Pi} = \begin{pmatrix} P_0 \\ P_1 \\ X^0 \\ X^1 \end{pmatrix} = \begin{pmatrix} \int_{B^0} \Omega \\ \int_{B^1} \Omega \\ \int_{A_0} \Omega \\ \int_{A_1} \Omega \end{pmatrix} = \begin{pmatrix} \kappa L_3 + \frac{c_2 \cdot D}{12} L_1 \\ -\kappa L_2 + \sigma L_1 \\ L_0 \\ L_1 \end{pmatrix}$$

$$= (2\pi i)^3 \begin{pmatrix} \frac{\zeta(3)\chi(M)}{(2\pi i)^3} & \frac{c_2 \cdot D}{24 \cdot 2\pi i} & 0 & \frac{\kappa}{(2\pi i)^3} \\ \frac{c_2 \cdot D}{24} & \frac{\sigma}{2\pi i} & -\frac{\kappa}{(2\pi i)^2} & 0 \\ 1 & 0 & 0 & 0 \\ 0 & \frac{1}{2\pi i} & 0 & 0 \end{pmatrix} \vec{\Pi}_0. \tag{79}$$

In the last equality, we have related this basis to a $\mathbb{C}$-basis of solutions with rational coefficients which arises as an application of the Frobenius method. Following this method, one solves the differential system via an ansatz involving power series and logarithms, depending on the form of the local exponents, leading to a rational recursion on the expansion coefficients [49]. We refer to $\vec{\Pi}_0$ as a local Frobenius basis. To render it unique, additional normalization conventions must be imposed. The logarithmic structure of this basis takes the form

$$\vec{\Pi}_0(z) = \begin{pmatrix} f_0(z) \\ f_0(z)\log(z) + f_1(z) \\ \frac{1}{2} f_0(z)\log^2(z) + f_1(z)\log(z) + f_2(z) \\ \frac{1}{6} f_0(z)\log^3(z) + \frac{1}{2} f_1(z)\log^2(z) + f_2(z)\log(z) + f_3(z) \end{pmatrix}, \tag{80}$$

for power series $f_n$ normalized by $f_0(0) = 1$ and $f_1(0) = f_2(0) = f_3(0) = 0$. Following equation (79), linear combinations of the Frobenius periods determined in terms of the topological data recorded in Table 1 and the parameter $\sigma := (\kappa \bmod 2)/2$ yield the canonical integral period vector.

At the generic conifold point with local exponents $(0, 1, 1, 2)$, we choose the Frobenius basis in local coordinate $\delta = (1 - z/\mu)$ to have the form

$$\vec{\Pi}_\mu = \begin{pmatrix} 1 + O(\delta^3) \\ \pi_{\mathbb{S}^3}(\delta) \\ \delta^2 + O(\delta^3) \\ \pi_{\mathbb{S}^3}(\delta)\log(\delta) + O(\delta^3) \end{pmatrix}. \tag{81}$$

Here, $\pi_{\mathbb{S}^3}(\delta) = \delta + O(\delta^2)$ is the unique power series which multiplies the logarithm in the logarithmic solution, up to normalization. As indicated, it corresponds to the period over the

---

[11]The above mentioned fact that vanishing periods are proportional to integrals over vanishing cycles $V \in H_3(M, \mathbb{Z})$ greatly simplifies this approach.

cycle with $\mathbb{S}^3$ topology which vanishes when $z$ approaches $\mu$ (shrinking lens spaces do not occur in the class of one-parameter hypergeometric models). This solution is (up to normalization) the analytic continuation of the period $P^0$ introduced in (79); in particular, this implies that the cycle $B^0 \in H_3(W, \mathbb{Z})$ has a representative with topology $\mathbb{S}^3$. The transition matrix $T_\mu$ introduced below in (86) implies this identification and also fixes the normalization constant: $P_0 = \sqrt{\kappa}(2\pi i)^2 \pi_{\mathbb{S}^3} = \Pi_{\mathbb{S}^3}$. The dual cycle $A_0 \in H_3(W, \mathbb{Z})$, whose associated period is the unique holomorphic period at $z = 0$ (up to normalization), has a representative with topology $T^3$. In the $A$ model, $P_0$ is related to the mass of the D6 brane, and $X^0$ to the mass of the D0 brane. These statements are universal for the hypergeometric class of models and also apply to the majority of the models in [52]; the conifold point in question is the closest conifold point to the standard M-point.

It follows from (79), the transition matrix $T_\mu$ introduced below in (86) and from the fact that hypergeometric models have three singular points that $\Gamma_W$ is generated by[12]

$$
M_0 = \begin{pmatrix} 1 & -1 & \frac{\kappa}{6} + \frac{c_2 \cdot D}{12} & \frac{\kappa}{2} + \sigma \\ 0 & 1 & \sigma - \frac{\kappa}{2} & -\kappa \\ 0 & 0 & 1 & 0 \\ 0 & 0 & 1 & 1 \end{pmatrix}, \quad M_\mu = \begin{pmatrix} 1 & 0 & 0 & 0 \\ 0 & 1 & 0 & 0 \\ -1 & 0 & 1 & 0 \\ 0 & 0 & 0 & 1 \end{pmatrix}, \quad M_\infty = (M_0 M_\mu)^{-1}.
$$

(82)

To analytically continue the integral basis of periods $\Pi$ defined in (79) everywhere on $\mathcal{M}_{cs}(W)$, we express it in terms of local Frobenius bases $\Pi_*$ on the overlap of their respective domains of convergence. The transition (or connection) matrices $T_*$ encode the respective linear combinations of the latter yielding the former, $\Pi = T_* \Pi_*$. These matrices can be easily determined numerically. The period matrix (or Wronskian) $W_*$ of a local solution is defined via $[W_*(z)]_{ij} = \partial_z^i \Pi_j^*$, $i, j = 0, \ldots 3$. For the integral basis at $z = 0$, this is

$$
W(z) := \begin{pmatrix} X^0 & X^1 & P_0 & P_1 \\ \partial_z X^0 & \partial_z X^1 & \partial_z P_0 & \partial_z P_1 \\ \partial_z^2 X^0 & \partial_z^2 X^1 & \partial_z^2 P_0 & \partial_z^2 P_1 \\ \partial_z^3 X^1 & \partial_z^3 X^1 & \partial_z^3 P_0 & \partial_z^3 P_1 \end{pmatrix}.
$$

(83)

The local Frobenius solutions at $z = \mu$ and $z = \infty$ yield corresponding expressions $W_\delta(z)$, $W_{w=1/z}(z)$. Evaluating these matrices at the optimal intermediate points with regard to the respective radii of convergence[13] yields

$$
T_\mu = W(\mu/2) W_\delta(\mu/2)^{-1}, \qquad T_\infty = T_\mu W_\delta \left( \mu \frac{1 + \sqrt{5}}{2} \right) W_w \left( \mu \frac{1 + \sqrt{5}}{2} \right)^{-1}.
$$

(84)

To obtain results to at least $n$ significant digits, all series contributing to these expressions must be expanded roughly to order $5n$. To evaluate the logarithms, we choose the points to lie above the real z-axis from $z = 0$ to $z = \mu$ and $z = \infty$, and the branch cuts to run along the positive real axis.

The matrix $T_\infty$ can be calculated exactly using a Mellin-Barnes integral representation for the Meijer G-functions,

$$
G_{44}^{n4} = \frac{1}{2\pi i} \int_{\mathcal{C}} \frac{\Gamma(s)^4 \prod_{k=1}^n \Gamma(a_{\sigma(k)} - s)((-1)^n x)^s}{\prod_{k=n+1}^4 \Gamma(1 - a_{\sigma(k)})} ds.
$$

(85)

---

[12]Note that the Hirzebruch-Riemann-Roch theorem identifies $\frac{\kappa}{6} + \frac{c_2 \cdot D}{12}$ with the holomorphic Euler characteristic $\chi(\mathcal{O}_D)$, which explains its integrality.

[13]To obtain the optimal intermediate point between the conifold point at $z = \mu$ and the third singular point at $w = 1/z = 0$, solve $\mu + \lambda \mu = \frac{1}{0 + \lambda \frac{1}{\mu}}$.

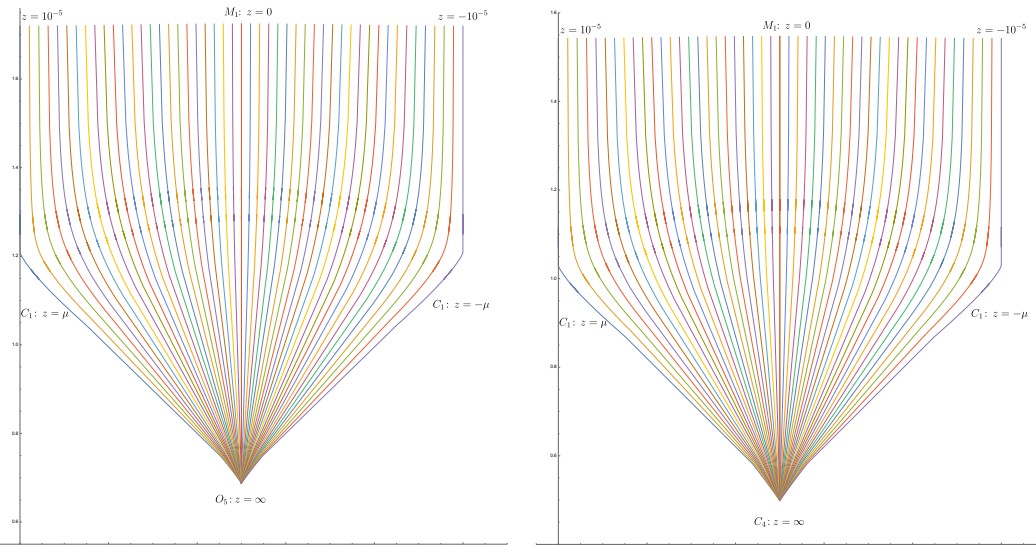

Figure 1: The values of $t = X^1/X^0$ over the cut $z$-plane for the model $X_5(1^5)$ with an O-point of unipotency $l = 5$ at $z = \infty$ (left) and the model $X_{4,2}(1^6)$ with a C-point of unipotency $l = 4$ at $z = \infty$ (right). The plots depict the $t$ image of rays in the $z$-plane emanating from the semi-circle with radius $10^{-5}$ in radial direction towards infinity. $\mathrm{Im}\, t \to \infty$ corresponds to $z = 0$.

Here, $z = \mu/x$, and the values of $n$ and the permutation of the indices $\sigma(k)$ have to be chosen to get four independent solutions. Two choices of contours for $|x| < 1$ and $|x| > 1$ have to be picked to perform the residua in (85). As a consequence, the exact expression for $T_\infty$ with regard to a Frobenius basis $\Pi_\infty$ contains $\pi$ factors, $l$-th unit roots and values of the $\Gamma$-function at rational arguments for O-, C-, K-points, and in addition a $\zeta(3)$ value for M-points at $z = \infty$, see [37] for explicit expressions.

The arithmetic properties of the transition matrix $\vec{\Pi} = T_\mu \vec{\Pi}_\mu$,

$$
T_\mu = \begin{pmatrix}
0 & \sqrt{\kappa}(2\pi\mathrm{i})^2 & 0 & 0 \\
\sigma w^+ + w^- & \sigma a^+ + a^- & \sigma e^+ + e^- & 0 \\
b & c & d & -\sqrt{\kappa}2\pi\mathrm{i} \\
w^+ & a^+ & e^+ & 0
\end{pmatrix},
\tag{86}
$$

are more intriguing [37], as the entries $w^\pm$ and $e^\pm$ are related to periods and quasi periods of modular forms of $\Gamma_0(N)$; the other entries can also be also written as integrals of modular forms using fibering out techniques. With this information, the periods can be approximated to arbitrary precision over the entire moduli space $\mathcal{M}_{cs}(W)$.

We extract graphical information in Figures 1 and 2 for representative models with O-, C-, K-, M-point at $z = \infty$ respectively. We see that the periods of all models behave uniformly, and in fact similarly to the local cases, in the region of convergence of the generic M-point at $z = 0$, which is bounded by the location of the generic conifold, i.e. $|z| \leq \mu$. Along the real axis, the periods of all models exhibit the phases

$$
\phi(P_0) = \begin{cases} \pi, & \text{if } z \leq \mu, \\ 0, & \text{if } \mu < z, \end{cases} \qquad
\phi(P_1) = \begin{cases} -\frac{\pi}{2}, & \text{if } \sigma = 0, \\ \left[-\frac{\pi}{2}, -a\frac{\pi}{2}\right], & \text{if } \sigma = \frac{1}{2}, \end{cases}
\tag{87}
$$

$$
\phi(X^0) = \begin{cases} -\frac{\pi}{2}, & \text{if } z \leq \mu, \\ \left[-\frac{\pi}{2}, -b\frac{\pi}{2}\right], & \text{if } \mu < z, \end{cases}
\tag{88}
$$

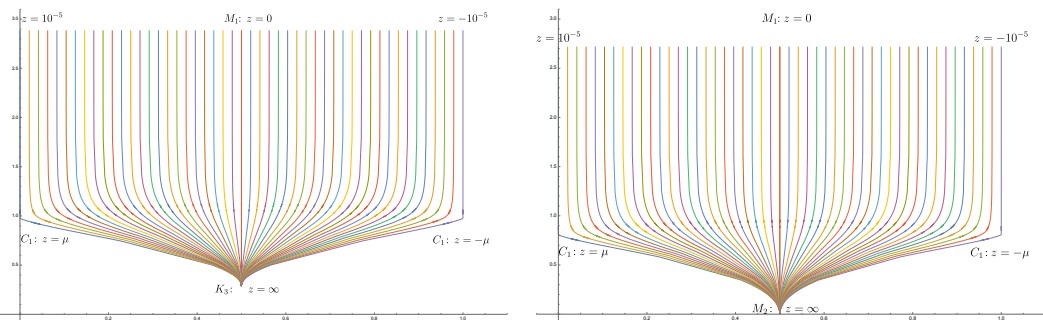

Figure 2: The values of $t = X^1/X^0$ over the cut $z$-plane, plotted as explained in the caption of Figure 1, for the model $X_{33}(1^6)$ with a K-point of unipotency $l = 3$ at $z = \infty$ (left) and the model $X_{2222}(1^9)$ with an M-point of unipotency $l = 2$ at $z = \infty$ (right).

and $\phi(X^1) = 0$, where $a$ and $b$ are slightly model dependent constants close to one: $2/3 < a, b < 9/10$ and the phase is monotonously increasing in the interval. The real values are plotted in Figure 3. In the region $|z| \leq \mu$, which is qualitatively similar for all models, the most significant point for us in the following will be where the masses of the dual D6 and D0 brane cross. We list the numerical values for all crossing points of masses in the region $0 \leq z \leq \mu$ in Table 2.

Close to the MUM point at $z = 0$, the D-brane masses grow exponentially and the relative behavior is not obvious from Figure 3. With the definition of the area below (43), we find for the one parameter case $A = -\frac{1}{2\pi} \log(\text{Re} z)$, i.e. we get a more suitable logarithmic scale $A \to \infty$ for $z \to 0$. The leading order behavior for the real quantity (7) is

$$-i e^{-K} = (2\pi)^6 \left( \frac{4A^3 \kappa}{3} + \frac{2\chi(W)\zeta(3)}{(2\pi)^3} \right) + \mathcal{O}(e^{-A}), \tag{89}$$

which is interpreted as the quantum (instanton) corrected volume. The leading behavior of the D-brane masses is

$$\left( m_{D_6}, m_{D_4}, m_{D_0}, m_{D_2} \right) \sim \left( \frac{1}{4}\sqrt{\frac{\kappa}{3}}A^{3/2}, \frac{1}{4}\sqrt{3\kappa}A^{1/2}, \frac{1}{2}\sqrt{\frac{3}{\kappa}}A^{-3/2}, \frac{1}{2}\sqrt{\frac{3}{\kappa}}A^{-1/2} \right), \tag{90}$$

up to exponentially suppressed terms and lower powers in $A$. Hence, the $D_6, D_4$ brane masses become exponentially large, while the $D_0, D_2$ brane masses are exponentially suppressed.

Another noticeable feature of Figure 3 is that for the orbifold models 1 to 6 of Table 1, none of the D-brane masses in the basis (90) vanish in the region $\mu < |z| \leq \infty$, despite the fact that the absolute values of the periods do vanish, as is clear from the local exponents listed in Table 1. Notice also that Figure 3 looks similar at first glance for all models. The physics however differs from model to model, depending on the existence of integral charge combinations of vanishing D-branes which create the corresponding CKM–boundary conditions at $z = \infty$ and

Table 2: The three rows in this table list $X^0 = z_0/\mu$, where $m_{D6}(z_0) = m_{D0}(z_0)$, $x_4 = z_4/(\mu 10^{-7})$, where $m_{D6}(z_4) = m_{D4}(z_4)$, and $x_2 = z_2/(\mu 10^{-2})$, where $m_{D6}(z_2) = m_{D2}(z_2)$.

| # | 1 | 2 | 3 | 4 | 5 | 6 | 7 | 8 | 9 | 10 | 11 | 12 | 13 | 14 |
|---|---|---|---|---|---|---|---|---|---|----|----|----|----|----|
| $X^0$ | .1409 | .0884 | .0581 | .0228 | .1618 | .0557 | .1997 | .1173 | .2568 | .2153 | .1150 | .00209 | .2998 | .00246 |
| $x_4$ | 1.687 | 1.567 | .9822 | .5478 | 1.983 | 1.991 | 1.786 | 1.393 | 1.777 | 1.898 | 2.164 | 1.646 | 1.683 | .2663 |
| $x_3$ | 5.483 | 1.693 | .4683 | .02596 | 0.5161 | 7.876 | 12.30 | 3.321 | 20.897 | 14.73 | 35.67 | 0.2904 | 28.06 | 0.02376 |

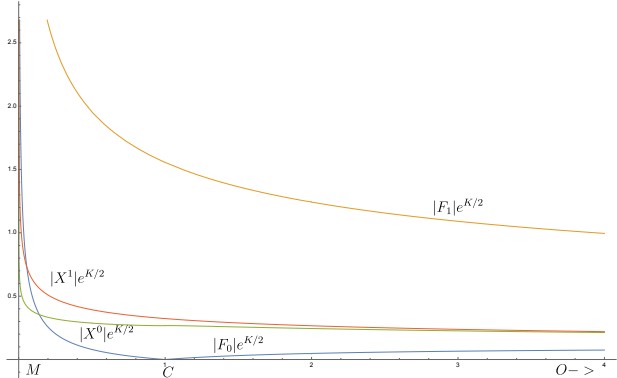

Figure 3: The mass of the D6 brane, $|P_0|e^{K/2}$, the D4 brane, $|P_1|e^{K/2}$, the D1 brane, $|X^1|e^{K/2}$, and the D0 brane, $|X^0|e^{K/2}$, along the real $z$-axis for the model $X_5(1^5)$. Note that up to the factor of $e^{K/2}$, this Figure, together with the information in (87), also gives the values of the corresponding periods. The mass of the D6 brane vanishes at the conifold point $C$. For us, the most interesting point is at $z = 0.14086550800127323750\mu$, where the absolute value of $P_0$ starts exceeding the one of $X^0$, which stays hierarchically smaller as we approach the $M$-point.

dominate the non-perturbative features of the topological string in the region $\mu \leq |z| \leq \infty$. We will discuss both points in section 3.2.2.

### 2.2.3 Wronskian, frames and holomorphic limits

The Wronskian $W$ of a Calabi–Yau motive has very special properties that follow from Griffiths transversality. Let $[W(z)]_{ij} = \partial^i \Pi_j$ be defined as in (83) with $\Pi$ an integral symplectic basis as in (79) with regard to an intersection form $\Sigma$ defined as in (34), or more generally defined for any frame by expanding (36). We can then define the skew symmetric matrix

$$Z = W \cdot \Sigma \cdot W^T, \quad \text{i.e.} \quad [Z(z)]_{ij} = \partial_z^i \Pi^T \cdot \Sigma \cdot \partial_z^j \Pi, \quad \text{for} \quad i,j = 0,\dots,3. \tag{91}$$

As a consequence of (8), we conclude that for any $4^{\text{th}}$ order Calabi–Yau 3-fold operator

$$L^{(4)} = \sum_{i=0}^{4} c_i(z)\partial^i, \tag{92}$$

with rational coefficients $c_i(z)$, hence in particular for operators of the form (74), the $[Z]_{ij}$ are rational up to a prefactor $(2\pi i)^3$, which corresponds to a Tate twist. Indeed, the entries of $Z$ can be calculated recursively by considering (8) and using

$$\Pi^T \cdot \Sigma \cdot \partial_z^i L^{(4)} \Pi = 0, \quad i = 0,\dots \tag{93}$$

In particular, the inverse of the matrix $Z$ can be evaluated in terms of the Yukawa coupling $C = C_{zzz}$ and its derivatives $C' = \partial_z C$ etc. to be

$$Z^{-1} = \frac{(2\pi i)^3}{C} \begin{pmatrix} 0 & \frac{C''}{C} - 2\frac{C'}{C} + \frac{c_2}{c_4} & -\frac{C'}{C} & 1 \\ 2\frac{C'}{C} - \frac{C''}{C} - \frac{c_2}{c_4} & 0 & -1 & 0 \\ \frac{C'}{C} & 1 & 0 & 0 \\ -1 & 0 & 0 & 0 \end{pmatrix}, \tag{94}$$

where $c_2$ and $c_4$ are the coefficients appearing in (92). As a consequence, the inverse of the Wronskian,

$$W^{-1} = \Sigma \cdot W^T \cdot Z^{-1}, \tag{95}$$

depends, up to the dependence on the rational function $c_2/c_4$, linearly on the periods and its derivatives, just as the Wronskian itself. One consequence of (95) and the rationality of $Z$ is that while the Wronskian transforms with right multiplication under monodromy and symplectic frame changes

$$W \to W \cdot M^T \,, \tag{96}$$

its inverse transforms as

$$W^{-1} \to (M^T)^{-1} W^{-1} \,. \tag{97}$$

Another important consequence is that disc amplitudes [53] or multiloop Feynman integrals [54] fulfill an inhomogeneous extension of the linear differential equation, of the form $L^{(4)}\Pi = g(z)$. The variation of constants method applied to solving this inhomogeneous equation implies by (95) that its solutions are simply, up to rational functions and $f(z)$, integrals over periods and their derivatives.

Let now $X_*^0$ and $X_*^1$ be two A-periods in a particular frame, and define by $W^*(z)$ the corresponding Wronskian as in (83). Then, (63) and (62) with $\mathcal{K}_i$ and $\Upsilon_{il}^m$ given by the holomorphic limit (51) imply that the propagators $\mathcal{S}^{zz}, \tilde{\mathcal{S}}^z, \tilde{\mathcal{S}}$ can be expressed by the determinants of the $2 \times 2$ minors of the Wronskian matrix $W^*(z)$, defined to be

$$w_{i,j} = \partial_z^j X_*^0 \partial_z^i X_*^1 - \partial_z^i X_*^0 \partial_z^j X_*^1 \,, \tag{98}$$

in the following manner:

$$
\mathcal{S}^{zz} = -\frac{1}{C}\left(\frac{w_{0,2}}{w_{0,1}} - q_{zz}^z\right), \qquad \tilde{\mathcal{S}}^z = -\frac{1}{C}\left(\frac{w_{1,2}}{w_{0,1}} - q_{zz}\right),
$$
$$
\tilde{\mathcal{S}} = -\frac{1}{2C}\left(\frac{w_{1,3}}{w_{0,1}} - \partial_z q_{zz} - q_{zz}q_{zz}^z\right) + \frac{\partial_z C}{2C^2}\left(\frac{w_{1,2}}{w_{0,1}} - q_{zz}\right) - \frac{q_z^z}{2} \,. \tag{99}
$$

As mentioned above, the entries of the Wronskian are themselves periods of a Calabi–Yau 4-fold; the transcendental functions in (99) are thus affine 4-fold periods. For the hypergeometric cases, the associated $5^{\text{th}}$ order Picard-Fuchs operator is given in (77).

We comment here that $GL(4, \mathbb{C})$ transformations changing the frame can be decomposed into two sets of $GL(2, \mathbb{C})$ transformations acting on the A-periods $X^0, X^1$ and B-periods $P_0, P_1$ respectively, and generators that exchange A- and B-periods. Since the minors are invariant under $SL(2, \mathbb{C})$ and the holomorphic limits of the propagators involve only ratios of the latter and rational functions of the modular parameter $z$, they are invariant even under $GL(2, \mathbb{C})$ actions on the A-periods.

## 3 Solution of the holomorphic anomaly equations

The perturbative free energies $F_g$ ($g \geq 1$) can be computed recursively by using the holomorphic anomaly equations of [22], up to the mentioned holomorphic integration kernel. The starting point of the recursions is the genus zero data defined in Section 2.1. Genus one is special due to the existence of an extra Killing vector field; the holomorphic anomaly here reads [39]

$$\bar{\partial}_{\bar{j}}\partial_i F_1 = \frac{1}{2}C_{ikl}C_{\bar{j}}^{kl} - \left(\frac{\chi(M)}{24} - 1\right)G_{i\bar{j}} \,. \tag{100}$$

Beyond genus 1, one has [22]

$$\bar{\partial}_{\bar{k}}F_g = \frac{1}{2}C_{\bar{k}}^{ij}\left(D_i D_j F_{g-1} + \sum_{r=1}^{g-1} D_i F_r D_j F_{g-r}\right). \tag{101}$$

An immediate consequence of (40), (100), (101) and (28) is that the $F_g(z)$ are sections of $\Gamma(\mathcal{L}^{2-2g})$. For the sum of topological string amplitudes in (1) yielding the perturbative topological string partition function to make sense, the topological string coupling $g_s$ must be a section of $\mathcal{L}$:

$$F_g \in \Gamma(\mathcal{L}^{2-2g}), \qquad g_s \in \Gamma(\mathcal{L}). \tag{102}$$

In this section, we will give a quick review of the solution of the holomorphic anomaly equations using the properties of the propagators defined in Section 2.1.3. In the following sections, we will extend the holomorphic anomaly equations non-perturbatively.

## 3.1 Solving the holomorphic anomaly equations

Using (58) and $G_{i\bar{j}} = \partial_{\bar{j}}\partial_i K = \partial_{\bar{j}} K_i$ we can integrate (100) with regard to $z_{\bar{j}}$ to obtain

$$\partial_i F_1 = C_i - \left(\frac{\chi(M)}{24} - 1\right)K_i, \tag{103}$$

where we have defined

$$C_i = \frac{1}{2}S^{jk}C_{ijk} + f_i^{(1)}. \tag{104}$$

Using (59), the complete integration of (103) can be performed. After taking into account the formula $R_{i\bar{j}} = -\partial_i \bar{\partial}_{\bar{j}} \log \det(G_{a\bar{b}})$, we can write [39]

$$F_1 = -\frac{1}{2}\log\det(G_{a\bar{b}}) - \left(\frac{\chi(M)}{24} - \frac{1}{2}(h_{11}(M) + 3)\right)K + \log(f_1\bar{f}_1). \tag{105}$$

The holomorphic limit of this expression can be taken directly. Alternatively, one can use (103) and the holomorphic limit of $S^{i\bar{j}}$ in (63). One finds,

$$\mathcal{F}_1 = -\frac{1}{2}\log\det\left(\frac{\partial t_*^a}{\partial z_i}\right) + \left(\frac{\chi(M)}{24} - \frac{1}{2}(h_{11}(M) + 3)\right)\log\frac{X_*^0}{(2\pi i)^3} + f_1(z). \tag{106}$$

In the hypergeometric one-parameter cases discussed mostly in the paper, the genus one holomorphic ambiguity is of the form

$$f_1(z) = \log\left(z^{-\frac{c_2 \cdot D + 12}{24}}(1 - \mu^{-1}z)^{-\frac{1}{12}}\right). \tag{107}$$

The exponent of $z$ is fixed by the leading behavior of $\mathcal{F}_1$ at the MUM point at $z = 0$ by (122), the definition of $t_M$ (43) and $c_2 \cdot D$ as given in Table 1. The exponent of the discriminant $(1 - \mu^{-1}z)$ is fixed by the behavior (124) at the conifold point at $z = \mu$. This fixes the genus one perturbative free energy for the hypergeometric one-parameter models in terms of the genus zero data entering $C_{ijk}, S^{ij}$ and $K_i$. The general situation is described in Section 3.2. Note that $f_i^{(1)}(z)$ in (104) is related to $f_1(z)$ in (106) by

$$\partial_i f_1(z) = f_i^{(1)}(z) + \frac{1}{2}q_{il}^l. \tag{108}$$

The free energies of higher genus $g \geq 2$ can be determined up to the holomorphic ambiguity by using the Feynman graphs of an auxiliary theory in which the $S$'s are the propagators (hence their name) and the genus $g$ $n$-point functions called $C_{i_1,\ldots,i_n}^{(g)}$ are the vertices [22]. Alternatively, one can use direct integration [24, 32, 33, 55], which is combinatorially a much more efficient approach. The direct integration method relies on the assumption that there are finite and functionally independent an-holomorphic objects which carry all an-holomorphic

dependence of $F_g$. In the approach of [22], the propagators $S^{ij}, S^i, S$ and the $K_i$ are the anholomorphic objects that appear in the $F_g$. Note that $K_i$ appears in the covariant derivatives that are used to define the $n$-point functions, but in an explicit way it only enters in $\partial_i F_1$. Assuming that these generators are all independent, and using (58), the holomorphic anomaly equations (101) are equivalent to

$$\frac{\partial F_g}{\partial S^{ij}} = \frac{1}{2} D_i D_j F_{g-1} + \frac{1}{2} \sum_{h=1}^{g-1} D_i F_h D_j F_{g-h}, \qquad \frac{\partial F_g}{\partial K_i} + S^i \frac{\partial F_g}{\partial S} + S^{ij} \frac{\partial F_g}{\partial S^j} = 0. \tag{109}$$

If one uses the shifted propagators (60) then the second equation in (109) becomes

$$\frac{\partial F_g}{\partial K_j} = 0, \tag{110}$$

while the first equation reads

$$\frac{\partial F_g}{\partial S^{jk}} - \frac{1}{2} \frac{\partial F_g}{\partial \tilde{S}^k} K_j - \frac{1}{2} \frac{\partial F_g}{\partial \tilde{S}^j} K_k + \frac{1}{2} \frac{\partial F_g}{\partial \tilde{S}} K_j K_k = \frac{1}{2} D_j D_k F_{g-1} + \frac{1}{2} \sum_{h=1}^{g-1} D_j F_h D_k F_{g-h}. \tag{111}$$

Hence, the explicit $K_i$ dependence in the $F_g$ can be removed by expressing them as $F_g(S^{ij}, \tilde{S}^i, \tilde{S})$ for $g > 1$, as observed in [33].

The holomorphic anomaly equations can also be formulated for the sum of the topological string amplitudes, rather than individually for each order in $g_s$ (see [20–22, 56] and in particular [23, 57]). This will be the starting point of the analysis in section 5. As $g = 0$ and $g = 1$ play distinguished roles, we need to introduce

$$\widetilde{F}^{(0)} = F^{(0)} - g_s^{-2} F_0 = \sum_{g \geq 1} F_g g_s^{2g-2}, \tag{112}$$

as well as

$$\widehat{F}^{(0)} = \widetilde{F}^{(0)} - F_1 = \sum_{g \geq 2} F_g g_s^{2g-2}, \tag{113}$$

for this purpose. A difficulty that needs to be overcome in this formulation is to incorporate the dependence on the Kähler potentials which the covariant derivatives on the RHS of (111) entail, even though the objects $\widetilde{F}^{(0)}$ and $\widehat{F}^{(0)}$ are sections of $\mathcal{L}^0$, by (102). For this purpose, we define the operator

$$\hat{D}_i = \mathcal{D}_i - K_i g_s \frac{\partial}{\partial g_s}, \tag{114}$$

where $\mathcal{D}_i$ is the covariant derivative with regard to the Kähler metric, i.e. it involves the Christoffel symbol but not the connection on $\mathcal{L}$. With these ingredients, the holomorphic anomaly equations (111) lead to the master equation

$$\frac{\partial \widehat{F}^{(0)}}{\partial S^{ij}} - \frac{1}{2} K_i \frac{\partial \widehat{F}^{(0)}}{\partial \tilde{S}^j} - \frac{1}{2} K_j \frac{\partial \widehat{F}^{(0)}}{\partial \tilde{S}^i} + \frac{1}{2} \frac{\partial \widehat{F}^{(0)}}{\partial \tilde{S}} K_i K_j = \frac{g_s^2}{2} \hat{D}_i \hat{D}_j \widetilde{F}^{(0)} + \frac{g_s^2}{2} \hat{D}_i \widetilde{F}^{(0)} \hat{D}_j \widetilde{F}^{(0)}, \tag{115}$$

together with the constraint coming from (110),

$$\frac{\partial \widehat{F}^{(0)}}{\partial K_i} = 0. \tag{116}$$

To solve the equations (111), it is convenient to render the implicit dependence on the Kähler potential of the Christoffel connection arising in the first term on the RHS explicit by

invoking (59). Then, as the $K_i$ are assumed to be functionally independent, one can write down equations for the derivatives with respect to individual propagators separately by comparing $K_i$ powers in (111). This leads to [58]

$$\frac{\partial F_g}{\partial S^{ij}} = \frac{1}{2}\partial_i(\partial_j' F_{g-1}) + \frac{1}{2}(C_{ijl}\tilde{S}^{lk} - q_{ij}^k)\partial_k' F_{g-1} + \frac{1}{2}(C_{ijk}\tilde{S}^k - q_{ij})c_{g-1} + \frac{1}{2}\sum_{h=1}^{g-1}\partial_i' F_h \partial_j' F_{g-h},$$

$$\frac{\partial F_g}{\partial \tilde{S}^i} = (2g-3)\partial_i' F_{g-1} + \sum_{h=1}^{g-1} c_h \partial_i' F_{g-h},$$

$$\frac{\partial F_g}{\partial \tilde{S}} = (2g-3)c_{g-1} + \sum_{h=1}^{g-1} c_h c_{g-h}.$$

(117)

Here we used the last equation in (61) and the fact the only explicit $K_i$ dependence arises in $\partial_i F_1$. We also define $c_g$ and the action of $\partial'$ on the free energies for $F_1$ as

$$c_g = \begin{cases} \frac{\chi}{24} - 1, & g = 1, \\ (2g-2)F_g, & g > 1, \end{cases} \qquad \partial_i' F_g = \begin{cases} \partial_i F_g + (\frac{\chi}{24} - 1)K_i, & g = 1, \\ \partial_i F_g, & g > 1. \end{cases}$$

(118)

In this way we only have to use the first three equations of (61) when computing the r.h.s of (117), so that its integration closes manifestly on the genus zero data $S^{ij}$, $\tilde{S}^i$ and $\tilde{S}$, up to a holomorphic ambiguity $f_g(z)$ in each step. Specializing to one modulus one gets e.g. for $F^{(2)}$

$$F_2 = \frac{5}{24}C_{111}^2 (S^{11})^3 + \frac{1}{8}\left(\partial_1 C_{111} - 3C_{111}q_{11}^1 + 4C_{111}f_1^{(1)}\right)(S^{11})^2$$

$$+ \left(\frac{1}{4}q_1^{11}C_{111} + \frac{1}{2}\partial_1 f_1^{(1)} + \frac{1}{2}f_1^{(1)}\left(f_1^{(1)} - q_{11}^1\right) + \frac{1}{2}\left(1 - \frac{\chi}{24}\right)q_{11}\right)S^{11}$$

$$+ \frac{\chi}{48}\left(C_{111}S^{11} + 2f_1^{(1)}\right)\tilde{S}^1 + \frac{\chi}{24}\left(\frac{\chi}{24} - 1\right)\tilde{S} + f_2(z).$$

(119)

Here $f_2(z)$ is the genus two holomorphic ambiguity. This does not look too impressive compared with the multi moduli expression (6.7) in [22], but the point is that the terms in the latter grow factorially with $g$ (see [59] for a recent evaluation of the asymptotic growth based on graph combinatorics for various actions), while the integration of (117) leads to expressions $F_g(S^{ij}, \tilde{S}^i, \tilde{S})$, which are weighted polynomials with rational coefficients of degree $3g-3$ in the $S's$ with weight $(1,2,3)$ respectively (the denominators of these polynomials can be absorbed by writing them in terms of the $C_{ijk}$). This makes the calculation of the higher $F_g$ rather efficient and one can reach up to genus $60-70$ on a good (2023) workstation.

Requiring only a few generators, singled out by automorphic symmetries such as (52), to re-sum a Feynman graph expansion might be viewed as too good to be true for people doing realistic perturbative quantum field theories. The current attempt of reducing the Feynman graph expansion to master integrals is a step in this direction [27]. The comparison is not so far fetched, as the coefficients of a perturbative amplitude in the Laurent expansion, with regard to the dimensional regularization parameter, are in the same class of functions as periods and chain integrals of Calabi–Yau manifolds, whose dimension grows with the loop order [54] [60]. In this context, it is quite remarkable that the propagators needed to increase the loop order in the topological string are also defined, as explained around (99), in terms of affine periods of higher dimensional Calabi–Yau manifolds with Picard-Fuchs operator (77). Note also that despite the polynomial growth of terms contributing to $F_g(z)$, the leading asymptotic of their values at a point in $\mathcal{M}_{cs}$ is expected to go with $(2g)!$, as e.g. at $z = 0$ in (322). The growth of the transcendentality of asymptotic values is again very reminiscent of the situation in more general evaluations of perturbative amplitudes and might be another reason for the factorial growth.

In the compact Calabi–Yau case, fixing the holomorphic ambiguities is the major obstacle to solving for the perturbative free energies. This is in contrast to the local case where the conifold gap conditions and orbifold regularity prove sufficient [61]. We discuss methods to fix the holomorphic ambiguity next.

## 3.2  Fixing the holomorphic ambiguity

Solving the holomorphic anomaly equation (100) at $g = 1$ and determining the topological string amplitudes $F_g$ for $g > 1$ recursively from the equations (101) at higher genus requires specifying holomorphic functions $f_g(z)$ at each genus. These are known as holomorphic ambiguities, as they lie in the kernel of the anti-holomorphic derivatives $\partial_{\bar{k}}$. The holomorphic ambiguities should not contribute to the transformation of the $F_g$ under the monodromy group. They are thus naturally chosen to be rational functions (except for $f_1(z)$, which is the log of a rational function) on $\overline{\mathcal{M}_{cs}(W)}$. Physically, it is expected that their singularities lie at the singularities of the geometry $W$. The latter form a subset of the discriminant locus of $\mathfrak{L}$. The form for $f_1$ for the hypergeometric cases was given in (107); as we will explain below, the general ansatz for $f_{g\geq1}(z)$ is

$$f_{g>1}(z) = \frac{\sum_{k=0}^{2g-2} a_k z^k}{(1-\mu^{-1}z)^{2g-2}} + \sum_{k=1}^{c_m} b_k z^k, \tag{120}$$

where $c_m \in \mathbb{N}$ is a model dependent integer and $a_k, b_k \in \mathbb{Q}$ are constants to be determined by physical or geometric boundary conditions. Typically, these come from expansions of the $F_g$ in a holomorphic limit, which, as explained in Section 2.1.2, requires the choice of a projective frame $*$ of A-periods $X_*^I$, $I = 0, \ldots, r$. Recall our conventions: the topological string amplitudes in the holomorphic limit in the projective frame $*$ are denoted as $\mathcal{F}_g(X_*)$, whereas the Kähler gauge invariant functions of the affine parameter $t_*$, obtained by the homogeneity of the topological string amplitudes, are notationally distinguished by their argument and denoted as

$$\mathcal{F}_g(t_*) = (X_*^0)^{2g-2} \mathcal{F}_g(X_*) \in \Gamma(\mathcal{L}^0). \tag{121}$$

In the following, we will first discuss the boundary conditions which can be imposed at torsion free MUM points using mirror symmetry, as well as the gap conditions imposed at generalized conifolds points, at which a lens space shrinks to zero size in $W$. Then, we will turn to the boundary behavior at more general degenerations, which in the case of one-parameter hypergeometric families occur in the $z = \infty$ region.

As we discuss in section 5.7, knowledge of the leading asymptotics at the MUM point and conifold divisor permits determining the instanton contributions to the associated Borel singularities analytically [20, 21, 23]. It would be interesting to extend this discussion to the regions dominated by the more general degenerations discussed in section 3.2.2 below.

### 3.2.1  Torsion free MUM points and generalized conifold points

The discussion of the boundary behavior of $\mathcal{F}_g$ requires special considerations at genus 1. We will discuss this case first, and then turn to the discussion of higher genus.

**Genus 1 at points of maximal unipotent monodromy (m):**  One can fix the ambiguity $f_1(z)$ at $g = 1$ by imposing the appropriate leading order behavior of $\mathcal{F}_1$ at points of maximal unipotent monodromy. This was determined in [39] by a geometric calculation on the mirror $M$:

$$\mathcal{F}_1(t) = -\frac{2\pi i}{24} \sum_{a=1}^{r} t^a\, c_2 \cdot D_a + \sum_{\beta \in H_2(M,\mathbb{Z})\backslash 0} \left(\frac{n_{0,\beta}}{12} + n_{1,\beta}\right) \mathrm{Li}_1(Q_\beta). \tag{122}$$

Here, $t^a$ are coordinates parameterizing the Kähler cone on $M$. Via mirror symmetry, they can be identified with the affine coordinate $t^a = t^a_m = X^a_m/X^0_m = X^a/X^0$ defined in (43). As explained above, the $X^I$ follow uniquely from the log structure of periods at the MUM point and provide a preferred integral symplectic basis of periods. For the one-parameter models, the first term in (122) determines the exponent of $z$ in (107). If the mirror $M$ of $W$ is known, the intersection of the second Chern class of the tangent bundle of $M$ with the divisors $D_a$ dual to the Mori cone generators can be readily calculated. Else, the constants $c_2 \cdot D_a$ can be read off from the transition matrix $T_0$, whose form for one-parameter models we have explicitly given in (79). The form of the world sheet instanton corrections, the $\mathcal{O}(Q_\beta)$ contribution in (122), is determined by the Gopakumar–Vafa formula, given below in (126), if $H_2(M, \mathbb{Z})$ is torsion free.

**Genus 1 at lens space conifolds ($\mathfrak{c}$):** If a lens space $\nu = S^3/\mathbb{Z}_N$ vanishes at a point $z_\mathfrak{c}$ in $\overline{\mathcal{M}_{cs}(W)}$, the Lefshetz monodromy theorem in odd dimensions implies that the Picard-Fuchs system exhibits a single logarithmic solution $f_{\hat{\nu}} = \pi_\nu \log(z - z_\mathfrak{c}) + \mathcal{O}((z - z_\mathfrak{c}))$, where $\pi_\nu$ is the solution that vanishes at $z = z_\mathfrak{c}$, see e.g. (81). We refer to the locus $z = z_\mathfrak{c}$ as the (generalized, if $N > 1$) conifold locus; it is a divisor in $\overline{\mathcal{M}_{cs}(W)}$. As indicated, this vanishing period corresponds, up to normalization, to the period over the geometric vanishing cycle $\nu$. Together with its dual $\hat{\nu}$, it can be chosen to be part of an integral symplectic basis. Lefshetz theory implies then that $\hat{\nu} \cap \nu = N$. For the one-parameter hypergeometric models, one obtains such an integral basis from the transition matrix (86) acting on the Frobenius solution (81). In particular,

$$\Pi_\nu = \int_\nu \Omega = P_0 = \sqrt{\kappa}(2\pi i)^2 \pi_\nu, \tag{123}$$

and $F_{\hat{\nu}} = \int_{\hat{\nu}} \Omega = X^0$. $\Pi_\nu$ extended to any integral symplectic basis will be the only element of this basis which vanishes on the conifold locus. This follows from the fact that the values in the first column of the transition matrix $T_\mu$, which, by the form of the Frobenius solution (81), give the values of the periods in an integral basis at $z = z_\mathfrak{c}$, are arithmetically independent over $\mathbb{Z}$ [37].

As argued in [62] based on physical considerations, the leading behavior of $\mathcal{F}_1(t_\mathfrak{c})$ near $z_\mathfrak{c}$ is given by

$$\mathcal{F}_1(t_\mathfrak{c}) = -\frac{N}{12} \log(t_\mathfrak{c}) + \mathcal{O}(t_\mathfrak{c}^0). \tag{124}$$

Here, $t_\mathfrak{c} = \Pi_\nu/\Pi_\Gamma$, where $\Pi_\Gamma = \int_\Gamma \Omega$ is a period which is finite at $z = z_\mathfrak{c}$ and does not involve logarithmic terms. In order for (124) to hold, the local exponents have to be $(1, 1, 0, \ldots, 0)$, with 1 for $\Pi_\nu$ and $F_{\hat{\nu}}$ and 0 for the other periods. The exponents thus do not discriminate between vanishing cycles of topology $\mathbb{S}^3/N$ for different $N$. Determining this topology geometrically can be challenging: typically, the mirror manifolds $W$ are obtained as resolved quotients, and the quotient action on the vanishing cycle needs to be determined.[14] A simpler way to determine $N$ is to calculate the (generalized) conifold monodromy: the shift is given by $\hat{\nu} \cap \nu = N$, as a consequence of the Lefshetz theorem. For the hypergeometric cases, it follows from inspection of $M_\mu$ in (82) that $N = 1$ in all cases.

The boundary conditions at $z = 0$ and $z = \mu$ fix the holomorphic ambiguity at genus 1 completely. This is in contradistinction to higher genera, where the behavior at $1/z = 0$ must also be invoked.

Note that the normalization of $t_\mathfrak{c}$ is not fixed in (124), and it follows from (79) and (86) that the cycle $\Gamma$ may be chosen to be an element of the integral symplectic basis containing $\nu$.

---

[14]See the Hulek-Verrill manifold for an easy example in which lens spaces arise upon quotienting [63].

We can hence define

$$t_{\mathfrak{c}} = \frac{\Pi_\nu}{\Pi_\Gamma} = \frac{X_{\mathfrak{c}}^1}{X_{\mathfrak{c}}^0},$$

(125)

with $\nu, \Gamma \in H_3(W, \mathbb{Z})$, $\nu \cap \Gamma = 0$ and $\Pi_\Gamma(z_{\mathfrak{c}}) \neq 0$.

**Genus $> 1$ at points of maximal unipotent monodromy ($\mathfrak{m}$):** The leading behavior of the topological string amplitudes at a MUM point can be extracted from the Gopakumar–Vafa formula [64]

$$\mathcal{F}^{(0)}(X) = \frac{c(t)}{\lambda^2} + l(t) + \sum_{g \geq 0} \sum_{\beta \in H_2(M,\mathbb{Z})} \sum_{k \geq 1} \frac{n_{g,\beta}}{k} \left( 2 \sin \frac{k\lambda}{2} \right)^{2g-2} Q_\beta^k.$$

(126)

Here, we have introduced a Kähler gauge-invariant loop counting parameter

$$\lambda = \frac{(2\pi i)^{3/2} g_s}{X^0} \in \Gamma(\mathcal{L}^0),$$

(127)

and $Q_\beta$ was defined in (48). The Gopakumar–Vafa formula follows from a one-loop Schwinger calculation in the effective $\mathcal{N} = 2$ supergravity theory obtained from compactifying on $M$, the mirror manifold to $W$; this computation yields the couplings between the self-dual curvature and the self-dual field strength of the graviphoton, $\mathcal{F}_g(t_*)(R_+)^2 F_+^{2g-2}$, and is sensitive to the BPS indices $n_{g,\beta} \in \mathbb{Z}$ known as Gopakumar–Vafa invariants. $c(t)$ and $l(t)$ in (126) are the cubic and linear logarithmic contributions to $\mathcal{F}_0$ and $\mathcal{F}_1$ at degree $\beta = 0$, as displayed in (47) and (122); they reflect classical topological data of $M$. While these are not part of the Schwinger-Loop amplitude, they are encoded in the large order behavior of the $n_{g,\beta}$, see [65].

To extract the leading contributions to the topological string amplitudes around a MUM point, we use the identities

$$\sin^{-2} x = \sum_{n=0}^{\infty} (-1)^{n-1} \frac{4(2n-1)B_{2n}(2x)^{2n-2}}{(2n)!}, \quad \sin^2 x = \sum_{n=1}^{\infty} (-1)^{n-1} \frac{(2x)^{2n}}{2(2n)!},$$

(128)

and obtain the so-called bubbling contributions, which for $g \geq 2$ take the form

$$\left( \frac{\lambda}{g_s} \right)^{2-2g} \mathcal{F}_g(X) = \sum_{\beta \in H_2(M,\mathbb{Z})} \left( (-1)^{g+1} \frac{(2g-1)B_{2g}}{(2g)!} n_{0,\beta} + \frac{2(-1)^g n_{2,\beta}}{(2g-2)!} + \cdots \right) \mathrm{Li}_{3-2g}(Q_\beta).$$

(129)

The polylogarithm function $\mathrm{Li}_{3-2g}$ captures the contributions of multi-coverings to $\mathcal{F}_g$. The form of these contributions was proven for genus zero in [66] and for higher genus in [67].

Note that equation (129) contains the leading constant map contributions to the topological string amplitudes at $\beta = 0$. Setting

$$n_{0,0} = \chi(M)/2, \quad n_{g,0} = 0, \quad \text{for} \quad g > 0,$$

(130)

reproduces the constant contribution $\zeta(3)\chi/2$ to $\mathcal{F}_0$ visible in (47). To obtain the constant map contribution at $g > 1$, we need to use $\zeta$-function regularization, replacing the divergent infinite sums that occur by evaluating the polylogarithm at $Q_0 = 1$ by $\zeta(-n) = -B_{n+1}/(n+1)$ for $n \in \mathbb{N}_0$. This yields

$$\left( \frac{\lambda}{g_s} \right)^{2-2g} \mathcal{F}_g(X) = \frac{(-1)^{g+1} B_{2g} B_{2g-2}}{4g(2g-2)(2g-2)!} \chi(M) + \mathcal{O}(Q_\beta) \simeq \frac{(-1)^g (2g)!}{(2\pi)^{4g-2} g(2g-2)} \chi(M) + \mathcal{O}(Q_\beta).$$

(131)

This formula was first obtained in [68] and proved in [67] for smooth $M$ without torsion in $H_2(M,\mathbb{Z})$. The approximation for large $g$ on the RHS of (131) uses

$$\frac{B_{2n}}{(2n)!} = \frac{2(-1)^{n-1}}{(2\pi)^{2n}}\zeta(2n) = \frac{2(-1)^{n-1}}{(2\pi)^{2n}}\prod_{p:\text{prime}}\frac{1}{1-p^{-2n}}, \tag{132}$$

with $\zeta(2n) \sim 1$ to leading order.

Formula (126) is inspired by the calculation of heterotic one-loop amplitudes in heterotic/type II duality [68,69] and needs to be modified, together with the above conclusions drawn from it, if $M$ has torsion classes in $H_2(M,\mathbb{Z})$. This phenomenon already occurs for an example in the class of hypergeometric models: to obtain the topological string amplitudes on $X_{2222}$ to genus $g = 32$, we need to extract boundary conditions from the second MUM degeneration of the mirror family to this geometry; this degeneration is mirror dual to the degeneration of the $X_8$ geometry with 84 nodes and $\mathbb{Z}_2$ torsion [35].

In general, the exact leading behavior (131) of the topological string amplitudes at a MUM point yields one constraint on the holomorphic ambiguity. Many more constraints at MUM points arise by imposing Castelnuovo bounds on the Gopakumar–Vafa invariants. These state, roughly, that $n_{g>g_{\max}(\beta),\beta} = 0$, where $g_{\max}(\beta) \le c\,\beta \cdot \beta$ for some constant $c$. These bounds can be formulated more precisely for the generic MUM point at $z = 0$ of the hypergeometric one parameter models, which correspond to torsionless geometries. Writing $\beta = d \in \mathbb{N}$, the following bounds hold for these cases:

$$g_{\max}(d) \le \left\lfloor \frac{d^2}{2\kappa} + \frac{d}{2}\right\rfloor + 1, \qquad g_{\max}(d) \le \left\lfloor \frac{2d^2}{3\kappa} + \frac{d}{3}\right\rfloor + 1, \quad 0 < d < \kappa. \tag{133}$$

This was observed in [24] and recently proved rigorously in [25] using vanishing results in Donaldson-Thomas theory, see also [70] for a proof in the case of the quintic. As the curves of degree $\beta$ of maximal genus $g_{\max}(\beta)$ are smooth, the associated invariants can be obtained via the formula

$$n_{g,\beta} = (-1)^{\dim_{\mathbb{C}}(\mathcal{M}_\beta)}\chi(\mathcal{M}_\beta), \tag{134}$$

where $\mathcal{M}_\beta$ denotes their deformation space [71]. For the complete intersection cases in Table 1, one gets

$$n_{g_{\max}(\kappa d),\kappa d} = \begin{cases} \frac{\omega(\omega-1)}{2}, & \text{for } d = 1, \\ (-1)^{\kappa d(d-1)/2}\omega\left(\omega + \frac{\kappa d(d-1)}{2}\right), & \text{otherwise,} \end{cases} \tag{135}$$

where $\omega$ is the number of weights equal to one in the weighted projective ambient space. Using the boundary conditions (131), (133), and (135), together with boundary conditions at the conifold at $z = \mu$ and at the orbifold at $z = \infty$, both of which we shall discuss below, one can solve the quintic to genus $g = 53$ [24].[15] The maximal genus to which one can obtain the topological string amplitudes solely by imposing the bounds described above are given in [24,25] for all hypergeometric models.

To fix the holomorphic ambiguity to even higher genus in the class of hypergeometric models, one can invoke modularity of the D4D2D0 rank 1 BPS indices and wall crossing transitions to the rank 1 $\overline{\text{D6}}$D2D0 Donaldson–Thomas invariants that capture the BPS invariants of the topological string [25]. This permits predicting the value of invariants $n_{g_{\max}(\beta)-\Delta,\beta}$ near the Castelnuovo bound (so far only for small $\Delta \in \mathbb{N}$) and imposing these values as constraints on the topological string amplitudes. The currently available data can be found at [72]. More information on attempts to classify non-hypergeometric one-parameter families are summarized in [47] and in the data list [52].

---

[15]In [24], the bound was given as $g = 51$ and the constraints $n_{52,20} = n_{53,20} = 0$ which follow from (133) and permit solving up to $g = 53$ were cited as a prediction.

**Genus > 1 at lens space conifolds (c):** As before, we define the conifold locus as the divisor in $\mathcal{M}_{cs}$ at which a 3-cycle with the topology of a lens space $\nu = S^3/\mathbb{Z}_N$ vanishes. Following [69, 73], it was observed in [24] that the expansion of the perturbative $g > 1$ amplitudes for any choice of local coordinates $t_c$ in the form (125) is given by

$$(X_c^0)^{2g-2}\mathcal{F}_g(X_c) = \frac{(-1)^{g-1}B_{2g}}{2g(2g-2)}\left(\frac{(2\pi i)^{1/2}}{t_c}\right)^{2g-2} + \mathcal{O}((t_c)^0), \qquad g > 1, \tag{136}$$

as long as $\Gamma \in H_3(W, \mathbb{Z})$ satisfies $\nu \cap \Gamma = 0$ and $\Pi_\Gamma(z_c) \neq 0$. $\mathcal{F}_g(t_c)$ is said to satisfy the gap condition. Imposing (136) yields $2g - 1$ conditions on the holomorphic ambiguity $f_{g>1}$ for each conifold divisor. In [74], the lens space factor was interpreted as scaling $g_s$ to $N g_s$.

### 3.2.2 Orbifolds, attractors, small gaps, K-points and torsion MUM points

As we have just reviewed, distinguished frames, specified by a choice of A-periods $X_*^I$ over cycles in $H_3(W, \mathbb{Z})$ are associated to both MUM points and conifold loci. In both cases, $X_*^0$ is the integral period with the most regular behavior at the critical locus, and the additional choice of $X_*^1$ yields an affine $t_*$ coordinate that transforms simply under the local monodromy and stays small near the critical locus. We now want to discuss how these considerations extend to the various types of critical points in the $w = 1/z = 0$ region indicated in the last column of Table 1. We shall then briefly describe what boundary conditions at these points, if any, can be imposed to fix the holomorphic ambiguity for the $F_g$, and highlight features which might be relevant to extending a non-perturbative analysis of the $F_g$ to this region.

The first six models listed in Table 1 exhibit orbifold points at $z = \infty$, four of which have a known rational conformal field theory description (in this case, the orbifold points are also referred to as Gepner points). At these points, $W$ is smooth, hence $(X_o^0)^{2g-2}F_g(X_o)$ must be regular. Postponing the discussion of integrality, a natural local frame given the observations above is defined by the A-periods $X_o^0 = w^{a_1} + \mathcal{O}(w^{a_1+1})$, $X_o^1 = w^{a_2} + \mathcal{O}(w^{a_2+1})$ with $t_o = X_o^1/X_o^0$. The $a_i$ here indicate the local exponents at $z = \infty$, see (75). They are listed for all hypergeometric one-parameter models in Table 1. Note that the factor $(X_o^0)^{2g-2} \sim w^{a_1(2g-2)}$ shields a possible singularity from positive powers of $z = 1/w$ in the second sum contributing to the holomorphic ambiguity in the form (120). Hence, powers of $z$ up to

$$c_m = \lfloor a_1(2g-2) \rfloor, \tag{137}$$

are admissible in the holomorphic ambiguity at genus $g$. This shielding is absent in local models; therefore, $c_m = 0$, and $b_0$ and $a_k$ can be completely determined by imposing the constant map contribution and the (generalized) conifold gap. This is why local models can be solved completely [61]. We can pick $P_1^o = w^{a_3} + \mathcal{O}(w^{a_3+1})$, $P_0^o = w^{a_4} + \mathcal{O}(w^{a_4+1})$ as B-periods, so that the Frobenius basis at $w = 0$ is $\Pi_o^T = (X_o^0, X_o^1, P_1^o, P_0^o)$. The reason that this identification of A- and B-periods is possible is that periods of the form $w^{a_i} + \mathcal{O}(w^{a_i+1})$ transform by a factor of $\exp(2\pi i a_i)$ under the monodromy $w \to e^{2\pi i}w$. Since the right hand side of (36) is monodromy invariant and $a_i + a_{5-i} = 1$, the intersection matrix $\Sigma^*$ can only have anti-diagonal entries. Note that $t_o$, which transforms with a unit root phase under the local monodromy $w \to e^{2\pi i}w$, defines the right coordinate to compare with equivariant orbifold Gromov-Witten theory [75].

To answer the question whether there is a hierarchy of vanishing periods over integral cycles, let us first consider the quintic. Here, $-i e^{-K} = \alpha \Gamma\left(\frac{1}{5}\right)^{10}/(2(2\pi i)^8)|w|^{2/5}$ with $\alpha = \sqrt{50 - 22\sqrt{5}}$ and the masses at $w = 0$ are

$$\left(m_{D_6}, m_{D_4}, m_{D_0}, m_{D_2}\right) = \frac{1}{\sqrt{\alpha}}\left(\sqrt{25 - 11\sqrt{5}}, \sqrt{\frac{15 - \sqrt{5}}{5}}, \sqrt{7 - 3\sqrt{5}}, 2\sqrt{\frac{5 - 2\sqrt{5}}{5}}\right), \tag{138}$$

Table 3: Models with rank two attractor points at their orbifold points. The leading behavior of $\Pi_{\Gamma_i}$ is $\Pi_{\Gamma_i} \sim w^{a_2} + \mathcal{O}(w^{a_3})$. The two cycles have non-vanishing intersection, hence yield mutually non-local charged states. Similarly to conformal Argyres–Douglas points, one does not expect logarithmic behavior of the periods near $w = 0$ even if both charges correspond to massless stable BPS states at $w = 0$. The fact that the entry in the last column is not rational implies that we cannot find an integral combination of $\Pi_{\Gamma_1}, \Pi_{\Gamma_2}$ that vanishes to order $w^{a_3}$.

| $M$ | $a_1, a_2, a_3, a_4$ | $\Pi_{\Gamma_1}$ | $\Pi_{\Gamma_2}$ | $\Gamma_1 \cap \Gamma_2$ | $\Pi_{\Gamma_1}/\Pi_{\Gamma_2}\vert_{w=0}$ |
|---|---|---|---|---|---|
| $X_6$ | $\frac{1}{6}, \frac{1}{3}, \frac{2}{3}, \frac{5}{6}$ | $P_0 - X^1$ | $2P_0 - P_1 + 3X^0$ | $2$ | $e^{2\pi i/6}$ |
| $X_{4,3}$ | $\frac{1}{4}, \frac{1}{3}, \frac{2}{3}, \frac{3}{4}$ | $P_0 - 2X^1$ | $3X^0 - 3X^1 - P_1$ | $1$ | $e^{2\pi i/6} - \frac{i}{3}\sqrt{3}$ |
| $X_{6,4}$ | $\frac{1}{6}, \frac{1}{4}, \frac{3}{4}, \frac{5}{6}$ | $P_0 - X^1$ | $2X^0 - P_0 - P_1$ | $1$ | $i$ |

hence in particular non-zero. Given the form of $e^K$, a massless state at $w = 0$ would arise from a D-brane wrapping an integral cycle with associated period vanishing faster than $w^{1/5}$. The coefficients of the leading $w^{1/5}$ contribution to the four integral periods determined by the action of the transition matrix $T_o$ on the Frobenius basis, $\Pi = T_o \Pi_o$, can be read off (up to an overall normalization) from the entries $(T_o)_{i,1}$. For the quintic, these are independent over $\mathbb{Z}$, demonstrating that such states do not exist. The same is true for the orbifold models $1, 3, 4$ in the Table 1. However, in an extension of the analysis of Moore in [76], we find that besides the model 2, the models 5 and 6 have rank two attractor points at $w = 0$. This implies that there are two independent integral cycles $\Gamma_i \in H_3(W, \mathbb{Z})$, $i = 1, 2$, which intersect (hence yield mutually non-local charged states) and have vanishing central charges $Z_i = e^{K/2}\Pi_{\Gamma_i}$ at $w = 0$. Incidentally, this also implies that one can choose flux potentials that drive the theory into a supersymmetric vacuum at $w = 0$. The corresponding periods for these models are listed in Table 3.

We will next discuss conifold degenerations with small or no gaps. As can be seen from Table 1, the models $7, 8, 9$ have exactly two degenerate local exponents $a_2 = a_3 = \frac{1}{2}$ at $w = 0$. These imply the existence of a vanishing integral period $X_\mathfrak{c}^\nu$ and a dual logarithmic integral period $P_{\hat{\gamma}}^\mathfrak{c}$, just as in the case of the generic conifold. The Frobenius basis is chosen to be of the form $X_\mathfrak{c}^0 = w^{a_1} + \mathcal{O}(w^{a_1+1})$, $2\pi X_\mathfrak{c}^\nu = X_\mathfrak{c}^1 = w^{a_2} + \mathcal{O}(w^{a_2+1})$, $P_1^\mathfrak{c} = X_\mathfrak{c}^1 \log(w) + \mathcal{O}(w^{a_2+1})$ and $P_0^\mathfrak{c} = w^{a_4} + \mathcal{O}(w^{a_4+1})$, yielding $t_\mathfrak{c} = X_\mathfrak{c}^1/X_\mathfrak{c}^0 = w^{\frac{1}{b}}$ and a mirror map $w = t_\mathfrak{c}^b + \mathcal{O}(t_\mathfrak{c}^{2b})$, with $b = 1/(a_2 - a_1)$. The local expansion of the $F_{g>1}$ around the $w = 0$ locus for the models 7 and 8 is of the form

$$(X_\mathfrak{c}^0)^{2g-2} \mathcal{F}_g(X_\mathfrak{c}) = c \frac{(2\pi i)^{3g-3} B_{2g}}{2g(2g-2)} \left(\frac{a}{t_\mathfrak{c}}\right)^{2g-2} + \mathcal{O}\left(t_\mathfrak{c}^{2-2g+\lceil\frac{2g-2}{b}\rceil b}\right), \tag{139}$$

where the parameters $a$ and $c$ are read off at low genus, where imposing them is not required to fix the holomorphic ambiguity. They are given in Table 4. Due to the orbifold structure, the expansion parameter in (139) on top of the $(t_\mathfrak{c})^{2-2g}$ off-set is $(t_\mathfrak{c})^b$. For this reason, (139) imposes only

$$1 + \left\lfloor \frac{2g-3}{b} \right\rfloor, \tag{140}$$

constraints on the holomorphic ambiguity: one coming from the coefficient of the leading pole, the remaining ones from the absence of further negative powers in $t_\mathfrak{c}$.

The three models 9, 10, and 11 have K-points at $w = 0$. The pattern $(a, a, b, b)$ of local exponents with $a, b \in \mathbb{Q}$, $a \neq b$, implies that there exist two independent logarithmic solutions to the Picard-Fuchs system with corresponding vanishing cycles. A natural choice of local

Table 4: Models with conifold rank two attractor points. Unlike at the generalized conifold loci discussed above, $X_{\mathfrak{c}}^{\nu} = \frac{1}{2\pi}\sqrt{w} + \mathcal{O}(w^{\frac{3}{2}})$ and $P_{\hat{\nu}}^{\mathfrak{c}} = \frac{1}{(2\pi)^2 i}\sqrt{w}\log(w) + \mathcal{O}(w^{\frac{3}{2}})$ both vanish at $w = 0$, as does the associated central charge. The numbers $a$, $c$ describe the expansion (139), while $d$ enters in $\mathcal{F}_1(t_{\mathfrak{c}}) = -\frac{d}{12}\log(t_{\mathfrak{c}}) + \mathcal{O}(t_{\mathfrak{c}}^b)$. The $X_{3,2,2}$ case has the same order of the leading singularity as in (139), however a genus dependent coefficient $c$ and no gap as observed in [24].

| $M$ | $a_1, a_2, a_3, a_4$ | $P_{\hat{\nu}}^{\mathfrak{c}}$ | $X_{\mathfrak{c}}^{\nu}$ | $\nu \cap \hat{\nu}$ | a | c | d |
|---|---|---|---|---|---|---|---|
| $X_{4,2}$ | $\frac{1}{4}, \frac{1}{2}, \frac{1}{2}, \frac{3}{4}$ | $P_0 - 2X^1$ | $2P_0 + P_1 - 4X^0$ | 2 | 2 | 2 | 2 |
| $X_{6,2}$ | $\frac{1}{6}, \frac{1}{2}, \frac{1}{2}, \frac{5}{6}$ | $P_0 - X^1$ | $2P_0 + P_1 - 4X^0$ | 3 | 8 | 1 | 1 |
| $X_{3,2,2}$ | $\frac{1}{3}, \frac{1}{2}, \frac{1}{2}, \frac{3}{3}$ | $P_0 - 3X^1$ | $2P_0 + P_1 - 4X^0$ | 1 | $\frac{1}{2}$ | $2 \cdot 7(2^{2g-2} - 1)$ | 12 |

frame is $X_{\mathfrak{k}}^0 = w^{a_1} + \mathcal{O}(w^{a_1+1})$, $X_{\mathfrak{k}}^1 = w^{a_3} + \mathcal{O}(w^{a_3+1})$. As discussed in [48], two combinations $X_{I\mathfrak{k}}^i = (\alpha_i X_{\mathfrak{k}}^0 + \beta_i X_{\mathfrak{k}}^1)$, $i = 0, 1$ span a lattice of vanishing central charges at $w = 0$ over integral cycles, which are mutually local, i.e. have vanishing intersection number. E.g. for $X_{4,4}$, we have with $\alpha_0 = -\frac{1}{2}\Gamma\left(\frac{1}{4}\right)^4$, $\beta_0 = \frac{1}{32}\Gamma\left(\frac{3}{4}\right)^4$, $\alpha_1 = i\alpha_0$ and $\beta_1 = i\beta_0$

$$X_{I\mathfrak{k}}^0 = P_0 - 2X^1, \qquad X_{I\mathfrak{k}}^i = 2X^0 - P_0 - P_1. \tag{141}$$

The universal leading behavior $e^{K/2} \sim -c\frac{1}{w^{a_1}\sqrt{\log(w)}}$, $c > 0$, at all K-points implies that the masses of these states vanish with $-\log(w)^{-1/2}$. The local structure of the periods for the $X_{3,3}$ central charges and masses can be found in [48]. As pointed out in [24], one can choose a frame in which the $(X_{\mathfrak{k}}^0)^{2g-2}\mathcal{F}_g(X_{\mathfrak{k}})$ are regular at the K-points. This regularity imposes the same number of constraints as for the model with orbifold points at $w = 0$.

The last model $M = X_{2,2,2,2}$ in Table 1 exhibits a second MUM point at $w = 0$. One can choose here local coordinates $X_{\mathfrak{m}}^0 = \sqrt{w} + \mathcal{O}(w^{\frac{3}{2}})$, $X_{\mathfrak{m}}^1 = X_{\mathfrak{m}}^0\log(w) + \mathcal{O}(w^{\frac{3}{2}})$ and $q = \exp(X_{\mathfrak{m}}^1/X_{\mathfrak{m}}^0)$. As observed in [24], one has the following constant contribution to the higher genus amplitudes

$$(X_{\mathfrak{m}}^0)^{2g-2}\mathcal{F}_g(X_{\mathfrak{m}}) = \frac{(-1)^{g-1}(2\pi i)^{3g-3}}{2^{2g-2}}(20 - 84 \cdot 2^{2g-2})\frac{B_{2g}B_{2g-2}}{2(2g-2)(2g-2)!} + \mathcal{O}(q). \tag{142}$$

This MUM point has a geometric interpretation provided by the resolution of a degeneration of the $X_8$ model with $n_s = 84$ nodes which carries $\mathbb{Z}_2$ torsion and has a transition to the $X_{2,2,2,2}$ model [35]. Reading off the torsion BPS invariants requires a modification of (126) suggested in [35, 77]. In the case at hand this explains the term inside the parentheses in (142) as due to the corresponding constant map contribution $\frac{\chi(M)}{2} + (1 - 2^{2g-2})n_s$. Likewise, the higher degree torsion BPS invariants can be calculated to some extent from the local expansion (142) and the BPS invariants of $X_8$. These provide new boundary conditions, allowing to solve the model to genus 32 [35].

The general situation indicating the maximal genus to which each model listed in Table 1 can be solved using all available boundary conditions is summarized in Table 1 of [25].[16]

---

[16]The resulting topological string amplitudes for all hypergeometric models can be downloaded from the website http://www.th.physik.uni-bonn.de/Groups/Klemm/data.php. We thank Claude Duhr for letting us run some of these computations on his MacPro workstation.

### 3.2.3  A note on normalization conventions

As $\mathcal{F}_g(t) \in \Gamma(\mathcal{L}^0)$, expressions such as (129) or (136) do not depend on the choice of normalization for the periods. In this sense, the RHS of these expressions can be considered as universal. One can, however, rescale these expressions by rescaling $g_s$ while keeping the topological string amplitude $F^{(0)} = \sum F_g g_s^{2g-2}$ fixed. The expressions we have presented in this section are based on the normalization of the holomorphic anomaly equations that we have introduced in section 3. Rescaling $g_s \to \alpha g_s$ amounts to introducing a factor of $\alpha^2$ on the RHS of (101), as is most clearly visible from the equations in the form (115); the rescaling

$$\bar{\partial}_{\bar{k}} F_g^{(0),\alpha} = \frac{\alpha^2}{2} C_{\bar{k}}^{ij} \left( D_i D_j F_{g-1}^{(0),\alpha} + \sum_{r=1}^{g-1} D_i F_r^{(0),\alpha} D_j F_{g-r}^{(0),\alpha} \right), \tag{143}$$

results in a rescaling of the topological string amplitudes as

$$F_g^{(0),\alpha} = \alpha^{2g-2} F_g. \tag{144}$$

We will mostly work with the convention $\alpha = 1$ in this paper. When we turn to numerics in section 6, however, we will offer the reader a larger palette of normalizations from which to choose.

## 4  Resurgent structures and trans-series

One of the basic ideas of the theory of resurgence is that the large order behavior of a perturbative series encodes secretly information about the non-perturbative sectors of the theory. Perhaps the first quantitative statement of this connection was made in [6], in the case of the perturbative series for the quartic anharmonic oscillator. It turns out that this series contains information about the instanton that implements quantum tunneling in the unstable quartic oscillator with negative coupling constant. Therefore, one can "discover" this instanton sector by looking at perturbation theory. This idea was generalized to many other situations in quantum mechanics and quantum field theory in subsequent work (see e.g. [78] for an early collection of articles) and continues to be explored in many different contexts.

The connection between perturbation theory and non-perturbative sectors can be formulated in a very precise mathematical form mainly due to the work of Jean Écalle [79], and leads to what was called in [80] a *resurgent structure* associated to a perturbative series. We will now review the definition and construction of a resurgent structure.

Let us suppose that we are given a factorially divergent series, of the form,

$$\varphi(z) = \sum_{n \geq 0} a_n z^n, \qquad a_n \sim n!. \tag{145}$$

Such series are also called Gevrey-1. The Borel transform of $\varphi(z)$ is defined by

$$\widehat{\varphi}(\zeta) = \sum_{k \geq 0} \frac{a_k}{k!} \zeta^k. \tag{146}$$

It is by construction a holomorphic function in a neighborhood of the origin. In favorable cases, this function can be analytically continued to the complex $\zeta$-plane (also called Borel plane). This is the property of "endless analytic continuation" in Écalle's theory, and the formal series whose Borel transforms have this property are called *resurgent functions*. The function obtained by analytic continuation of $\widehat{\varphi}(\zeta)$ will have singularities. Let $\Omega$ be the possibly infinite set of indices $\omega$ labeling the singularities $\{\zeta_\omega \in \mathbb{C}\}_{\omega \in \Omega}$ of $\widehat{\varphi}(\zeta)$ in the Borel plane. We shall

assume that all these singularities are logarithmic branch cuts, i.e. the local expansion or $\widehat{\varphi}(\zeta)$ near all $\zeta = \zeta_\omega$ is

$$\widehat{\varphi}(\zeta_\omega + \xi) = -\frac{\mathsf{S}_\omega}{2\pi} \log(\xi)\, \widehat{\varphi}_\omega(\xi) + \text{regular}, \tag{147}$$

where the series

$$\widehat{\varphi}_\omega(\xi) = \sum_{n\geq 0} \widehat{c}_n \xi^n, \tag{148}$$

has a finite radius of convergence. Resurgent functions with this property are called *simple* in Écalle's theory. Note that we might want to make specific choices of normalization for $\widehat{\varphi}_\omega(\xi)$, and that's why we have introduced an additional (in general complex) number $\mathsf{S}_\omega$ in (147), which is called a *Stokes constant*. We will regard $\widehat{\varphi}_\omega(\xi)$ as the Borel transform of

$$\varphi_\omega(z) = \sum_{n\geq 0} c_n z^n, \qquad c_n = n!\,\widehat{c}_n. \tag{149}$$

Through this simple route we have reached the following conclusion: given a formal power series $\varphi(z)$, the expansion of its Borel transform around its singularities generates additional formal power series:

$$\varphi(z) \rightarrow \{\varphi_\omega(z)\}_{\omega\in\Omega}. \tag{150}$$

We call the resulting set of formal power series and Stokes constants the *resurgent structure* associated to the formal power series $\varphi(z)$. In practice, one uses the formal objects

$$\Phi_\omega = e^{-\zeta_\omega/z} \varphi_\omega(z), \tag{151}$$

which include an exponential small term with the location of the Borel singularity. These objects are examples of *trans-series*.

In many situations in physics, the trans-series (151) correspond to non-perturbative sectors of the theory. For example, there are situations in which $\varphi(z)$ is the expansion of the path integral around a reference saddle point, while the $\Phi_\omega$ are obtained by expanding the path integral around a different saddle point and can be identified as instantons. In other examples in quantum field theory, the trans-series correspond to so-called renormalons (see e.g. [81] for recent progress on renormalons from the point of view of resurgence). We should however point out that not all non-perturbative sectors are necessarily included in the resurgent structure associated to the perturbative series, but determining this structure is a crucial and necessary step.

One important application of the resurgent structure of the power series $\varphi(z)$ is a precise determination of its asymptotic behavior. Let $A$ be the Borel singularity which is closest to the origin of the Borel plane, and let

$$\Phi_A = e^{-A/z} \sum_{k\geq 0} c_k z^k, \tag{152}$$

be the corresponding trans-series, where we have assumed again that we have only log singularities. With the analytic structure of the Borel transform $\widehat{\varphi}$ thus governed by the relations (146) and (147), the all-orders asymptotic behavior of the perturbative coefficients $a_n$ is determined by

$$a_n \sim \frac{\mathsf{S}_A}{2\pi} \sum_{k\geq 0} c_k A^{k-n} \Gamma(n-k), \quad n \gg 1, \tag{153}$$

where $\mathsf{S}_A$ is the Stokes constant associated to the singularity. If there are more than one Borel singularity at the same distance from the origin, their contributions simply add in the asymptotic formula (153).

A natural step after building the Borel transform $\varphi(\zeta)$ of a resurgent Gevrey-1 series $\varphi(z)$ is the construction of the Borel resummation, which is the Laplace transform of the Borel transform:

$$s(\varphi)(z) = \int_0^\infty \widehat{\varphi}(\zeta z)\mathrm{e}^{-\zeta}\mathrm{d}\zeta = \frac{1}{z}\int_{\mathcal{C}_{\arg z}} \widehat{\varphi}(\zeta)\mathrm{e}^{-\zeta/z}\mathrm{d}\zeta, \tag{154}$$

where $\mathcal{C}_\theta = \mathrm{e}^{\mathrm{i}\theta}\mathbb{R}_+$. The Borel resummation provides a "sum" of the divergent series, returning a finite value of the series for a finite input value of $z$. Nevertheless, the singularities $\zeta_w$ of the Borel transform may obstruct the integration in the definition of Borel resummation. In the Borel plane, the ray $\mathcal{C}_{\arg\zeta_\omega} = \mathrm{e}^{\mathrm{i}\arg\zeta_\omega}\mathbb{R}_+$ that passes through a singular point $\zeta_\omega$ is called a Stokes ray. When the argument of $z$ equals that of a singular point, so that the contour of integration $\mathcal{C}_{\arg z}$ coincides with a Stokes ray $\mathcal{C}_\theta$, the Borel resummation is not defined. Instead we can define a pair of lateral Borel resummations

$$s_\pm(\varphi)(z) = \int_0^{\mathrm{e}^{\mathrm{i}0\pm}\infty} \widehat{\varphi}(\zeta z)\mathrm{e}^{-\zeta}\mathrm{d}\zeta = \frac{1}{z}\int_{\mathcal{C}_{\theta\pm0}} \widehat{\varphi}(\zeta)\mathrm{e}^{-\zeta/z}\mathrm{d}\zeta, \tag{155}$$

with the integration path bent slightly above or below the Stokes ray. The values of the lateral Borel resummations are certainly different, and the difference, known as the Stokes discontinuity, is exponentially suppressed. By using (147), one finds that the Stokes discontinuity is in fact related to the Borel resummation of the new asymptotic series $\varphi_\omega(z)$ for all the singularities $\omega \in \Omega_\theta$ on the Stokes ray $\mathcal{C}_\theta$:

$$\mathrm{disc}_\theta\,\varphi(z) = s_+(\varphi)(z) - s_-(\varphi)(z) = \mathrm{i}\sum_{\omega\in\Omega_\theta} \mathsf{S}_\omega \mathrm{e}^{-\zeta_\omega/z} s_-(\varphi_\omega)(z). \tag{156}$$

Sometimes it is more convenient to rewrite the formula (156) for the Stokes discontinuity, which involves Borel resummed power series, as a relationship between formal power series themselves. The idea is to introduce the operator $\mathfrak{S}_\theta$ called the Stokes automorphism along the ray $\mathcal{C}_\theta$ by

$$s_+ = s_-\mathfrak{S}_\theta. \tag{157}$$

Then, after dropping the Borel resummation $s_-$, (156) can be written as

$$\mathfrak{S}_\theta(\varphi) = \varphi + \mathrm{i}\sum_{\omega\in\Omega_\theta} \mathsf{S}_\omega \mathrm{e}^{-\zeta_\omega/z}\varphi_\omega, \tag{158}$$

which is a linear transformation of the formal power series. We can further define the (pointed) alien derivative $\dot{\Delta}_{\zeta_\omega}$ associated to the individual singular point $\zeta_\omega, \omega \in \Omega_\theta$ by

$$\mathfrak{S}_\theta = \exp\left(\sum_{\omega\in\Omega_\theta} \dot{\Delta}_{\zeta_\omega}\right). \tag{159}$$

Then the formal power series $\varphi_w$ are also related to the perturbative series $\varphi$ by

$$\dot{\Delta}_{\zeta_\omega}\varphi = \mathsf{s}_\omega\mathrm{e}^{-\zeta_\omega/z}\varphi_w, \tag{160}$$

where the coefficients $\mathsf{s}_\omega$ can be obtained from the Stokes constants $\mathsf{S}_\omega$ appearing in the Stokes discontinuity, see e.g. [9] for further clarifications. The alien derivatives and the Stokes discontinuity therefore contain the same amount of non-perturbative information, and one could be derived from the other. The alien derivatives, however, have better analytic properties. The most important property is that they are indeed derivations, and they satisfy the Leibniz rule when acting on a product of formal power series:

$$\dot{\Delta}_{\zeta_\omega}(\phi_1(z)\phi_2(z)) = \left(\dot{\Delta}_{\zeta_\omega}\phi_1(z)\right)\phi_2(z) + \phi_1(z)\left(\dot{\Delta}_{\zeta_w}\phi_2(z)\right). \tag{161}$$

# 5 Trans-series solution to the holomorphic anomaly equations

*Note: In all other sections of this paper, $(X^I, P_I)$ denote integral periods as determined at a MUM point in accordance with (43). When we wish to leave the basis of periods unspecified, we write $(X^I_*, P^*_I)$. To avoid a proliferation of $*$'s in this section, we shall take the starless $(X^I, P_I)$ to denote the periods corresponding to the holomorphic limit we wish to consider.*

## 5.1 Resurgence and topological strings

As we discussed in section 2, the free energies $F_g(z)$ appearing in the formal power series (1) can be computed in different frames, and they are well-defined in a neighbourhood of a base point in the Calabi–Yau moduli space. For a fixed value of $z$ in this neighbourhood, we expect to have the factorial growth

$$F_g(z) \sim (2g)!\,, \tag{162}$$

which has been verified in many different situations in the local case (see e.g. [17, 18, 21, 23, 82]). Some evidence for this growth in the case of the quintic Calabi–Yau manifold was also found in [83].

In view of the factorial growth (162), we can ask what the resurgent structure of the series (1) is, which we will regard as a Gevrey-1 series in $g_s$ in which the odd terms vanish (in asking this question, we assume the endless analytic continuation of the Borel transform). This turns out to be a difficult mathematical problem. One case where a detailed solution can be found is when the Calabi–Yau manifold is the resolved conifold. This was first addressed by Pasquetti and Schiappa in [84] (see [85–87] for further results building on [84]). In that case, the functions $F_g(z)$ are known for all $g \geq 0$ and the Borel transform can be calculated explicitly.

Let us review the result of [84], since it will be useful in the following. We recall that $X^0 = cnst$ since we are considering a toric Calabi–Yau manifold, and $X^1 \sim t$, where $t$ can be identified with the complexified Kähler parameter of the resolved conifold. Then, the Borel singularities are located at $\mathcal{A} = \ell \mathcal{A}_m$, where $\ell \in \mathbb{Z}\backslash\{0\}$ and

$$\mathcal{A}_m \sim (X^1 + mX^0)\,, \quad m \in \mathbb{Z}\,, \tag{163}$$

with the normalization constant depending on a choice of conventions. The trans-series associated to the Borel singularity is of the form

$$\Phi_{\mathcal{A}} = \frac{1}{2\pi} \left( \frac{1}{\ell} \frac{\mathcal{A}_m}{g_s} + \frac{1}{\ell^2} \right) e^{-\ell \mathcal{A}_m}\,. \tag{164}$$

With this choice of normalization for the trans-series, the Stokes constant is the same for all $\mathcal{A}$, and it has the value

$$\mathsf{S}_{\mathcal{A}} = 1\,. \tag{165}$$

We can think about (164) as an $\ell$-instanton amplitude for the topological string. Although this result is for a very simple Calabi–Yau 3-fold, it will have a counterpart in the more general case studied in this paper, as we will see below.

For more general Calabi–Yau 3-folds, we do not have the luxury of analytic results in $g_s$ for the free energies, and other methods to find the resurgent structure have to be found. It turns out that there is one aspect of the resurgent structure which is more amenable to a formal treatment, and this is the determination of the trans-series (151) associated to a given singularity. A powerful method to address this problem was proposed by Couso, Edelstein, Schiappa and Vonk (CESV) in [20,21]. Their idea is inspired by the theory of ODE's of Écalle (see [7,88,89] for introductions to this theory). Let us suppose that we have an ODE with an irregular singular point at $z = 0$, so that the power series solution around this point, $y_p(z)$, is

factorially divergent. Then, much information about the resurgent structure of $y_p(z)$ can be obtained by considering a trans-series ansatz for the solution, of the form

$$y(z) = y_p(z) + C e^{-\mathcal{A}/z} y^{(1)}(z) + \mathcal{O}(C^2),\tag{166}$$

where $C$ is a so-called trans-series parameter. Since $y_p(z)$ is a solution, $A$ and $y^{(1)}(z)$ are found by solving the linearized ODE around $y_p(z)$. One can then show that $e^{-\mathcal{A}/z} y^{(1)}(z)$ is the trans-series associated to a Borel singularity at $\mathcal{A}$ (the proof of this statement uses the so-called "bridge equations", see e.g. [7, 88]).

Note that, in Écalle's theory, the variable appearing in the ODE is the expansion parameter in the divergent series. We have a similar situation in non-critical string theory, where one can often obtain the free energy as a solution to an ODE in the string coupling constant. In the case of topological strings with a Calabi–Yau target, we do not have such an ODE. Rather, we have the holomorphic anomaly equations reviewed in the last section. The proposal of CESV is to solve these equations with an ansatz similar to (166), involving exponentially small terms. Detailed calculations in [21] in the local case provided evidence that this ansatz leads to solutions that are indeed related to the trans-series attached to the singularities. The picture of [21] was developed in [23] in two ways. First, the trans-series solutions were obtained in closed form. Second, a precise conjecture was put forward relating the resurgent structure to the trans-series solution to the holomorphic anomaly equations. This was done for local Calabi–Yau models with one modulus. In this paper, we will extend the results of [20, 21, 23] to arbitrary Calabi–Yau 3-folds.

## 5.2 Warm-up: trans-series in the one-modulus case

Before presenting the general case, we will focus on compact Calabi–Yau manifolds with a single modulus, and we will solve for the trans-series ansatz "by hand." In the next sections we will introduce and develop an operator formalism which generalizes [23, 57].

The starting point of the analysis is the holomorphic anomaly equation (115) written in terms of the total topological free energy. As in [20, 21, 23], we consider the following trans-series ansatz for the solution of this master equation:

$$F = \sum_{\ell \geq 0} \mathcal{C}^\ell F^{(\ell)} = F^{(0)} + \mathcal{C} F^{(1)} + \mathcal{O}(\mathcal{C}^2),\tag{167}$$

where $F^{(1)}$ is of the form

$$F^{(1)} = e^{-\mathcal{A}/g_s} \sum_{n \geq 0} F_n^{(1)} g_s^{n-1}.\tag{168}$$

This is similar to the ansatz (166) for ODEs. We will refer to $F^{(1)}$ as the one-instanton correction. By $g_s \in \Gamma(\mathcal{L})$, this ansatz requires $\mathcal{A} \in \Gamma(\mathcal{L})$, i.e. $\mathcal{A}$ should be a period of the Calabi–Yau manifold. Furthermore, by $F^{(0)} \in \Gamma(\mathcal{L})$ and (167), we must have $F_n^{(1)} \in \Gamma(\mathcal{L}^{1-n})$. We will find below that independently of the choice of leading power of $F^{(1)}$ in $g_s$, $F_0^{(1)} \sim \mathcal{A}$. This confirms that having the series (168) start with the power $g_s^{-1}$ is the correct choice. After plugging the trans-series ansatz in (115), we find that the one-instanton correction solves the linearization of the master equation,

$$\frac{\partial F^{(1)}}{\partial S^{ij}} - \frac{1}{2} K_i \frac{\partial F^{(1)}}{\partial \tilde{S}^j} - \frac{1}{2} K_j \frac{\partial F^{(1)}}{\partial \tilde{S}^i} + \frac{1}{2} \frac{\partial F^{(1)}}{\partial \tilde{S}} K_i K_j = \frac{g_s^2}{2} \hat{D}_i \hat{D}_j F^{(1)} + g_s^2 \hat{D}_i \widetilde{F}^{(0)} \hat{D}_j F^{(1)},\tag{169}$$

as well as

$$\frac{\partial F^{(1)}}{\partial K_i} = 0.\tag{170}$$

We will solve this equation in full generality in section 5.3, but to get a taste for the structure of the trans-series, we will solve for the very first orders of the one-instanton correction "by hand," and in the one-modulus case, as in [20, 21].

In the one-modulus case, the propagators are $S^{zz}$, $\tilde{S}^z$ and $\tilde{S}$. The first equation which follows from (169) is

$$\frac{\partial \mathcal{A}}{\partial S^{zz}} = \frac{\partial \mathcal{A}}{\partial \tilde{S}^z} = \frac{\partial \mathcal{A}}{\partial \tilde{S}} = \frac{\partial \mathcal{A}}{\partial K_z} = 0 \,. \tag{171}$$

This means that $\mathcal{A}$ is a purely holomorphic object. In addition, we find

$$\frac{\partial F_0^{(1)}}{\partial S^{zz}} - \frac{\partial F_0^{(1)}}{\partial \tilde{S}^z} K_z + \frac{1}{2} \frac{\partial F_0^{(1)}}{\partial \tilde{S}} K_z^2 = \frac{1}{2} (D_z \mathcal{A})^2 F_0^{(1)} \,, \tag{172}$$

$$\frac{\partial F_0^{(1)}}{\partial K_z} = 0 \,,$$

where

$$D_z \mathcal{A} = \mathcal{D}_z \mathcal{A} + K_z \mathcal{A} \,. \tag{173}$$

Note that $\mathcal{A}$ indeed transforms as a section of $\mathcal{L}$, as required by the consistency the ansatz (168). As we will see, $\mathcal{A}$ is in fact an integral period of the Calabi–Yau manifold.[17] By expanding the first equation in (172) and comparing terms in $K_z$, we find three equations

$$\frac{\partial F_0^{(1)}}{\partial S^{zz}} = \frac{1}{2} (\partial_z \mathcal{A})^2 F_0^{(1)} \,, \qquad \frac{\partial F_0^{(1)}}{\partial \tilde{S}^z} = -\mathcal{A} \partial_z \mathcal{A} F_0^{(1)} \,, \qquad \frac{\partial F_0^{(1)}}{\partial \tilde{S}} = \mathcal{A}^2 F_0^{(1)} \,, \tag{174}$$

which integrate to

$$F_0^{(1)} = f_0^{(1)} \exp\left( \frac{1}{2} (\partial_z \mathcal{A})^2 S^{zz} - \mathcal{A} \partial_z \mathcal{A} \tilde{S}^z + \mathcal{A}^2 \tilde{S} \right) \,. \tag{175}$$

Here, $f_0^{(1)}(z)$ is independent of the propagators and of $K_z$, so it is an undetermined function of the modulus $z$. It is the non-perturbative counterpart of the holomorphic ambiguity, and it has to be fixed with additional information. This was done in [20, 21] in the local case by relying on a conjecture which can be generalized to the compact case. The conjecture goes as follows. Since $\mathcal{A}$ is a period of the Calabi–Yau manifold, one can define a frame $(X_\mathcal{A}^I, P_I^\mathcal{A})$ such that

$$\mathcal{A} = \aleph X_\mathcal{A}^1 \,. \tag{176}$$

The proportionality constant $\aleph$ depends on the normalization of $g_s$, as we explained at the end of section 3.2. For the conventions chosen in section 3.1, we have

$$\aleph = i\sqrt{2\pi i} \,. \tag{177}$$

We will denote the holomorphic limits of the propagators in this frame as $\mathcal{S}_\mathcal{A}^{zz}$, $\tilde{\mathcal{S}}_\mathcal{A}^z$, $\tilde{\mathcal{S}}_\mathcal{A}$. The holomorphic limit of the coefficient (175) is then

$$\mathcal{F}_{0,\mathcal{A}}^{(1)} = f_0^{(1)} \exp\left( \frac{1}{2} (\partial_z \mathcal{A})^2 \mathcal{S}_\mathcal{A}^{zz} - \mathcal{A} \partial_z \mathcal{A} \tilde{\mathcal{S}}_\mathcal{A}^z + \mathcal{A}^2 \tilde{\mathcal{S}}_\mathcal{A} \right) \,. \tag{178}$$

The conjecture states that in any such frame, the one-instanton amplitude has the form that was found in [84] for the resolved conifold, i.e. $\mathcal{F}_\mathcal{A}^{(1)}$ must take the form (164) with $\ell = 1$.

---

[17]Recall that integral periods are defined only up to one overall normalization, which corresponds to the choice of normalization of the holomorphic 3-form $\Omega$. Having fixed a normalization for $\Omega$ in (79), the relation between $\mathcal{A}$ and integral periods in this normalization will involve a proportionality constant.

We will demonstrate that this conjecture holds for the dominant instanton actions both in the vicinity of MUM points and of conifold divisors in section 5.7, and provide further numerical evidence for its validity in section 6. For now, let us see how this conjecture allows us to fix the non-perturbative holomorphic ambiguity. It implies

$$\mathcal{F}_{\mathcal{A}}^{(1)} = \frac{1}{2\pi} \left( \frac{\mathcal{A}}{g_s} + 1 \right) e^{-\mathcal{A}/g_s}, \tag{179}$$

and therefore (recall the expansion introduced in (168)),

$$\mathcal{F}_{0,\mathcal{A}}^{(1)} = \frac{1}{2\pi} \mathcal{A}. \tag{180}$$

This fixes

$$F_0^{(1)} = \frac{1}{2\pi} \mathcal{A} \exp\left( \frac{1}{2} (\partial_z \mathcal{A})^2 (S^{zz} - \mathcal{S}_{\mathcal{A}}^{zz}) - \mathcal{A}\partial_z \mathcal{A} (\tilde{S}^z - \tilde{\mathcal{S}}_{\mathcal{A}}^z) + \mathcal{A}^2 (\tilde{S} - \tilde{\mathcal{S}}_{\mathcal{A}}) \right). \tag{181}$$

It is not very difficult to obtain a general equation for the coefficient $F_n^{(1)}$, $n \geq 1$. From the coefficient $g_s^{n-1}$, we conclude that $F_n^{(1)} \in \Gamma(\mathcal{L}^{1-n})$. Hence, the covariant derivative acts as

$$D_z F_n^{(1)} = (\mathcal{D}_z - (n-1)K_z) F_n^{(1)}. \tag{182}$$

We then find the following holomorphic anomaly equation for the coefficients $F_n^{(1)}$:

$$\frac{\partial F_n^{(1)}}{\partial S^{zz}} - \frac{\partial F_n^{(1)}}{\partial \tilde{S}^z} K_z + \frac{1}{2} \frac{\partial F_n^{(1)}}{\partial \tilde{S}} K_z^2 = \frac{1}{2} (D_z \mathcal{A})^2 F_n^{(1)} + \frac{1}{2} D_z^2 F_{n-2}^{(1)} - \frac{1}{2} \left( D_z^2 \mathcal{A} + 2 D_z \mathcal{A} D_z \right) F_{n-1}^{(1)}$$

$$- D_z \mathcal{A} \sum_{\ell=1}^{\left[\frac{n+1}{2}\right]} D_z F_\ell^{(0)} F_{n+1-2\ell}^{(1)} + \sum_{\ell=1}^{\left[\frac{n}{2}\right]} D_z F_\ell^{(0)} D_z F_{n-2\ell}^{(1)}, \tag{183}$$

as well as

$$\frac{\partial F_n^{(1)}}{\partial K_z} = 0. \tag{184}$$

In this section, we will sometimes denote the perturbative topological string amplitudes $F_g$ as $F_g^{(0)}$, to clearly set them apart from their non-perturbative counterparts. The equation for $n = 0$ was solved above. For $n = 1$ we find,

$$\frac{\partial F_1^{(1)}}{\partial S^{zz}} - \frac{\partial F_1^{(1)}}{\partial \tilde{S}^z} K_z + \frac{1}{2} \frac{\partial F_1^{(1)}}{\partial \tilde{S}} K_z^2 = \frac{1}{2} (D_z \mathcal{A})^2 F_1^{(1)} - \frac{1}{2} \left( D_z^2 \mathcal{A} + 2 D_z \mathcal{A} D_z \right) F_0^{(1)} - D_z \mathcal{A} D_z F_1^{(0)} F_0^{(1)}. \tag{185}$$

To solve this, we proceed as in [20, 21]: we factor out the exponent appearing in (181), and we write

$$F_1^{(1)} = e^{\phi_0^{(1)}} \Phi_1^{(1)}, \tag{186}$$

where

$$\phi_0^{(1)} = \frac{1}{2} (\partial_z \mathcal{A})^2 (S^{zz} - \mathcal{S}_{\mathcal{A}}^{zz}) - \mathcal{A}\partial_z \mathcal{A} (\tilde{S}^z - \tilde{\mathcal{S}}_{\mathcal{A}}^z) + \mathcal{A}^2 (\tilde{S} - \tilde{\mathcal{S}}_{\mathcal{A}}), \tag{187}$$

and $\Phi_1^{(1)}$ satisfies

$$\frac{\partial \Phi_1^{(1)}}{\partial S^{zz}} - \frac{\partial \Phi_1^{(1)}}{\partial \tilde{S}^z} K_z + \frac{1}{2} \frac{\partial \Phi_1^{(1)}}{\partial \tilde{S}} K_z^2 = -\frac{1}{2} \mathcal{A} D_z^2 \mathcal{A} - D_z \mathcal{A} \left( D_z \mathcal{A} + \mathcal{A}\partial_z \phi_0^{(1)} \right) - \mathcal{A} D_z \mathcal{A} D_z F_1^{(0)}. \tag{188}$$

The equation simplifies significantly if we use that

$$\mathcal{A}'' = \mathcal{A} \left( C_z \tilde{S}_{\mathcal{A}}^z - h_{zz} \right) - \mathcal{A}' \left( C_z \mathcal{S}_{\mathcal{A}}^{zz} - s_z^{zz} \right). \tag{189}$$

This follows from the explicit formulae for the propagators (99) in the one-modulus case. The non-perturbative ambiguity is fixed by the boundary condition (179), which implies

$$\mathcal{F}_{1,\mathcal{A}}^{(1)} = \frac{1}{2\pi}\,. \tag{190}$$

One then finds

$$\Phi_1^{(1)} = \frac{1}{2\pi}\Bigg(1 - \Delta(\mathcal{A}')^2 + 2\Delta_1 \mathcal{A}\mathcal{A}' - 2\mathcal{A}^2\Delta_2 - \frac{1}{6}C_z\mathcal{A}(\Delta\mathcal{A}' - \Delta_1\mathcal{A})^3$$
$$- \mathcal{A}\left(\frac{1}{2}C_z S^{zz} + f_z^{(1)}\right)(\mathcal{A}'\Delta - \mathcal{A}\Delta_1) + \mathcal{A}\left(\frac{\chi}{24} - 1\right)\left(2\mathcal{A}\Delta_2 - \Delta_1\mathcal{A}'\right)\Bigg), \tag{191}$$

where we have denoted

$$\Delta = S^{zz} - \mathcal{S}_{\mathcal{A}}^{zz}\,, \qquad \Delta_1 = \tilde{S}^z - \tilde{\mathcal{S}}_{\mathcal{A}}^z\,, \qquad \Delta_2 = \tilde{S} - \tilde{\mathcal{S}}_{\mathcal{A}}\,. \tag{192}$$

This result is valid for any compact Calabi–Yau manifold with a single modulus.

The formula obtained above for $F_n^{(1)}$, with $n = 0, 1$, can be checked in the local limit considered in [20,21,23], by setting to zero the propagators $\tilde{S}^z$ and $\tilde{S}$, as well as the holomorphic functions $h_{zz}$, $h_z^z$, $h_z$ appearing in (61).

One can in principle push the integration of the equations further and find expressions for the $F_n^{(1)}$. However, the complexity of the answers grows very fast and one needs a better approach.

## 5.3 Operator formalism

In [23], based on previous insights of [57,90], an operator formalism was introduced to study the holomorphic anomaly equations in the one-modulus, local case. This formalism made it possible to find exact solutions for the multi-instantons. We will now construct this operator formalism for *arbitrary* Calabi–Yau manifolds.

The construction requires working in the big moduli space of the Calabi–Yau manifold, involving the $r$ complex deformation coordinates $z^a$, as well as an additional coordinate given by the string coupling constant. We recall that, as in section 2, the Greek index $\alpha$ takes the values 0 and $a = 1, \cdots, r$. We first define derivative operators acting on the functions of $z^a$, $g_s$ and the propagators, as follows:

$$\mathfrak{d}_\alpha = (\mathfrak{d}_0, \mathfrak{d}_a)\,, \tag{193}$$

where

$$\mathfrak{d}_0 = -g_s\frac{\partial}{\partial g_s}\,, \qquad \mathfrak{d}_a = \mathfrak{D}_a - K_a g_s\frac{\partial}{\partial g_s}\,. \tag{194}$$

Here, $\mathfrak{D}_a$ acts on a function of $z^a$, $S^{ab}$, $\tilde{S}^a$, $\tilde{S}$ and $K_a$ as

$$\mathfrak{D}_a f = \frac{\partial f}{\partial z^a} + \partial_a S^{de}\frac{\partial f}{\partial S^{de}} + \partial_a \tilde{S}^c\frac{\partial f}{\partial \tilde{S}^c} + \partial_a \tilde{S}\frac{\partial f}{\partial \tilde{S}} + \partial_a K_c\frac{\partial f}{\partial K_c}\,, \tag{195}$$

where the derivatives of the propagators are computed with the rules (61). As in [32,34], we will unify the propagators in a matrix $S^{\alpha\beta}$ with

$$S^{00} = 2S\,, \qquad S^{0a} = -S^a\,. \tag{196}$$

The operators (193) are formulated in the same spirit as the operator (114) introduced above: they act only on sections of $\mathcal{L}^0$, i.e. on functions. Sections of $\mathcal{L}^n$ are mapped to functions by

dividing by the appropriate power of $g_s$. Then[18]

$$g_s^n \mathfrak{d}_\alpha \left( \frac{f}{g_s^n} \right) = D_\alpha f, \quad f \in \Gamma(\mathcal{L}^n). \tag{197}$$

So for example, we have

$$\mathfrak{d}_0 \left( \frac{X^I}{g_s} \right) = \frac{X^I}{g_s}, \tag{198}$$

and we can write the matrix $\chi_\alpha^I$ in (33) as

$$\chi_\alpha^I = g_s \mathfrak{d}_\alpha \left( \frac{X^I}{g_s} \right), \qquad \alpha, I = 0, 1, \cdots, h^{2,1}(X). \tag{199}$$

As in [23, 57], we want to find an operator depending on the instanton action $\mathcal{A}$ and the propagators which, in the holomorphic limit, gives the derivative with regard to the coordinates $X^I$ in moduli space. Let us first define

$$\mathfrak{t}^\alpha = g_s^2 S^{\alpha\beta} \mathfrak{d}_\beta \left( \frac{\mathcal{A}}{g_s} \right). \tag{200}$$

The propagators appearing in this equation are the unshifted ones. We will shortly pass to shifted propagators as defined in (60). We have introduced the appropriate factors of $g_s$ everywhere: the propagators have charge $-2$, so that they should be multiplied by a factor $g_s^2$. In (200) we have made implicitly a choice of frame, through the propagators $S^{\alpha\beta}$. We now recall that there is a frame associated to $\mathcal{A}$ and defined by (176), which we will call the $\mathcal{A}$-frame. This frame is not necessarily unique, but as we will see, the final physical answer we are looking for, namely the holomorphic limit of the instanton amplitudes, will not depend on that choice of frame. We now define

$$\mathfrak{t}_\mathcal{A}^\alpha = g_s^2 S_\mathcal{A}^{\alpha\beta} \mathfrak{d}_\beta \left( \frac{\mathcal{A}}{g_s} \right), \tag{201}$$

where $S_\mathcal{A}^{\alpha\beta}$ is the holomorphic limit of the propagator in the $\mathcal{A}$-frame. We consider as well the holomorphic limit of the operator $\mathfrak{d}_a$ in the $\mathcal{A}$-frame,

$$\mathfrak{d}_a^\mathcal{A} = \frac{\partial}{\partial z^a} - \mathcal{K}_a^\mathcal{A} g_s \frac{\partial}{\partial g_s}, \tag{202}$$

where

$$\mathcal{K}_a^\mathcal{A} = -\frac{\partial_a X_\mathcal{A}^0}{X_\mathcal{A}^0}, \tag{203}$$

is the holomorphic limit of the connection $K_a$ in the $\mathcal{A}$-frame. We note that $\mathfrak{d}_0$ does not depend on the frame, and we will denote $\mathfrak{d}_0^\mathcal{A} = \mathfrak{d}_0$. With all these ingredients, we can already define the wished-for operator,

$$\mathsf{D} = \mathfrak{t}^\alpha \mathfrak{d}_\alpha - \mathfrak{t}_\mathcal{A}^\alpha \mathfrak{d}_\alpha^\mathcal{A}. \tag{204}$$

An explicit calculation shows that

$$\mathsf{D} = T^\alpha \mathfrak{d}_\alpha = T^j \mathfrak{d}_j + T^0 \mathfrak{d}_0, \tag{205}$$

---

[18]More generally, the operators can act on sections of $\text{Sym}^k(T^*\mathcal{M}_{cs})^{1,0}$, but in this case, the covariant derivative on the RHS of (197) is the one associated to the bundle $\mathcal{L}$, i.e. does not involve the Christoffel connection.

where

$$T^j = g_s \left\{ -\mathcal{A}\left( \tilde{S}^j - \tilde{\mathcal{S}}_{\mathcal{A}}^j \right) + \partial_m \mathcal{A}(S^{mj} - \mathcal{S}_{\mathcal{A}}^{mj}) \right\},$$
$$T^0 = g_s \left\{ 2\mathcal{A}(\tilde{S} - \tilde{\mathcal{S}}_{\mathcal{A}}) - \partial_m \mathcal{A}(\tilde{S}^m - \tilde{\mathcal{S}}_{\mathcal{A}}^m) \right\} - K_j T^j, \tag{206}$$

which we have expressed already in terms of the shifted propagators. We will sometimes decompose

$$\mathsf{D} = \mathsf{D}_0 + \mathsf{D}_1, \tag{207}$$

where

$$\mathsf{D}_0 = T^0 \mathfrak{d}_0, \qquad \mathsf{D}_1 = T^j \mathfrak{d}_j. \tag{208}$$

An important fact is that the only dependence of the total operator $\mathsf{D}$ on $K_i$ stems from the $K_i$ dependence of the operator $\mathfrak{D}_a$ defined in (195); this dependence is tucked away in the contribution from $\partial_a K_c$, see (61). If we introduce

$$\tilde{T}^0 = T^0 + K_j T^j = g_s \left\{ 2\mathcal{A}(S - \mathcal{S}_{\mathcal{A}}) - \partial_m \mathcal{A}(\tilde{S}^m - \tilde{\mathcal{S}}_{\mathcal{A}}^m) \right\}, \tag{209}$$

we can write

$$\mathsf{D} = T^j \mathfrak{D}_j + \tilde{T}^0 \mathfrak{d}_0, \tag{210}$$

such that the coefficients of the operators $\mathfrak{D}_j$ and $\mathfrak{d}_0$ are manifestly independent of $K_i$.

We finally note the following property: any quantity of the form $\mathsf{D}f$ vanishes in the $\mathcal{A}$-frame, since $T_{\mathcal{A}}^j = T_{\mathcal{A}}^0 = 0$.

As an illustration, let us consider the one-modulus case. We have,

$$T^1 = g_s \left( \mathcal{A}'\Delta - \mathcal{A}\Delta_1 \right), \qquad T^0 = g_s \left\{ 2\Delta_2 \mathcal{A} - \mathcal{A}'\Delta_1 - K_z \left( \mathcal{A}'\Delta - A\Delta_1 \right) \right\}, \tag{211}$$

and the operator $\mathsf{D}$ can be written as

$$\mathsf{D} = g_s \left( \mathcal{A}'\Delta - \mathcal{A}\Delta_1 \right) \mathfrak{D}_z - g_s \left( 2\Delta_2 \mathcal{A} - \mathcal{A}'\Delta_1 \right) g_s \frac{\partial}{\partial g_s}. \tag{212}$$

In the case of one-modulus, toric Calabi–Yau manifolds we recover the operator introduced in [23, 57].

In the following, a crucial role will be played by the relations

$$\mathfrak{d}_i T^j = -\Gamma_{ik}^j T^k - T^0 \delta_i^j,$$
$$\mathfrak{d}_i T^0 = 0. \tag{213}$$

These relations can be obtained from the following ingredients. We noted the holomorphic limits of the shifted propagators in (64) and (65). From these, one can easily deduce the following result, generalizing (189):

$$\partial_{ij}^2 \mathcal{A} = -\partial_l \mathcal{A} \left( C_{ijm} \mathcal{S}_{\mathcal{A}}^{ml} - q_{ij}^l \right) + \mathcal{A} \left( C_{ijm} \tilde{\mathcal{S}}_{\mathcal{A}}^m - q_{ij} \right). \tag{214}$$

By using (64), (65) and (214), (213) follow by direct calculation.

As a consequence of (213), we have (when acting on sections of $(T^*\mathcal{M}_{cs})^{0,0}$)

$$\mathsf{D}_1^2 = T^i T^j \hat{D}_i \hat{D}_j - T^0 \mathsf{D}_1. \tag{215}$$

It also follows that the operators $\mathsf{D}_0, \mathsf{D}_1$ do not commute:

$$[\mathsf{D}_0, \mathsf{D}_1] = -T^0 \mathsf{D}_1. \tag{216}$$

The operator D plays the role of the covariant derivative, but the holomorphic anomaly equations also involve derivatives with regard to the propagators. We will now introduce an appropriate operator for this. We define

$$\delta^S_{ij} = \frac{1}{g_s^2} \left( \frac{\partial}{\partial S^{ij}} - K_{(i} \frac{\partial}{\partial \tilde{S}^{j)}} + \frac{1}{2} K_i K_j \frac{\partial}{\partial \tilde{S}} \right),$$

$$\delta^S_{0i} = -\frac{1}{g_s^2} \left( \frac{\partial}{\partial \tilde{S}^i} - K_i \frac{\partial}{\partial \tilde{S}} \right), \tag{217}$$

$$\delta^S_{00} = \frac{1}{2g_s^2} \frac{\partial}{\partial \tilde{S}}.$$

Let us now define the operator

$$\omega_S = T^i T^j \delta^S_{ij} + T^0 T^i \delta^S_{0i} + T_0^2 \delta^S_{00}. \tag{218}$$

Then, by using (216), we can rewrite the equations (111), (117) in the form

$$\omega_S \widehat{F}^{(0)} = \frac{1}{2} \left\{ D^2 \widetilde{F}^{(0)} + \left( D \widetilde{F}^{(0)} \right)^2 \right\}. \tag{219}$$

In this equation, $F_1$ requires special treatment: we have to set

$$D F_1 = T^j C_j - \tilde{T}^0 \left( \frac{\chi}{24} - 1 \right) = T^j D_j F_1 - T^0 \left( \frac{\chi}{24} - 1 \right), \tag{220}$$

where $C_j$ was defined in (104). We note that $D F_1$ does not depend on $K_j$.

The goal of this formalism is to find closed-form solutions for the instanton amplitudes. Let us first consider the one-instanton amplitude, which satisfies the linearized equation (169). In the operator language that we have just introduced, this equation reads simply

$$\omega_S F^{(1)} - D \widetilde{F}^{(0)} D F^{(1)} = \frac{1}{2} D^2 F^{(1)}. \tag{221}$$

This suggests defining the operator

$$W = \omega_S - D \widetilde{F}^{(0)} D, \tag{222}$$

so that (221) becomes

$$W F^{(1)} = \frac{1}{2} D^2 F^{(1)}. \tag{223}$$

As in [23, 57], the basic operators of our algebra will be D and W, and we need to calculate their commutator. The building blocks are the following commutators:

$$\left[ \delta^S_{ij}, \mathfrak{d}_k \right] = -\Gamma^m_{ik} \delta^S_{jm} - \Gamma^m_{jk} \delta^S_{im},$$

$$\left[ \delta^S_{0i}, \mathfrak{d}_j \right] = -2\delta^S_{ij} - \Gamma^m_{ij} \delta^S_{0m} + \frac{1}{g_s^2} C_{ijm} \frac{\partial}{\partial K_m}, \tag{224}$$

$$\left[ \delta^S_{00}, \mathfrak{d}_i \right] = -\delta^S_{0i},$$

which leads to the simple result

$$\left[ \omega_S, \mathfrak{d}_k \right] = \frac{1}{g_s^2} T^0 T^i C_{ikm} \frac{\partial}{\partial K_m}. \tag{225}$$

With some additional work, one finds

$$[W, D] = D \mathcal{G} D + K, \tag{226}$$

where

$$G = \frac{\mathcal{A}}{g_s} + \mathsf{D}\widetilde{F}^{(0)} = \frac{\mathcal{A}}{g_s} + \sum_{g \geq 1} \mathsf{D}\left(g_s^{2g-2}F_g\right), \tag{227}$$

and

$$\mathsf{K} = f_m \frac{\partial}{\partial K_m}, \qquad f_m = \frac{1}{g_s^2} T^0 T^i T^j C_{ijm}. \tag{228}$$

With this, the operator formalism is set up. Before exploiting it to obtain exact results for the instanton amplitudes, let us consider the holomorphic limit of the operators.

## 5.4 Holomorphic limit

Since $\mathcal{A}$ is a linear combination of Calabi–Yau periods, we can write (recall the notational conventions fixed at the beginning of this section)

$$\mathcal{A} = c^J P_J + d_J X^J, \tag{229}$$

and

$$\mathfrak{d}_\alpha\left(\frac{\mathcal{A}}{g_s}\right) = \frac{1}{g_s} \chi_\alpha^I (c^J \tau_{JI} + d_I), \tag{230}$$

where one uses the property $X^I \tau_{IJ} = P_J$ noted in (54). As we have explained, in the $\mathcal{A}$-frame $\mathcal{A} = \aleph X_{\mathcal{A}}^1$, so that the coefficients appearing in (229) are entries of the symplectic transformation (52):

$$c^J = \aleph \, \mathfrak{C}^{1J}, \qquad d_J = \aleph \, \mathfrak{D}^1_J, \tag{231}$$

so that we can write

$$\mathfrak{d}_\alpha\left(\frac{\mathcal{A}}{g_s}\right) = \frac{\aleph}{g_s} (\mathfrak{C}\tau + \mathfrak{D})^1_I \chi_\alpha^I. \tag{232}$$

We will now show that, in the holomorphic limit, $\mathsf{D}$ becomes a derivative operator in the big moduli space with regard to the periods $X^I$. Let us denote by $\widetilde{T}^0_h$, $T^j_h$ the holomorphic limit of the quantities introduced in (206), (209). By using (73), we find

$$\begin{aligned} T^j_h &= g_s \left[ (\mathfrak{C}\tau + \mathfrak{D})^{-1}\mathfrak{C} \right]^{IJ} \left( \partial_m \mathcal{A}\chi_I^m + \mathcal{A}h_I \right) \chi_J^j, \\ \widetilde{T}^0_h &= g_s \left[ (\mathfrak{C}\tau + \mathfrak{D})^{-1}\mathfrak{C} \right]^{IJ} \left( \partial_m \mathcal{A}\chi_I^m + \mathcal{A}h_I \right) h_J. \end{aligned} \tag{233}$$

From the explicit expression for $h_I$ in (66), as well as (232), we can simplify

$$T^j_h = g_s c^J \chi_J^j, \qquad \widetilde{T}^0_h = g_s c^J h_J. \tag{234}$$

Let $f$ be a homogeneous function of degree $n$ in the $X$ coordinates, i.e. $f \in \mathcal{L}^n$. As explained above, $\mathsf{D}$ is defined to act on the image $g_s^{-n}f$ of such a function in $\mathcal{L}^0$. By Euler's theorem, we have

$$nf = X^I \frac{\partial f}{\partial X^I}. \tag{235}$$

Therefore, in the holomorphic limit

$$\mathsf{D}(g_s^{-n}f) \to g_s^{-n} \left\{ T^j_h \frac{\partial X^I}{\partial z^j} + \widetilde{T}^0_h X^I \right\} \frac{\partial f}{\partial X^I}, \tag{236}$$

which simplifies to

$$\mathsf{D}(g_s^{-n}f) \to g_s^{-n+1} c^I \frac{\partial f}{\partial X^I}. \tag{237}$$

Note that this expression only depends on the coefficients $c^I$.

The holomorphic limit of (220) is slightly different. We have

$$T_h^j \partial_j F_1 = g_s c^J \frac{\partial F_1}{\partial X^J}, \tag{238}$$

where we used that $F_1$ is homogeneous of degree zero. Furthermore,

$$T_h^0 = g_s c^J \chi_{J,h}^0, \tag{239}$$

where $\chi_{J,h}^0$ denotes the holomorphic limit of $\chi_J^0$. An explicit computation shows that

$$\chi_{J,h}^0 = \frac{1}{X^0} \delta_{J0}. \tag{240}$$

We conclude that, in the holomorphic limit,

$$\mathrm{D}F_1 \to g_s c^J \frac{\partial \mathcal{F}_1}{\partial X^J} - g_s \frac{c^0}{X^0} \left( \frac{\chi}{24} - 1 \right). \tag{241}$$

## 5.5 One-instanton amplitude

We are now ready to present an explicit, exact result for the one-instanton amplitude, generalizing the local, one-modulus case studied in [23]. As in [23], let us first consider the ansatz

$$E = \exp(\Sigma). \tag{242}$$

We want to solve for $\Sigma$ so that $E$ satisfies the one-instanton equation (223). It is easy to see that this is equivalent to the following equation for $\Sigma$:

$$\mathrm{W}\Sigma = \frac{1}{2} \left( \mathrm{D}^2 \Sigma + (\mathrm{D}\Sigma)^2 \right). \tag{243}$$

We will now prove that this is solved by

$$\Sigma = \sum_{k \geq 1} \frac{(-1)^k}{k!} \mathrm{D}^{k-1} G. \tag{244}$$

The proof is very similar to what was done in [23] in the local case, although some steps are more involved. We first note that, in terms of the operators $\mathrm{D}$ and $\mathrm{W}$ introduced above, the holomorphic anomaly equations for the perturbative series can be written as

$$\mathrm{W}\widehat{F}^{(0)} = \frac{1}{2} \mathrm{D}^2 \widetilde{F}^{(0)} - \frac{1}{2} \left( \mathrm{D}\widetilde{F}^{(0)} \right)^2 + \mathrm{D}F_1^{(0)} \mathrm{D}\widetilde{F}^{(0)}. \tag{245}$$

As in [23], the first step is to prove

$$\mathrm{W}G = \frac{1}{2} \mathrm{D}^2 G. \tag{246}$$

This is done by direct calculation. We have

$$\mathrm{W}G = \mathrm{W}\left( \frac{\mathcal{A}}{g_s} \right) + \mathrm{W}\mathrm{D}F_1^{(0)} + \mathrm{W}\mathrm{D}\widehat{F}^{(0)}. \tag{247}$$

We can now use the commutation relation (226) and (245) to write

$$\mathrm{W}G = \mathrm{W}\left( \frac{\mathcal{A}}{g_s} \right) + \mathrm{W}\mathrm{D}F_1^{(0)} - \mathrm{D}G\mathrm{D}F_1^{(0)} + \mathrm{D}\widetilde{F}^{(0)} \mathrm{D}\left( \frac{\mathcal{A}}{g_s} \right) + \frac{1}{2} \mathrm{D}^3 \widetilde{F}^{(0)} + \mathrm{D}\left( \mathrm{D}F_1^{(0)} \mathrm{D}\widetilde{F}^{(0)} \right), \tag{248}$$

where we have used that $\widehat{F}^{(0)}$ is independent of $K_m$, as follows from (116). On the other hand,

$$\frac{1}{2}\mathsf{D}^2 G = \frac{1}{2}\left(\mathsf{D}^2\left(\frac{\mathcal{A}}{g_s}\right) + \mathsf{D}^3\widetilde{F}^{(0)}\right). \tag{249}$$

By using the definition of $G$ on the RHS of (248), we conclude that

$$\mathsf{W}G - \frac{1}{2}\mathsf{D}^2 G = \omega_S\left(\mathsf{D}F_1^{(0)}\right) - \mathsf{D}\left(\frac{\mathcal{A}}{g_s}\right)\mathsf{D}F_1^{(0)} - \frac{1}{2}\mathsf{D}^2\left(\frac{\mathcal{A}}{g_s}\right). \tag{250}$$

The RHS can be seen to vanish by a direct calculation, starting from (220).

We are now ready to prove (243). We note that $\Sigma$ can be written as

$$\Sigma = \mathsf{O}G, \tag{251}$$

where the operator $\mathsf{O}$ is

$$\mathsf{O} = \sum_{k\geq 1}\frac{(-1)^k}{k!}\mathsf{D}^{k-1} = \frac{1}{\mathsf{D}}(e^{-\mathsf{D}} - 1) = -\int_0^1 du\, e^{-u\mathsf{D}}. \tag{252}$$

Let us define the iterated commutator $[A,B]_n$ as

$$[A,B]_{n\geq 1} = [A,[A,B]_{n-1}], \qquad [A,B]_0 = B. \tag{253}$$

Then, Hadamard's lemma says that

$$e^A B e^{-A} = \sum_{n=0}^{\infty}\frac{1}{n!}[A,B]_n. \tag{254}$$

Let us compute $[\mathsf{D},\mathsf{K}]_n$, where $\mathsf{K}$ is the operator introduced in (226). We have

$$\left[\mathsf{D}, f_m\frac{\partial}{\partial K_m}\right] = \left[T^j\mathfrak{d}_j + T^0\mathfrak{d}_0, f_m\frac{\partial}{\partial K_m}\right] = \left(\mathsf{D}(f_m) - T^j\Gamma_{jm}^r f_r\right)\frac{\partial}{\partial K_m}. \tag{255}$$

This suggests defining the following transformation acting on a vector $v_m$:

$$\mathcal{I}(v_m) = \mathsf{D}(v_m) - T^j\Gamma_{jm}^r v_r, \tag{256}$$

such that

$$[\mathsf{D},\mathsf{K}]_n = f_m^{(n)}\frac{\partial}{\partial K_m}, \qquad n \geq 0, \tag{257}$$

where $f_m^{(0)} = f_m$ and

$$f_m^{(n)} = \mathcal{I}^n(f_m), \qquad n \geq 1. \tag{258}$$

From here, we deduce that

$$[\mathsf{D},\mathsf{W}]_0 = \mathsf{W}, \qquad [\mathsf{D},\mathsf{W}]_{n\geq 1} = -\mathsf{D}^n G\mathsf{D} - f_m^{(n-1)}\frac{\partial}{\partial K_m}. \tag{259}$$

We now note that $G$ is independent of $K_m$. This follows from the fact that $\mathsf{D}F_1$ does not depend on $K_m$, as noted after (220), and from the commutation relation (255), together with (116). We conclude that

$$e^{u\mathsf{D}}\mathsf{W}e^{-u\mathsf{D}}G = \mathsf{W}G - \sum_{n\geq 1}\frac{u^n}{n!}\mathsf{D}^n G\mathsf{D}G. \tag{260}$$

This result allows us to express $W\Sigma$ as

$$W\Sigma = WOG = -\int_0^1 du\, W e^{-uD} G = -\int_0^1 du\left(e^{-uD}WG - e^{-uD}\sum_{n\geq 1}\frac{u^n}{n!}D^n GDG\right). \quad (261)$$

To evaluate the RHS, we use the crucial identity (246) for $G$ in the first term to obtain $\frac{1}{2}D^2\Sigma$. In the second term, we use the identity

$$e^D(fg) = \left(e^D f\right)\left(e^D g\right), \quad (262)$$

to write

$$\int_0^1 du\, e^{-uD}((e^{uD}-1)G)DG = \int_0^1 du\,[(1-e^{-uD})G][e^{-uD}DG]. \quad (263)$$

Combining these results yields

$$W\Sigma = \frac{1}{2}D^2\Sigma + \int_0^1 du\,[(1-e^{-uD})G][e^{-uD}DG]. \quad (264)$$

It remains to prove that the last term is equal to $\frac{1}{2}(D\Sigma)^2$. This can be shown by a direct calculation:

$$\frac{1}{2}(D\Sigma)^2 = \frac{1}{2}\int_0^1 du\int_0^1 dv\,(e^{-uD}DG)(e^{-vD}DG) \quad (265)$$

$$= \int_0^1 du\int_0^u dv\,(e^{-uD}DG)(e^{-vD}DG)$$

$$= \int_0^1 du\,(e^{-uD}DG)(1-e^{-uD}G).$$

Although (242) solves the equation (223) for the one-instanton amplitude, it does not satisfy the boundary condition (179). Indeed, we have $G_{\mathcal{A}} = \mathcal{A}/g_s$, therefore

$$E_{\mathcal{A}} = e^{-\mathcal{A}/g_s}, \quad (266)$$

and it misses the prefactor in (179). We then write an ansatz of the form

$$F^{(1)} = \frac{1}{2\pi}\mathfrak{a}\exp(\Sigma). \quad (267)$$

Proving that the ansatz (267) solves (223) is equivalent to showing that

$$W\mathfrak{a} = \frac{1}{2}D^2\mathfrak{a} + D\Sigma D\mathfrak{a}. \quad (268)$$

We now show that

$$\mathfrak{a} = 1 + G + D\Sigma, \quad (269)$$

provides a solution to (268) and implements the correct boundary condition, since

$$\mathfrak{a}_{\mathcal{A}} = \frac{\mathcal{A}}{g_s} + 1. \quad (270)$$

To show that (269) solves (268), we compute

$$W\mathfrak{a} = WG + WD\Sigma = \frac{1}{2}D^2 G + DGD\Sigma + f_m\frac{\partial}{\partial K_m}\Sigma + \frac{1}{2}D^3\Sigma + D\Sigma D^2\Sigma, \quad (271)$$

where we have again used the identity (246) and the commutation relation (226), as well as the relation (243). It is easy to see that $\Sigma$ does not depend on $K_m$, by repeatedly using (255), so the third term on the RHS vanishes. By

$$\mathrm{D}\mathfrak{a} = \mathrm{D}G + \mathrm{D}^2\Sigma, \quad \mathrm{D}^2\mathfrak{a} = \mathrm{D}^2 G + \mathrm{D}^3\Sigma, \tag{272}$$

(268) follows immediately from (271) and (272). This concludes the demonstration that the ansatz (267) solves (223).

Let us now determine the holomorphic limit of the one-instanton amplitude. Since the holomorphic limit of D is the derivative operator (237), it follows from (244) and (227) that the holomorphic limit of $\Sigma$ is

$$\mathcal{F}\left(X^I - g_s c^I\right) - \mathcal{F}\left(X^I\right), \tag{273}$$

where the coefficients $c^I$ were defined in (229). Here,

$$\mathcal{F} = \frac{1}{g_s^2}\tilde{\mathcal{F}}_0 + \tilde{\mathcal{F}}_1 + \sum_{g \geq 2} g_s^{2g-2}\mathcal{F}_g, \tag{274}$$

with the tildes indicating that the genus zero and genus one free energies appearing in the total free energy of (273) are special. $\tilde{\mathcal{F}}_0$ is defined by the equation

$$c^J \frac{\partial \tilde{\mathcal{F}}_0}{\partial X^J} = c^J P_J + d_J X^J = \mathcal{A}, \tag{275}$$

so it might differ from the usual $\mathcal{F}_0$ in quadratic terms in the $X^J$. By (241), the genus one free energy appearing in (273) is

$$\tilde{\mathcal{F}}_1 = \mathcal{F}_1 - \left(\frac{\chi}{24} - 1\right)\log X^0. \tag{276}$$

Similarly, the holomorphic limit of $\mathfrak{a}$ is

$$1 + g_s c^J \frac{\partial \mathcal{F}}{\partial X^J}\left(X^I - g_s c^I\right). \tag{277}$$

We conclude that the holomorphic limit of $F^{(1)}$ is[19]

$$\mathcal{F}^{(1)} = \frac{1}{2\pi}\left(1 + g_s c^J \frac{\partial \mathcal{F}}{\partial X^J}\left(X^I - g_s c^I\right)\right)\exp\left[\mathcal{F}\left(X^I - g_s c^I\right) - \mathcal{F}\left(X^I\right)\right]. \tag{278}$$

The exponential of (273) gives

$$e^{-\mathcal{A}/g_s}\exp\left[\frac{c^I c^J}{2}\tau_{IJ}\right]\left(1 + g_s \Upsilon_1 + g_s^2\left(\Upsilon_2 + \frac{1}{2}\Upsilon_1^2\right) + \cdots\right), \tag{279}$$

where

$$\begin{aligned}
\Upsilon_1 &= -\frac{1}{3!}c^I c^J c^K C_{IJK} - c^I \frac{\partial \mathcal{F}_1}{\partial X^I} + \frac{c^0}{X^0}\left(\frac{\chi}{24} - 1\right), \\
\Upsilon_2 &= \frac{1}{4!}c^I c^J c^K c^L \frac{\partial^4 \mathcal{F}_0}{\partial X^I \partial X^J \partial X^K \partial X^L} + \frac{c^I c^J}{2}\frac{\partial^2 \mathcal{F}_1}{\partial X^I \partial X^J} + \frac{\left(c^0\right)^2}{2\left(X^0\right)^2}\left(\frac{\chi}{24} - 1\right).
\end{aligned} \tag{280}$$

We have used the standard notation for the derivatives of the prepotential, as in (54),

$$\tau_{IJ} = \frac{\partial^2 \tilde{\mathcal{F}}_0}{\partial X^I \partial X^J}, \qquad C_{IJK} = \frac{\partial^3 \mathcal{F}_0}{\partial X^I \partial X^J \partial X^K}. \tag{281}$$

---

[19]It has been pointed out by S. Shatashvili that this expression is reminiscent of the string field theory of [91,92].

The prefactor gives

$$\frac{1}{2\pi}\left(\frac{\mathcal{A}}{g_s}+1-c^I c^J \tau_{IJ}+g_s\left(\frac{1}{2}c^I c^J c^K C_{IJK}+c^I\frac{\partial \mathcal{F}_1}{\partial X^I}-\frac{c^0}{X^0}\left(\frac{\chi}{24}-1\right)\right)+\mathcal{O}(g_s^2)\right). \tag{282}$$

If we now write, as in (168),

$$\mathcal{F}^{(1)}=\frac{1}{g_s}e^{-\mathcal{A}/g_s}\sum_{n\geq 0}\mathcal{F}_n^{(1)}g_s^n, \tag{283}$$

we obtain

$$
\begin{aligned}
\mathcal{F}_0^{(1)} &= \frac{1}{2\pi}\mathcal{A}\exp\left[\frac{c^I c^J}{2}\tau_{IJ}\right], \\
\mathcal{F}_1^{(1)} &= \frac{1}{2\pi}\left(1-c^I c^J \tau_{IJ}+\mathcal{A}\Upsilon_1\right)\exp\left[\frac{c^I c^J}{2}\tau_{IJ}\right], \\
\mathcal{F}_2^{(1)} &= \frac{1}{2\pi}\left\{\mathcal{A}\left(\Upsilon_2+\frac{1}{2}\Upsilon_1^2\right)+\Upsilon_1\left(1-c^I c^J \tau_{IJ}\right)+\frac{1}{2}c^I c^J c^K C_{IJK}\right. \\
&\qquad\left.+c^I\frac{\partial \mathcal{F}_1}{\partial X^I}-\frac{c^0}{X^0}\left(\frac{\chi}{24}-1\right)\right\}\exp\left[\frac{c^I c^J}{2}\tau_{IJ}\right].
\end{aligned}
\tag{284}
$$

## 5.6 Multi-instanton amplitudes

We will now solve for the multi-instanton amplitudes. The derivation will closely follow the local case in [23].

Recall that the holomorphic anomaly equation (223) for the one-instanton amplitude $F^{(1)}$ is linear in $F^{(1)}$. We can arrive at very similar equations in the multi-instanton case, as demonstrated in [23], by considering the holomorphic anomaly equations for the partition function, rather than for the free energy [22, 56].

Following [23], we define the reduced partition function $Z_r$ by

$$Z_r = Z/Z^{(0)}=e^{F_r}, \tag{285}$$

where

$$Z=e^F, \qquad Z^{(0)}=e^{F^{(0)}}, \tag{286}$$

are respectively the full and the perturbative partition function. The correction terms to the holomorphic anomaly equations involving $F_0^{(0)}$ and $F_0^{(1)}$ (requiring the introduction of $\tilde{F}^{(0)}$ and $\hat{F}^{(0)}$ above) largely cancel when considering the quotient (285); it satisfies the linear equation

$$\mathsf{W}Z_r = \frac{1}{2}\mathsf{D}^2 Z_r. \tag{287}$$

We now look for trans-series solutions to (287) in the multi-instanton sectors with the primitive instanton action $\mathcal{A}$. We first comment that, as pointed out in [20,21,23], there could be both instanton solutions of magnitude $e^{-\mathcal{A}/g_s}$ and anti-instanton solutions of magnitude $e^{\mathcal{A}/g_s}$. So a generic trans-series solution should include both instanton and anti-instanton sectors, as well as mixed sectors. We therefore make the ansatz

$$Z_r = 1+\sum_{\substack{n,m\geq 0\\(n,m)\neq 0}}C_+^n C_-^m Z^{(n|m)}, \tag{288}$$

for the reduced partition function, where the components are such that in the small $g_s$ limit, they behave as

$$Z^{(n|m)}\sim \exp\left(-\frac{n-m}{g_s}\mathcal{A}\right). \tag{289}$$

When $m = 0$, i.e. in the absence of anti-instantons, we will often drop it from our notation. The non-perturbative free energies in the multi-instanton sectors can easily be obtained by considering a similar decomposition of the reduced free energy

$$F_r = \sum_{\substack{n,m \geq 0 \\ (n,m) \neq 0}} C_+^n C_-^m F^{(n|m)} \,, \tag{290}$$

taking the logarithm of both sides of (285) and then comparing coefficients of $C_\pm$. For example

$$F^{(2)} = Z^{(2)} - \frac{1}{2}\left(Z^{(1)}\right)^2 \,, \tag{291a}$$

$$F^{(1|1)} = Z^{(1|1)} - Z^{(1|0)}Z^{(0|1)} \,. \tag{291b}$$

As the equation for the reduced partition function $Z_r$ is linear, it is easy to write down the equations satisfied by its components:

$$\mathsf{W} Z^{(n|m)} = \frac{1}{2}\mathsf{D}^2 Z^{(n|m)} \,. \tag{292}$$

To solve this equation, we also need to specify boundary conditions. As in the one-instanton sector, we will impose these in the $\mathcal{A}$-frame, with $\mathcal{A}$ proportional to an A-period. To begin with, we will impose the rather generic boundary condition

$$\mathcal{Z}_{\mathcal{A}}^{(n|m)} = \left(\sum_k a_k (\mathcal{A}/g_s)^k\right) e^{-(n-m)\mathcal{A}/g_s} \,, \tag{293}$$

in such a frame, where on the LHS, the subscript $\mathcal{A}$ means that the partition function is evaluated in the $\mathcal{A}$-frame. The actual boundary conditions of interest will be a specialization of this class. Note that unlike the local cases discussed in [23], it is appropriate to always accompany $\mathcal{A}$ with a factor of $1/g_s$, such that the instanton partition function, like $Z^{(0)}$, is a section of $\mathcal{L}^0$.

As the equation satisfied by $Z^{(n|m)}$ and by $F^{(1)}$, (292) and (221), coincide, we can make the same ansatz

$$Z^{(n|m)} = \mathfrak{a}^{(n|m)} \exp \Sigma^{(n|m)} \,, \tag{294}$$

as before to solve it. In particular, we will search for solutions such that the exponent satisfies the equation

$$\mathsf{W}\Sigma^{(n|m)} = \frac{1}{2}\left(\mathsf{D}^2\Sigma^{(n|m)} + (\mathsf{D}\Sigma^{(n|m)})^2\right) \,, \tag{295}$$

while the prefactor satisfies

$$\mathsf{W}\mathfrak{a}^{(n|m)} = \frac{1}{2}\mathsf{D}^2\mathfrak{a}^{(n|m)} + \mathsf{D}\Sigma^{(n|m)}\mathsf{D}\mathfrak{a}^{(n|m)} \,. \tag{296}$$

In fact, equation (295) can be solved by setting

$$\Sigma^{(n|m)} = \Sigma^{(n-m)} = \mathsf{O}^{(n-m)}G \,, \tag{297}$$

where

$$\mathsf{O}^{(\ell)} = \sum_{k \geq 1} \frac{(-1)^k}{k!} \ell^k \mathsf{D}^{k-1} = \frac{1}{\mathsf{D}}(e^{-\ell \mathsf{D}} - 1) = -\int_0^\ell e^{-u\mathsf{D}} \, du \,. \tag{298}$$

The proof follows by repeating the argument from the previous section, and noting that it did not depend on the value of the upper bound on the integration in (298). Changing this value from 1 to $\ell$ is required to obtain the correct small $g_s$ limit (289)

By introducing the operator $\mathsf{M}^{(n|m)}$

$$\mathsf{M}^{(n|m)} = \mathsf{W} - \mathsf{D}\Sigma^{(n|m)}\mathsf{D} - \frac{1}{2}\mathsf{D}^2 \,, \tag{299}$$

equation (296) becomes the condition that $\mathfrak{a}^{(n|m)}$ is annihilated by $\mathsf{M}^{(n|m)}$

$$\mathsf{M}^{(n|m)}\mathfrak{a}^{(n|m)} = 0 \,. \tag{300}$$

Since the equation (292) is linear, we can without loss of generality consider the simpler boundary condition

$$\mathfrak{a}_{\mathcal{A}}^{(n|m)} = (\mathcal{A}/g_s)^k \,. \tag{301}$$

Our problem is to find $\mathfrak{a}^{(n|m)}$ which satisfies both equations (300) and (301). We first notice that the following object

$$X^{(n|m)} = G + \mathsf{D}\Sigma^{(n|m)} \,, \tag{302}$$

satisfies the equation (300), as follows from the computation (272) in section 5.5. But it only satisfies the simplest boundary condition

$$X_{\mathcal{A}}^{(n|m)} = \mathcal{A}/g_s \,. \tag{303}$$

We claim that equation (300) with the generic boundary condition (301) is solved by

$$\mathfrak{m}_k(X) = \sum_{\boldsymbol{k},d(\boldsymbol{k})=k} C_{\boldsymbol{k}} \mathfrak{X}_{\boldsymbol{k}} \,, \tag{304}$$

with coefficients

$$C_{\boldsymbol{k}} = \frac{k!}{\prod_{j\geq 1} k_j!(j!)^{k_j}} \,, \tag{305}$$

and generators $\mathfrak{X}_{\boldsymbol{k}}$ given by words made out of the letters $X, \mathsf{D}X, \mathsf{D}^2X, \ldots$, $X$ being short for $X^{(n|m)}$,

$$\mathfrak{X}_{\boldsymbol{k}} = X^{k_1}(\mathsf{D}X)^{k_2}(\mathsf{D}^2X)^{k_3}\cdots \,. \tag{306}$$

Each word is labelled by a partition $\boldsymbol{k} = (k_1, k_2, \ldots)$ of the integer $k$ so that

$$d(\boldsymbol{k}) = \sum_j jk_j = k \,. \tag{307}$$

The same proof as in [23] goes through here, and we will not repeat it. Some examples of $\mathfrak{m}_k$ are

$$\mathfrak{m}_2 = X^2 + \mathsf{D}X \,, \tag{308a}$$
$$\mathfrak{m}_3 = X^3 + 3X\mathsf{D}X + \mathsf{D}^2X \,. \tag{308b}$$

We note that the generator $X^{(n|m)} = X^{(n-m)}$ in fact also depends only on the difference $n - m$, as this is the case for $\Sigma^{(n|m)}$ from which it derives its $n, m$ dependence.

To summarize, (292) is solved by the ansatz

$$Z^{(n|m)} = \mathfrak{a}^{(n|m)} \exp \Sigma^{(n|m)} \,, \quad \Sigma^{(n|m)} = \mathsf{O}^{(n-m)}G \,, \tag{309}$$

with $G$ given in (227) above, $\mathsf{O}^{(\ell)}$ defined in (298), and $\mathfrak{a}^{(n|m)}$ given by (304) for the boundary condition (301), from which the general boundary condition (293) is obtained by superposition.

Let us make some comments regarding this exact solution. Note that with (237), one can easily evaluate the multi-instanton partition functions in the holomorphic limit. For instance, the letter $D^{k-1}X$ for the prefactor evaluates to

$$D^{k-1}X \to g_s^k c^{J_1} \dots c^{J_k} \partial_{X^{J_1}} \dots \partial_{X^{J_k}} \mathcal{F}^{(0)}(X^I - (n-m)g_s c^I), \tag{310}$$

while the exponent $\Sigma^{(n-m)}$ evaluates to

$$\Sigma^{(n-m)} \to \mathcal{F}^{(0)}(X^I - (n-m)g_s c^I) - \mathcal{F}^{(0)}(X^I). \tag{311}$$

As emphasized in [23], this form of the exponent is very similar to instanton amplitudes in matrix models, obtained by eigenvalue tunneling (see e.g. [93]). It suggests that the flat coordinates $X^I$ are quantized in units of the string coupling constant. Such a picture is somewhat expected in the local Calabi–Yau case considered in [23], but it is certainly more surprising in the case of compact Calabi–Yau manifolds.

Up to now, we have considered the generic boundary condition (293) for the partition function. The numerical studies in this paper show that similar to the case of local Calabi–Yau manifolds [20, 21, 23], there is a special family of boundary conditions for the free energies which is relevant for the resurgent structure of the topological string, given by

$$\mathcal{F}_\mathcal{A}^{(k|0)} = \frac{\tau_k}{k^2}\left(1 + \frac{k\mathcal{A}}{g_s}\right)e^{-k\mathcal{A}/g_s}, \qquad \mathcal{F}_\mathcal{A}^{(0|k)} = \frac{\tau_k}{k^2}\left(1 - \frac{k\mathcal{A}}{g_s}\right)e^{+k\mathcal{A}/g_s}. \tag{312}$$

The subscript $\mathcal{A}$ is to indicate a special choice of frame as in section 5.2. Note that due to the symmetry $\mathcal{F}^{(0)}(-g_s) = \mathcal{F}^{(0)}(g_s)$ of the perturbative free energy, the anti-instanton boundary condition (on the right in (312)) can be obtained from the instanton boundary condition (on the left in (312)) by the map $g_s \to -g_s$. We will further specialize the boundary conditions (312) by restricting the constants $\tau_k$ to be of the form

$$\tau_k^\ell = \frac{\delta_{k\ell}}{2\pi}, \tag{313}$$

yielding a subfamily of boundary conditions labelled by a positive integer $\ell$. The partition functions as well as free energies solved with this particular boundary condition will be denoted by $Z_\ell^{(n|m)}$, $F_\ell^{(n|m)}$, respectively. In this notation, the multi-instanton contributions in the MUM and conifold frame we shall identify in section 5.7 are of the form $\mathcal{F}_\ell^{(\ell)}$. The specialized boundary conditions on the free energy translate to the following boundary condition on the partition function:

$$\mathcal{Z}_{\ell,\mathcal{A}}^{(a|b)} = \begin{cases} \frac{1}{(2\pi)^{n+m} n! m! \ell^{2n+2m}}\left(1 + \frac{\ell\mathcal{A}}{g_s}\right)^n \left(1 - \frac{\ell\mathcal{A}}{g_s}\right)^m e^{-\frac{(n-m)\ell\mathcal{A}}{g_s}}, & \text{if } a = n\ell, \ b = m\ell, \\ 0, & \text{else.} \end{cases} \tag{314}$$

Note that a vanishing boundary condition $\mathcal{Z}_{\ell,\mathcal{A}}^{(a|b)} = 0$ implies that the associated non-holomorphic partition function $Z_\ell^{(a|b)}$ vanishes. The same does not hold for the free energies, as e.g.

$$F_1^{(2)} = Z_1^{(2)} - \frac{1}{2}\left(Z_1^{(1)}\right)^2, \tag{315}$$

and the RHS does not vanish. Note further that the relation between $F_\ell^{(\ell)}$ and $Z_\ell^{(\ell)}$ is particularly simple. In fact,

$$F_\ell^{(\ell)} = Z_\ell^{(\ell)}, \tag{316}$$

as $F^{(k)}$ is a polynomial in $Z^{(j)}$ for $j \leq k$, but $Z_\ell^{(j)} = 0$ for $j < \ell$. More general boundary conditions than $\ell$ for $F^{(\ell)}$ are needed to discuss the general resurgent structure of the topological string, as we briefly touch upon in the introduction to section 6.

When applying the results (294), (297), (304), (227) to find the multi-instanton partition function or free energies, we can use the trick that while a monomial boundary condition $(\mathcal{A}/g_s)^k$ leads to the prefactor $\mathfrak{m}_k(X)$ of the exact solution, a monomial boundary condition $(1 \pm \ell \mathcal{A}/g_s)^k$ leads to the prefactor $\mathfrak{m}_k(1 \pm \ell X)$.

Let us now give a concrete examples. In the family of solutions with boundary condition $\ell = 1$, when $n = 2, m = 0$, our construction gives

$$Z_1^{(2)} = \frac{1}{(2\pi)^2} \frac{1}{2} \left( (1 + X^{(2)})^2 + \mathrm{D}X^{(2)} \right) \exp \Sigma^{(2)}, \tag{317}$$

where we have explicitly written down the superscripts for both $X$ and $\Sigma$. In the holomorphic limit, this becomes

$$\mathcal{Z}_1^{(2)} = \frac{1}{(2\pi)^2} \frac{1}{2} \left( (1 + g_s c_J \partial_{X_J} \mathcal{F}^{(0)}(X_I - 2g_s c_I))^2 + g_s^2 c_J c_K \partial_{X_J} \partial_{X_K} \mathcal{F}^{(0)}(X_I - 2g_s c_I) \right)$$
$$\times \exp(\mathcal{F}^{(0)}(X_I - 2g_s c_I) - \mathcal{F}^{(0)}(X_I)). \tag{318}$$

## 5.7 Boundary conditions for the non-perturbative holomorphic anomaly

As we saw above, determining a trans-series solution to the holomorphic anomaly equations requires as external input two pieces of information: an action $\mathcal{A}$, and appropriate boundary conditions at each loop order to fix the holomorphic ambiguities.

We make the following conjectures about this input, in analogy to the case of local Calabi–Yau manifolds: [20, 21, 23]:

- The action $\mathcal{A}$ is a period over an integral cycle (see footnote 17 regarding the question of normalization).

- The relevant boundary condition is the following: The action $\mathcal{A}$ determines, non-uniquely, a frame in which it is (up to normalization) an A-period. In this frame (the $\mathcal{A}$-frame), the holomorphic limit of the multi-instanton amplitude in the instanton sector of action $\ell \mathcal{A}$ is of the simple form (312), (313).

These conjectures are made based on the analysis of the resurgent structure of topological string theory. As reviewed in Section 4, the analytic structure of the Borel transform of a perturbative series encodes the instanton corrections to it, and can be determined from the series via various methods, including by studying its large order behavior (153) or its Stokes discontinuities (156). The former method proves particularly powerful in topological string theory. Trans-series associated to the instanton sectors labelled by $(\omega)$ of the form[20]

$$\mathcal{F}^{(\omega)} = g_s^{b_\omega - 2} e^{-\mathcal{A}_\omega/g_s} \left( \mathcal{F}_0^{(\omega)} + g_s \mathcal{F}_1^{(\omega)} + \dots \right), \tag{319}$$

imply the following asymptotic behavior of the perturbative free energy:

$$\mathcal{F}_g \sim \sum_\omega \frac{S_\omega}{2\pi} \sum_{k \geq 0} \mathcal{F}_k^{(\omega)} \frac{\Gamma(2g - b_\omega - k)}{\mathcal{A}_\omega^{2g - b_\omega - k}}$$
$$= \sum_\omega{}' \frac{S_\omega}{\pi} \frac{\Gamma(2g - b_\omega)}{\mathcal{A}_\omega^{2g - b_\omega}} \left( \mathcal{F}_0^{(\omega)} + \frac{\mathcal{F}_1^{(\omega)} \mathcal{A}_\omega}{2g - b_\omega - 1} + \frac{\mathcal{F}_2^{(\omega)} \mathcal{A}_\omega^2}{(2g - b_\omega - 1)(2g - b_\omega - 2)} + \dots \right). \tag{320}$$

---

[20]We have argued above that consistency of our formalism requires $b_\omega = 1$. We will find this value confirmed below.

Note that we have indicated the instanton sector as a superscript in parentheses. This is compatible with the notation introduced above in (167), if we label instanton sectors with the associated actions, and interpret the $^{(\ell)}$ appearing there as shorthand for $^{(\ell \mathcal{A})}$. As the perturbative free energies $F_g$ vanish at odd $g$, the Borel plane has an $\mathcal{A} \mapsto -\mathcal{A}$ symmetry. The primed sum in (320) takes this into account by summing over only one representative for each pair of instanton actions $\pm \mathcal{A}_\omega$. If one instanton sector has an action which is smaller in absolute value than all others, it will typically dominate the asymptotic behavior of $\mathcal{F}_g$.[21]

If we have analytic control over the asymptotics of $\mathcal{F}_g$ in a given frame, we can draw conclusions regarding the leading instanton contribution, together with the associated loop corrections. In topological string theory, we have such analytic control over $\mathcal{F}_g$ in two frames: the large radius frame, with asymptotics determined via the Gopakumar–Vafa formula (126), and the conifold frame, with asymptotics governed by the gap condition, (136). We can thus verify the two conjectures rigorously in these two frames.

### 5.7.1 Boundary conditions in the MUM frame

We start with the discussion of the $\mathcal{F}_g(X)$ near the MUM point, where the BPS indices $n_{g,\beta}$ in the Gopakumar–Vafa expansion (129), specifically the constant contribution $n_{0,0} = \chi/2$ leading to (131) and the first subleading $n_{0,|\beta|=1}$ contribution, yield the asymptotic behavior. This implies a dominating instanton sector with actions $\aleph \ell X^0$. We will show that the subleading asymptotics is governed by simple instanton amplitudes with actions $\aleph \ell (\beta \cdot X + n X^0)$ ($|\beta| \geq 1$). Here $X = (X^1, \ldots, X^{h_2(M)})$ and the $\cdot$ is used like in (48). In particular, we will show that the genus zero Gopakumar–Vafa invariants are realized as Stokes constants, as found empirically in a special case in [94].

We first consider the leading asymptotics, which is obtained from (131), as

$$\frac{\mathcal{F}_g(X)}{\Gamma(2g-1)} \sim \chi \left(\frac{X^0}{(2\pi i)^{3/2}}\right)^{2-2g} \frac{(-1)^{g+1} B_{2g} B_{2g-2}}{(2g)!(2g-2)!} \frac{2g-1}{2(2g-2)}$$
$$= -\frac{\chi}{2\pi^2} \left(\aleph X^0\right)^{2-2g} \left(1 + \frac{1}{2g-2}\right), \tag{321}$$

where the constant $\aleph$ was defined in (177). Comparing to (320) allows us to identify

$$\mathsf{S} = -\chi, \quad \mathcal{A} = \pm \aleph X^0, \quad b = 1, \quad \mathcal{F}_0^{(1)} = \frac{\mathcal{A}}{2\pi}, \quad \mathcal{F}_1^{(1)} = \frac{1}{2\pi}. \tag{322}$$

Recall that $^{(1)}$ is shorthand for $^{(\mathcal{A})}$. The subleading asymptotics of the Bernoulli numbers given in (132), keeping in mind

$$\zeta(2g) = \sum_{\ell \geq 1} \ell^{-2g}, \tag{323}$$

allows us to identify the multi-instanton contributions in this sector. From

$$\sum_{k \geq 1} \sum_{m \geq 1} k^{-2g+2} m^{-2g} = \sum_{\ell \geq 1} \sum_{m \mid \ell} l^{-2g} \left(\frac{\ell}{m}\right)^2 = \sum_{\ell \geq 1} \sigma_2(\ell) l^{-2g}, \tag{324}$$

where $\sigma_2(\ell)$ is the divisor sigma function

$$\sigma_2(\ell) = \sum_{m \mid \ell} m^2, \tag{325}$$

---

[21]Below, we will see instances of instanton actions which are appreciably different in absolute value competing against each other, as the associated instanton amplitudes differ by many orders of magnitude.

we conclude

$$\mathcal{F}_g(X) \sim -\frac{\chi}{2\pi^2} \sum_{\ell \geq 1} \ell^{-2} \sigma_2(\ell) \left( \aleph \ell X^0 \right)^{2-2g} \Gamma(2g-1) \left( 1 + \frac{1}{2g-2} \right). \tag{326}$$

Hence,

$$\mathsf{S}_\ell = -\chi \sigma_2(\ell), \quad \mathcal{A}_\ell = \ell \mathcal{A} = \pm \aleph \ell X^0, \quad b_\ell = 1, \quad \mathcal{F}_0^{(\ell)} = \frac{\mathcal{A}}{2\pi\ell}, \quad \mathcal{F}_1^{(\ell)} = \frac{1}{2\pi\ell^2}, \tag{327}$$

which is again of the simple form (312), (313). Note that the Stokes constants are proportional to the Euler character of the Calabi–Yau manifold.

To move beyond the contribution of the dominant action, consider the Gopakumar–Vafa formula (129) and substitute the leading asymptotics of the Bernoulli numbers (132) given by setting $\zeta(2n) \sim 1$ to obtain

$$\begin{aligned}
\mathcal{F}_g(X) &\sim \sum_{|\beta| \geq 1} \frac{2(2g-1)}{(2\pi)^{2g}} n_{0,\beta} \left( \frac{X^0}{(2\pi \mathrm{i})^{3/2}} \right)^{2-2g} \mathrm{Li}_{3-2g}(Q_\beta) \\
&= \sum_{|\beta| \geq 1} \frac{n_{0,\beta}}{2\pi^2} \Gamma(2g) \frac{(-1)^{g-1}}{(2g-2)!} \left( \frac{X^0}{\sqrt{2\pi\mathrm{i}}} \right)^{2-2g} \mathrm{Li}_{3-2g}(Q_\beta) \\
&= \sum_{|\beta| \geq 1} \frac{n_{0,\beta}}{2\pi^2} \Gamma(2g-1) \left( 1 + \frac{1}{2g-2} \right) \sum_{n \in \mathbb{Z}} \left( \frac{1}{\aleph(\beta \cdot X + nX^0)} \right)^{2g-2}.
\end{aligned} \tag{328}$$

In the final step, we have used that

$$\sum_{n \in \mathbb{Z}} \frac{1}{(2\pi n - 2\pi t \cdot \beta)^{2g-2}} = \frac{(-1)^{g-1}}{(2g-3)!} \mathrm{Li}_{3-2g}(Q_\beta). \tag{329}$$

Comparing to (320), we read off an infinite number of Borel singularities, each in accord with the boundary conditions in the form (312), (313):[22]

$$\mathsf{S}_\beta = n_{0,\beta}, \quad \mathcal{A}_{(\beta,n)} = \pm \aleph(\beta \cdot X + nX^0), \quad \mathcal{F}_0^{(1)} = \frac{\mathcal{A}_{\beta,n}}{2\pi}, \quad \mathcal{F}_1^{(1)} = \frac{1}{2\pi}. \tag{330}$$

Subleading contributions to the Bernoulli numbers given by contributions to (323) at $\ell > 1$ lead to multi-instanton sectors governed by Borel singularities at

$$\mathcal{A}_{(\beta,n),\ell} = \pm \aleph \ell (X \cdot \beta + nX^0), \tag{331}$$

with associated Stokes constants and instanton amplitudes

$$\mathsf{S}_{\beta,\ell} = n_{0,\beta}, \quad \mathcal{F}_0^{(\ell)} = \frac{\mathcal{A}_{(\beta,n)}}{2\pi\ell}, \quad \mathcal{F}_1^{(\ell)} = \frac{1}{2\pi\ell^2}. \tag{332}$$

Let us note that this computation leads to an identification of an infinite number of Stokes constants with genus zero Gopakumar–Vafa invariants, which are in particular *integers*. The integrality of Stokes constants has been conjectured in [23] and is in line with a similar situation in complex Chern-Simons theory [15, 16]. Here, it follows from the Gopakumar–Vafa formula, in which the multicovering-multibubbling of rational curves gets unravelled into a sequence of Borel singularities.

---

[22]In the one parameter cases discussed in section 6, we will write $d$, which stands for degree, instead of $\beta$ and accordingly $dX^1 + nX^0$ for $\beta \cdot X + nX^0$ etc.

Let us note further that higher genus Gopakumar–Vafa invariants do not seem to appear in the contributions from these singularities. The $g = 1$ contribution is entirely absent from (129): as is evident from the Gopakumar–Vafa form (126) of the topological string amplitude, there are no multi-covering contributions from $g = 1$ to other genera, $n_{1,\beta}$ only contributes to (122). Terms involving $n_{g,\beta}$ with $g \geq 2$ do arise in (129), but they do not have factorial growth: by (329), they lead to corrections of order $1/g$ to the factorial asymptotics. Of course, we expect to be able to find an *exact* formula for the $\mathcal{F}_g$ by adding contributions from all the Borel singularities. It seems that the part of the $\mathcal{F}_g$ due to genus zero Gopakumar–Vafa invariants comes directly from the Borel singularities that we studied above. The higher genus contributions must be contained in other singularities on the Borel plane.

### 5.7.2 Boundary conditions in a conifold frame

We will begin by slightly generalizing the result [20,21] that the asymptotics of the topological string amplitudes near a generalized conifold point in the associated conifold frame is dominated by an instanton sector whose action is $N$ times the conifold period $X_{\mathfrak{c}}^1 = \Pi_\nu = \int_\nu \Omega$, where $\nu = S^3/\mathbb{Z}_N$ is the vanishing lens space. The corresponding instanton amplitudes in the conifold frame are of the simple form (312), (313).

As explained in section 3.2, we can choose the coordinate

$$t_{\mathfrak{c}} = \frac{\Pi_\nu}{\Pi_\Gamma} = \frac{X_{\mathfrak{c}}^1}{X_{\mathfrak{c}}^0}, \tag{333}$$

to parametrize the transversal direction to a lens space conifold divisor $D_{\mathfrak{c}} \in \overline{\mathcal{M}}_{cs}(W)$. Here, $\nu, \Gamma \in H_3(W, \mathbb{Z})$, $\nu \cap \Gamma = 0$ and $\Pi_\Gamma(z_{\mathfrak{c}}) \neq 0$. From the gap condition (136), we obtain

$$\mathcal{F}_g(X_{\mathfrak{c}}) \sim \frac{(-1)^{g-1} B_{2g}}{2g(2g-2)} \left( \frac{X_{\mathfrak{c}}^1}{(2\pi i)^{1/2} N} \right)^{2-2g}. \tag{334}$$

Let us evaluate the asymptotics of

$$\frac{\mathcal{F}_g(X_{\mathfrak{c}})}{\Gamma(2g-1)} \sim \frac{(-1)^{g-1} B_{2g}}{2g(2g-2)(2g-2)!} \left( \frac{X_{\mathfrak{c}}^1}{(2\pi i)^{1/2} N} \right)^{2-2g}$$
$$= (-1)^{g-1} \frac{B_{2g}}{(2g)!} \left( \frac{X_{\mathfrak{c}}^1}{(2\pi i)^{1/2} N} \right)^{2-2g} \left( 1 + \frac{1}{2g-2} \right). \tag{335}$$

Invoking once again the asymptotics of the Bernoulli numbers which follows from (132) by setting $\zeta(2n) \sim 1$, we obtain

$$\frac{\mathcal{F}_g(X_{\mathfrak{c}})}{\Gamma(2g-1)} \sim \frac{1}{2\pi^2} \left( \frac{\aleph X_{\mathfrak{c}}^1}{N} \right)^{2-2g} \left( 1 + \frac{1}{2g-2} \right). \tag{336}$$

Comparing to (320), this allows us to identify

$$\mathsf{S}_\mu = 1, \quad \mathcal{A}_\mu = \pm \frac{\aleph X_{\mathfrak{c}}^1}{N}, \quad b = 1, \quad \mathcal{F}_0^{(1)} = \frac{\mathcal{A}_\mu}{2\pi}, \quad \mathcal{F}_1^{(1)} = \frac{1}{2\pi}. \tag{337}$$

Using (323) in (132) allows us to also identify multi-instanton sectors. From

$$\mathcal{F}_g(X_{\mathfrak{c}}) \sim \frac{1}{2\pi^2} \sum_{\ell \geq 1} \ell^{-2} \left( \frac{\aleph \ell X_{\mathfrak{c}}^1}{N} \right)^{2-2g} \Gamma(2g-1) \left( 1 + \frac{1}{2g-2} \right), \tag{338}$$

we conclude that

$$S_\ell = 1\,, \quad \mathcal{A}_\ell = \ell \mathcal{A}_\mu = \frac{\aleph \ell X_\mathfrak{c}^1}{N}\,, \quad b_\ell = 1\,, \quad \mathcal{F}_0^{(\ell)} = \frac{\mathcal{A}_\mu}{2\pi\ell}\,, \quad \mathcal{F}_1^{(\ell)} = \frac{1}{2\pi\ell^2}\,. \tag{339}$$

The instanton action is indeed an integral period, while the instanton amplitude is of the form (312), (313). We also find that the associated Stokes constant is trivial.

Note that for hypergeometric one-parameter models, the transition matrix (86) implies that we can choose

$$(X_\mathfrak{c}^0, X_\mathfrak{c}^1) = (X^1, P_0)\,. \tag{340}$$

# 6 Experimental evidence for resurgent structure

The computation of topological string amplitudes in one-parameter hypergeometric models via the holomorphic anomaly equations, described in detail in section 3, yields a trove of data to numerically test predictions following from the non-perturbative structure of topological string we obtained above, including the two conjectures concerning boundary conditions.

For this purpose, we will need additional tools from resurgence theory, in addition to formula (320) governing the large order behavior of the perturbative series as reviewed in section 4. Let $\mathcal{F}_g(X^*)$ be the holomorphic free energies at genus $g$ in a given frame. The Borel transform

$$\widehat{\mathcal{F}}^{(0)}(\zeta) = \sum_{g \geq 0} \frac{1}{(2g)!} \mathcal{F}_g(X^*)\zeta^{2g}\,, \tag{341}$$

of the perturbative free energy $\mathcal{F}^{(0)}(g_s)$

$$\mathcal{F}^{(0)}(g_s) = \sum_{g \geq 0} g_s^{2g-2} \mathcal{F}_g(X^*)\,, \tag{342}$$

will have singular points filling a subset of a lattice in the complex plane. These so-called Borel singularities of $\mathcal{F}^{(0)}(g_s)$ coincide with instanton actions $\mathcal{A}$, and therefore can be used to identify the latter.

In practice, $\mathcal{F}^{(0)}(g_s)$ is only known to finite order in $g_s$, so the Borel transform and consequently the Borel singularities cannot be computed exactly. Instead, we consider the Padé approximant of the Borel transform of the truncated series to approximate the exact Borel transform, and the position of the poles of the Padé approximant approximates the position of the Borel singularities. In the case of topological string theory, the Borel singularities of $\mathcal{F}^{(0)}$ turn out to be logarithmic branch points. Poles of the Padé approximants then accumulate to indicate the location of the branch cuts, and allow us to read off the location of the Borel singularities as the associated branch points.

Once we have identified instanton actions via a determination of the Borel singularities, we can proceed to test the associated instanton amplitudes. As reviewed in section 4, in addition to the large order formula (320), another useful numerical tool is the Stokes discontinuity. Suppose the perturbative free energy has Borel singularities $\ell \mathcal{A}_\omega$ ($\ell \geq 1$) along the Stokes ray $\mathcal{C}_\theta$. The Stokes discontinuity of the perturbative free energy across the Stokes ray $\mathcal{C}_\theta$ is a linear combination of the Borel resummation of the associated instanton amplitudes:

$$\text{disc}_\theta \mathcal{F}^{(0)}(g_s) = s_{\theta^+}\mathcal{F}^{(0)}(g_s) - s_{\theta^-}\mathcal{F}^{(0)}(g_s) = i \sum_{\omega \in \Omega_\theta} \sum_{\ell \geq 1} S_{\ell \mathcal{A}_\omega} s_{\theta^-} \mathcal{F}^{(\ell \mathcal{A}_\omega)}(g_s)\,. \tag{343}$$

In contradistinction to the large order formula (320), this allows the study of instanton sectors with instanton actions lying on a particular ray, independently of whether these are dominant.

Before turning to our numerical analysis, we wish to make several comments:

- The maximal genus to which perturbative free energies are available in compact models is not sufficient to check the non-trivial multi-instanton amplitudes presented in section 5.6. However, analogous results for local Calabi–Yau manifolds are testable [23], as the topological string amplitudes can be computed to higher genus in these cases.

- Although we will mainly focus on testing solutions of instanton amplitudes in the instanton sectors of action $X^1$ and $P_0$, the results derived in section 5 should apply generally. Once an instanton sector of action $\mathcal{A}$ has been identified (e.g. by studying the Borel singularities of $\mathcal{F}^{(0)}$), the first conjecture in section 5.7 implies that we can find a frame $\Gamma_{\mathcal{A}}$ in which $\mathcal{A}$ is an A-period; the second conjecture in section 5.7 then dictates that the instanton amplitude in frame $\Gamma_{\mathcal{A}}$ is of the simple form (312) and (313). We can then use the results of sections 5.5 and 5.6 to write down the instanton amplitudes in any other frame.

- Both the evaluation of perturbative and instanton free energies depend on the choice of frame. Nevertheless, our numerical studies seem to indicate that the resurgent structure of topological string is independent of the choice of frame. In particular, this means that the location of Borel singularities as well as the associated Stokes constants do *not* seem to change across different frames, at least as far as the leading Borel singularities, which are accessible by numerics, are concerned.

- When we are not in an $\mathcal{A}$-frame, the instanton amplitude $\mathcal{F}^{(\ell)}$ in the $\ell\mathcal{A}$ instanton sector itself contains a nontrivial power series and we can apply the same resurgent analysis as before: looking for Borel singularities and studying the associated higher instanton amplitudes. On the other hand, as already pointed out in [23] in the case of local Calabi–Yau manifolds, the instanton amplitudes are expressed completely in terms of the perturbative free energy, and consequently the resurgent structure of the former can be deduced from the latter. To discuss this point, it is convenient to use the language of alien derivatives. The formula (343) for the Stokes discontinuity suggests

$$\dot{\Delta}_{\ell\mathcal{A}}\mathcal{F}^{(0)}(g_s) = \mathsf{s}_{\ell\mathcal{A}}\mathcal{F}^{(\ell)}_{\ell}(g_s), \tag{344}$$

where the constants $\mathsf{s}_{\ell\mathcal{A}}$ can be derived from the Stokes constants $\mathsf{S}_{\ell\mathcal{A}}$. The subscript notation $\mathcal{F}^{(n)}_{\ell}$ used on the RHS was introduced below (313) to specify that the instanton amplitude is computed with the boundary condition (313). Then, for the full family of trans-series $\mathcal{F}^{(n|m)}_{\ell}$, we conjecture that

$$\dot{\Delta}_{\ell\mathcal{A}}\mathcal{F}^{(\ell n|\ell m)}_{\ell} = \mathsf{s}_{\ell\mathcal{A}}(n+1)\mathcal{F}^{(\ell(n+1)|\ell m)}_{\ell}(g_s). \tag{345}$$

This should follow directly from (344) by taking derivatives. The derivation of the case $\ell = 1$ goes through exactly as in [23].

*Another note on normalization conventions:*

The numerical calculations performed in this section require choosing a normalization for the periods, as well as a normalization of the topological string amplitudes, determined by the choice of constant $\alpha$ introduced in (143). We will indicate the normalization of the periods by citing the constant term of the holomorphic period at the MUM point. E.g., for the normalization chosen in (79), this constant is $(2\pi i)^3$. The values cited in this section are in terms of a proportionality constant $\aleph$. We performed our computations with a normalization $(2\pi i)^{3/2}$ of the periods (this is the normalization for which the triple intersection number $C_{zzz}$ has no factors of $2\pi i$) and $\alpha^2 = 1$; for this choice, $\aleph$ coincides with the value cited in (177). It is however equally possible to relate our results to a different normalization of the periods, by adjusting both $\alpha^2$ and $\aleph$. Two additional choices are indicated in table 5.

Table 5: The normalization constant $\aleph$ appearing in the numerical results cited in this section as a function of the normalization of the periods and of the topological string amplitudes $F_g$.

| normalization of periods | $\alpha^2$ | $\aleph$ |
|---|---|---|
| 1 | $(2\pi\mathrm{i})^{-3}$ | $\mathrm{i}(2\pi\mathrm{i})^2$ |
| $(2\pi\mathrm{i})^{3/2}$ | 1 | $\mathrm{i}\sqrt{2\pi\mathrm{i}}$ |
| $(2\pi\mathrm{i})^3$ | $(2\pi\mathrm{i})^3$ | $\frac{1}{2\pi}$ |

## 6.1 Mapping out the Borel plane

In this subsection, we investigate the Borel singularities of the perturbative free energy $\mathcal{F}^{(0)}$ evaluated in different frames and at different points of moduli space.

We first consider the vicinity of the MUM point $z = 0$. For the perturbative free energy $\mathcal{F}^{(0)}(X^0, X^1)$ evaluated in the MUM frame, the discussion in section 5.7.1 predicted Borel singularities at[23]

$$\pm \aleph \ell X^0, \quad \pm \aleph (dX^1 + nX^0), \quad \ell, d \geq 1, n \in \mathbb{Z}. \tag{346}$$

The first few of these, in the case of the quintic, are nicely visible in the plot reproduced in Figure 4 (left). In particular, we find the branch cuts due to $\pm \aleph X^0$ along the positive and negative imaginary axes, and the two towers of Borel singularities due to $\aleph(\pm X^1 + \mathbb{Z}X^0)$, forming a peacock pattern already observed in local Calabi–Yau manifolds [80,84]. The peacock pattern becomes more prominent if we subtract the dominant constant map contributions from the free energies. The result is plotted in Figure 4 (right). On the other hand, the Borel singularities $\aleph \ell X^0$ with higher $\ell$ are hidden behind the branch cuts of $\ell = 1$. But we can uncover the first few by combining the Padé approximant of the Borel transform with a conformal map

$$\zeta = \frac{1}{\mathrm{i}} \frac{2\mathcal{A}\xi}{1 - \xi^2}. \tag{347}$$

See for example [95] for a summary of this type of techniques in Borel analysis. The resulting plot, shown in Figure 5, displays clearly the singularities with $\ell = 2, \ldots$ in the $\xi$-plane.

We can also evaluate the free energies in other frames, for example in the conifold frame defined by $(X^1, P_0)$, or frames that are not associated to special points in the moduli space, such as e.g. the frame defined by specifying as A-periods $(P_0, P_1)$. The Borel singularities of $\mathcal{F}^{(0)}(X^1, P_0)$ and $\mathcal{F}^{(0)}(P_0, P_1)$ near the MUM point are plotted in Figure 6. We find that the positions of the visible singularities coincide with those of $\mathcal{F}^{(0)}(X^0, X^1)$.

Next, we consider loci in moduli space close to the conifold point $z = \mu$. Figure 7 (left) shows a plot of the Borel singularities of $\mathcal{F}^{(0)}(X^1, P_0)$. The only singular points that are visible are the branch points at

$$\pm \aleph P_0, \tag{348}$$

consistent with the asymptotic analysis in section 5.7.2 based on the gapped form (136) of the topological string amplitudes $\mathcal{F}_g$ close to the conifold locus. Figure 7 (right) shows the Borel singularities of $\mathcal{F}^{(0)}(X^1, P_0)$ upon subtracting the leading poles before the gap to obtain the regularized free energies

$$\mathcal{F}_g^{\mathrm{reg}}(X^1, P_0) = \mathcal{F}_g(X^1, P_0) - \frac{(-1)^{g-1} B_{2g}}{2g(2g-2)} \left( \frac{(2\pi\mathrm{i})^{1/2}}{P_0} \right)^{2g-2}, \quad g > 1. \tag{349}$$

---

[23]Due to the symmetry of perturbative free energies under $g_s \to -g_s$, Borel singularities $\pm \mathcal{A}$ always appear in pairs.

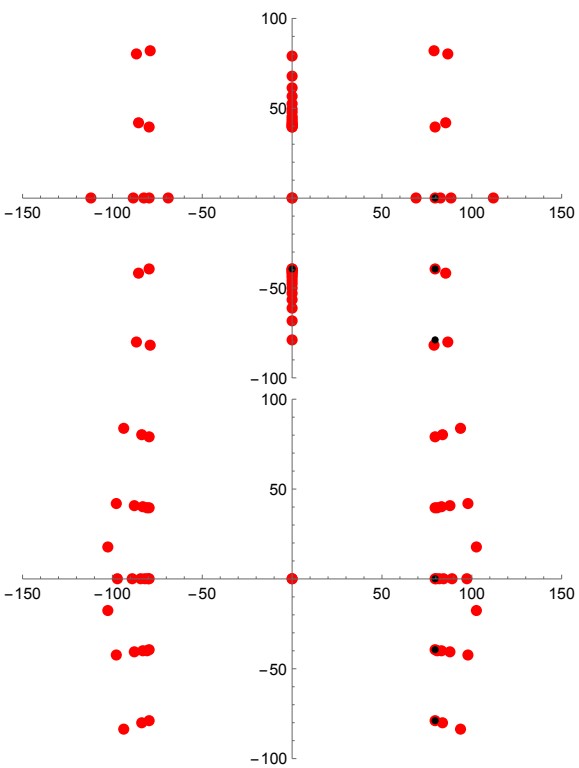

Figure 4: The location of the poles of the Padé approximant to $\widehat{\mathcal{F}}^{(0)}(X^0, X^1)$ (the hat indicates the Borel transform), evaluated to order $g = 64$ at $z = 10^{-2}\mu$. On the right, the leading constant map contribution is subtracted. The black dots correspond to the position of the periods $\aleph(mX^0 + nX^1)$, $(m,n) = (1,0)$ (on the imaginary axis), $(0,1),(1,1),(2,1),\ldots$

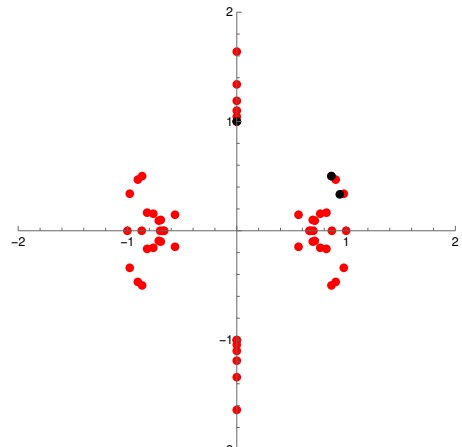

Figure 5: The location of the poles of the Padé approximant to $\widehat{\mathcal{F}}^{(0)}(X^0, X^1)$ as a function of $\zeta$, evaluated to order $g = 61$ at $z = 10^{-3}\mu$, mapped to the $\xi$-plane by the conformal map (347). The black dots correspond to the position of the periods $\aleph mX^0$, $m = 1, 2, 3$. The red dots inside a circle of radius one are mappings of the singularities from the two towers.

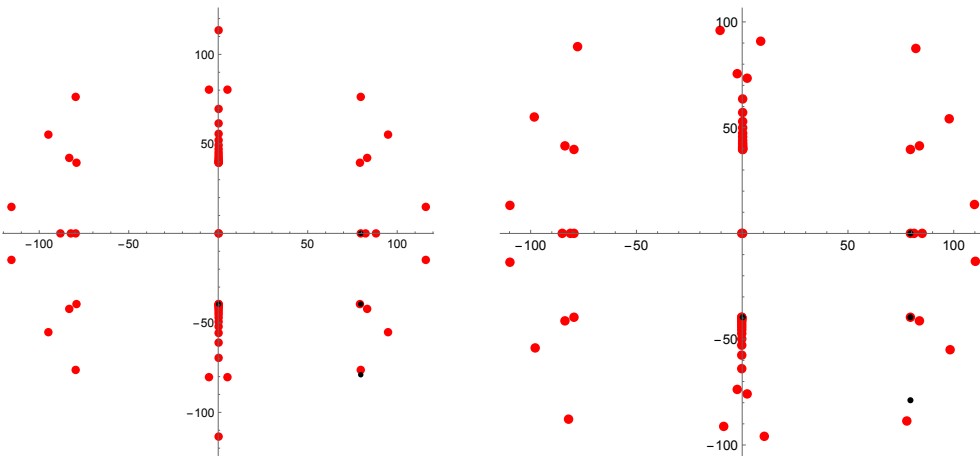

Figure 6: The location of the poles of the Padé approximant to $\widehat{\mathcal{F}}^{(0)}(X^1, P_0)$, on the left, and $\widehat{\mathcal{F}}^{(0)}(P_0, P_1)$, on the right, evaluated to order $g = 64$ at $z = 10^{-2}\mu$. The black dots correspond to the position of the periods $\aleph(mX^0 + nX^1)$, $(m, n) = (1, 0)$ (on the imaginary axis), $(0, 1), (1, 1), (2, 1), \ldots$

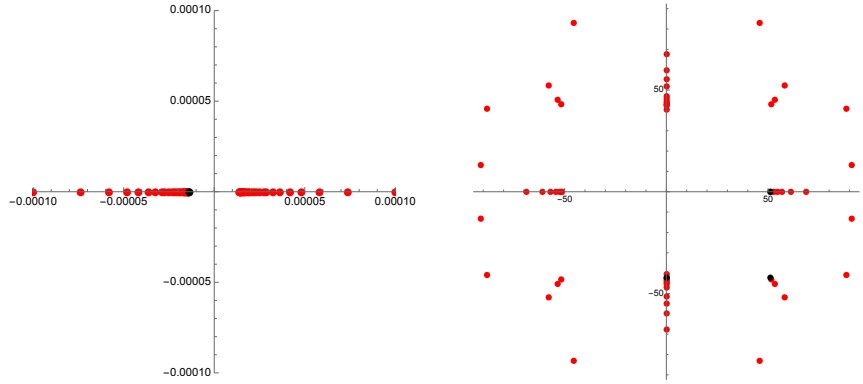

Figure 7: The location of the poles of the Padé approximant to $\widehat{\mathcal{F}}^{(0)}(X^1, P_0)$, without (left) and with (right) the leading singularity removed, evaluated to order $g = 61$ at $z = (1 - 10^{-6})\mu$. In the left diagram, the black dot is at the position of the period $\aleph P_0$. In the right diagram, the black dots are at the position of the periods $\aleph X^0$, $\aleph X^1$, $\aleph(X^0 + X^1)$.

Compared to the figure on the left, additional singularities located at

$$\pm \aleph X^0, \quad \pm \aleph X^1, \quad \pm \aleph(X^0 \pm X^1), \tag{350}$$

become visible. Their location cannot be predicted analytically. We will study these additional singular points in section 6.2.

Similarly, we can evaluate the free energies in other frames, for instance in the MUM frame defined by the designation of A-periods $(X^0, X^1)$ or the frame defined by the designation of A-periods $(P_0, P_1)$. The Borel singularities of $\mathcal{F}^{(0)}(X^0, X^1)$ and of $\mathcal{F}^{(0)}(P_0, P_1)$ close to the conifold point are plotted in Figure 8. They coincide with the Borel singularities in the conifold frame close to this point, plotted in Figure 7 (left).

We note that all three frames considered exhibit the same visible Borel singularities both close to the MUM point and close to the conifold point. As far as the leading singularities are concerned, the reason for this universality can perhaps be traced to the fact that the dominant contribution to the topological string amplitudes in both of these regions is due to the

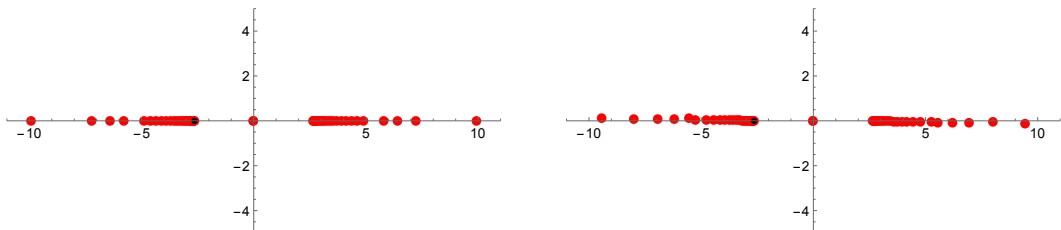

Figure 8: The location of the poles of the Padé approximant to $\widehat{\mathcal{F}}^{(0)}(X^0, X^1)$, on the left, and $\widehat{\mathcal{F}}^{(0)}(P_0, P_1)$, on the right, evaluated to order $g = 64$ at $z = \frac{5}{6}\mu$. The black dots correspond to the position of the period $\aleph P_0$.

holomorphic ambiguity $f_g$. In Figure 9, we plot the singularities of the Borel transform of

$$\mathcal{F}^{\text{hol}}(g_s) = \sum_g f_g g_s^{2g-2}, \tag{351}$$

close to the conifold and the MUM point. Indeed, the dominant Borel singularities coincide with those in the three frames considered above.

Before ending this subsection, we comment on an interesting observation regarding the monodromy action on the Borel singularities. As we have seen, the Borel singularities of the perturbative free energies are given by geometric periods of the Calabi–Yau manifolds. The periods are known to enjoy monodromy actions around singular points of the moduli space. For example, in the case of the quintic, if we start from some locus in the moduli space, circle around the MUM point and come back to the original locus, the periods transform by (we omit here the transformation of the other two periods, which are also non-trivial)

$$X^1 \to X^1 + X^0. \tag{352}$$

Therefore, near the MUM point the peacock pattern of Borel singularities arises by consistency: as long as there exists one singular point of the type $\aleph(dX^1 + nX^0)$ with $d \neq 0$, all the other singular points of the type $\aleph(dX^1 + \mathbb{Z}X^0)$ will appear by repeatedly applying the MUM monodromy action, generating a vertical tower of Borel singularities displaced by $dX^1$ from the origin. In other words, the distribution of Borel singularities near the MUM point is invariant under the MUM monodromy action.

The same phenomenon is observed near an orbifold point. Taking again the quintic as an example, which has an orbifold point at $z = \infty$, it is natural to use the orbifold frame defined by the periods $(X^0_o, X^1_o) = (w^{1/5}(1 + O(w)), w^{2/5}(1 + O(w)))$ with $w = 1/z$. A plot of Borel singularities of $\mathcal{F}^{(0)}(X^0_o, X^1_o)$ is given in Figure 10. The visible Borel singularities are located at

$$\aleph \vec{q} \cdot \vec{\Pi}, \tag{353}$$

with charge vectors

$$\vec{q} \in \{\pm(0,0,0,1),\ \pm(0,0,-1,1),\ \pm(-1,-1,3,-1),\ \pm(2,1,-3,-2),\ \pm(-1,0,1,1)\}, \tag{354}$$

defined relative to the period vector defined in (79). This distribution of Borel singularities is also invariant under the monodromy action around the orbifold point given by

$$\vec{q} \to \vec{q} \cdot M_\infty, \tag{355}$$

where the orbifold monodromy action is

$$M_\infty = \begin{pmatrix} 1 & 1 & -5 & 2 \\ 0 & 1 & -3 & 5 \\ 1 & 1 & -4 & 2 \\ 0 & 0 & -1 & 1 \end{pmatrix}. \tag{356}$$

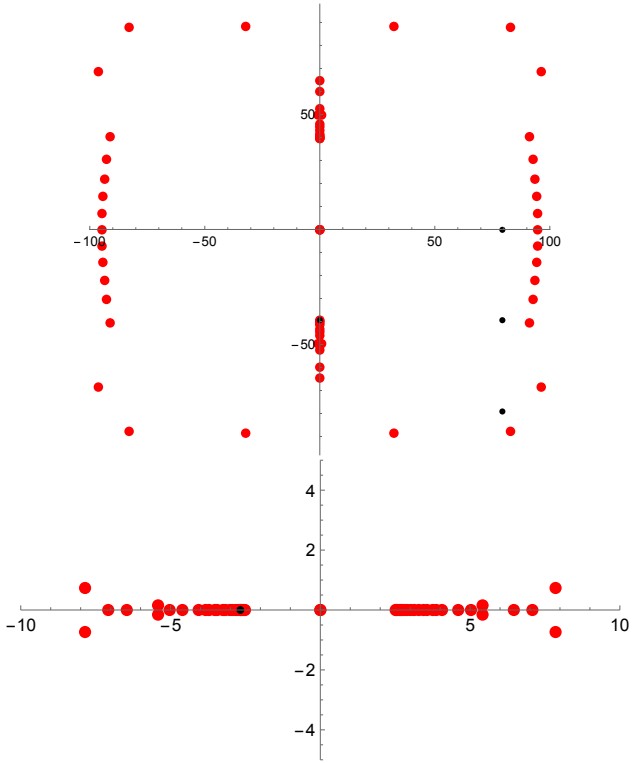

Figure 9: The location of the poles of the Padé approximant to $\widehat{\mathcal{F}}^{\text{hol}}$ evaluated to order $g = 64$ at the point $z = 10^{-2}\mu$ on the left, and $z = \frac{5}{6}\mu$ on the right. The black dots on the left give the positions of the periods $\aleph(mX^0 + nX^1)$, $(m, n) = (1, 0), (0, 1), (1, 1), (2, 1)$, while the black dot on the right gives the position of $\aleph P_0$.

In fact, the ten charge vectors in (354) with the overall $\pm$ signs form two orbits of length 5 of the monodromy action $M_\infty$.

The situation near the conifold point $z = \mu$ is different. The monodromy action around the conifold point of the quintic is

$$X^0 \to X^0 + P_0\,. \tag{357}$$

By the same argument as near the MUM point, near the conifold point we would expect due to the presence of the Borel singularities at $\pm\aleph X^0$ two (horizontal) towers of Borel singularities given by $\aleph(X^0 + \mathbb{Z}P_0)$, which are nevertheless conspicuously absent in Figure 7.

The failure of monodromy invariance is reminiscent of a phenomenon pointed out by Seiberg and Witten [96] regarding the BPS spectrum of $\mathcal{N} = 2$ supersymmetric gauge theory. Stokes constants in topological string theory have been conjectured to be related to BPS invariants [14, 80]. Above, we have found Borel singularities of perturbative free energies to be located at geometric periods, which according to homological mirror symmetry are proportional to central charges of D-branes in type IIA superstring theory. It is tempting to link the Borel singularities to stable D-branes. Then, if a wall of marginal stability ends in the conifold point, it cannot be avoided when circling the conifold point and some stable D-branes will decay. As a consequence, we shall not expect the spectrum of stable D-branes, hence the distribution of Borel singularities, to be invariant under the conifold monodromy action. Conversely, when the distribution of Borel singularities is observed to be invariant under the MUM (resp. orbifold) monodromy action near the MUM point (resp. orbifold point) as above, this may indicate that walls of marginal stability can be avoided upon circling the MUM point (resp. orbifold point).

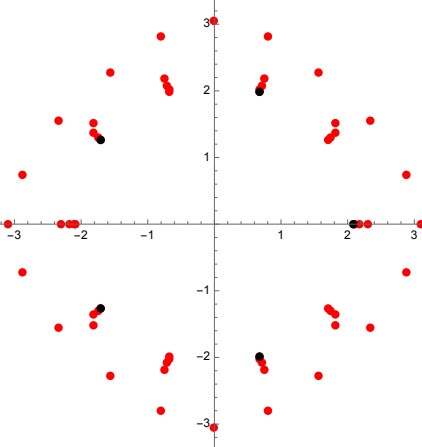

Figure 10: The location of the poles of the Padé approximant to $\mathcal{F}^{(0)}(X_o^0, X_o^1)$, evaluated to order $g = 60$, at $z = 10^8 \mu$. The black dots correspond to periods with charge vectors $(0,0,0,1), (0,0,-1,1), (-1,-1,3,-1), (2,1,-3,-2), (-1,0,1,1)$.

## 6.2 Experimental evidence for boundary conditions

In this subsection, we provide additional numerical evidence for the conjecture that the instanton amplitudes simplify in a frame where the instanton action $\mathcal{A}$ is an A-period and take the form given in (312), (313).

We start by numerically verifying the subleading asymptotics of $\mathcal{F}^{(0)}(X^0, X^1)$ near the MUM point, as worked out based on the Gopakumar–Vafa form of the topological string amplitude in section 5.7.1.

Let us consider the sequence

$$s_g = 2\pi^2 \left(\mathcal{A}_{(1,0)}\right)^{2g-2} \frac{\mathcal{F}_g(X^0, X^1)}{\Gamma(2g-1) + \Gamma(2g-2)}, \tag{358}$$

which, according to (328) restricted to $d = 1$ and $n \in \{0, \pm 1\}$, should have an oscillatory behavior

$$s_g \sim n_{0,1} \left\{1 + 2 \left(1 + \left(X^0/X^1\right)^2\right)^{1-g} \cos\left(2(g-1)\phi\right)\right\}, \tag{359}$$

where

$$\phi = \tan^{-1}\left(X^0/X^1\right). \tag{360}$$

In Figure 11, on the left, we show the sequence $s_g$ (black dots) as compared to the expected oscillatory behavior on the r.h.s. of (359) (red line), when $z = 10^{-5}$. We can subtract the oscillatory part from $s_g$ to obtain a sequence that converges to $n_{0,1} = 2875$ (without subleading $1/g$ corrections). This is shown in Figure 11, on the right. This gives an approximation to 2875 with a precision of $10^{-10}$.

The discussion regarding the implications of the Gopakumar–Vafa formula for the asymptotics of $\mathcal{F}^{(0)}(X)$ is general. It applies to multi-parameter Calabi–Yau manifolds as well as to local Calabi–Yau manifolds. As the topological string amplitudes are computable to higher genus in the local case, the asymptotics displayed in (328) can be numerically checked beyond degree 1. Consider e.g. local $\mathbb{P}^2$. Having tested the asymptotics (328) for $d = 1$, one subtracts the resolved conifold piece corresponding to $d = 1$, so as to eliminate all the Borel singularities with $d = 1$. One then proceeds with $d = 2$, and so on. Let us give an example. Consider the sequence

$$s_g = \pi^2 \frac{\Gamma(2g)}{2g-2} \mathcal{A}_{(3,0)}^{2g-2} \left\{\mathcal{F}_g(t) - \frac{(-1)^{g-1} B_{2g}}{2g(2g-2)!} \left(3\text{Li}_{3-2g}(e^{-t}) - 6\text{Li}_{3-2g}(e^{-2t})\right)\right\}. \tag{361}$$

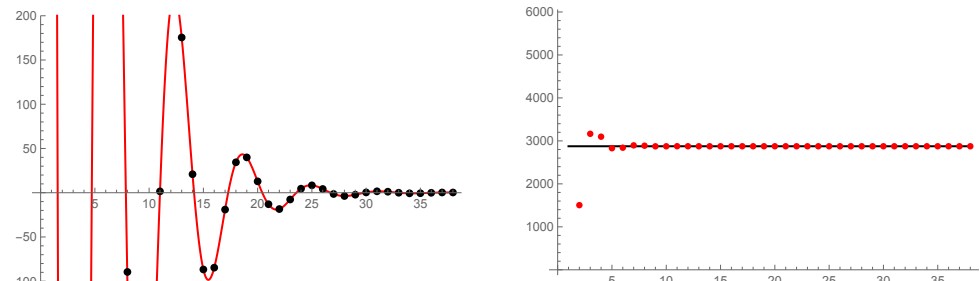

Figure 11: On the left, the sequence $s_g$ (black dots) as compared to the expected oscillatory behavior (359) (red line), when $z = 10^{-5}$. On the right, the sequence $s_g$ minus its oscillatory part (red dots) and the prediction 2875 (black line).

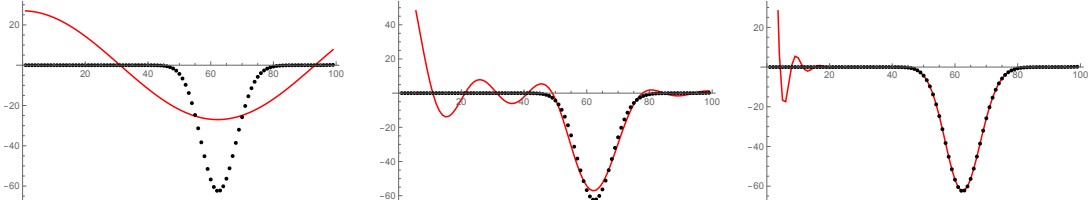

Figure 12: The sequence $s_g$ for $z = -10^{-8}$ in local $\mathbb{P}^2$ is represented in black dots and compared to the expected oscillatory behavior (362) (red line), where we include only a finite number of terms in the sum. In the first figure, we include only $m = 0$, in the second $m = 0, 1, 2$, and in the third $m = 0, 1, \cdots, 8$.

Here, we have subtracted the pieces that correspond to the Gopakumar–Vafa invariants of degree one, $n_{0,1} = 3$, and two, $n_{0,2} = -6$. What remains is the asymptotics governed by the degree 3 genus zero Gopakumar–Vafa invariant $n_{0,3} = 27$. Therefore, according to (328), the sequence should asymptote to

$$s_g \sim 27 \sum_{m \geq 0} \left| \frac{\mathcal{A}_{(3,m)}}{\mathcal{A}_{(3,0)}} \right|^{2-2g} \cos\left(2(g-1)\phi_m\right), \tag{362}$$

where $\phi_m = \arg(\mathcal{A}_{(3,m)})$ and only the term with $\ell = 1$ is kept (the terms $\ell \geq 2$ lead to exponential corrections to the asymptotics). In order to check this, we have to go to a region where $\mathcal{A}_{(3,0)}$ is smaller than the conifold action. Since the actions $\mathcal{A}_{(3,m)}$ are quite close to each other in absolute value, for small values of $m$, one has to add them to correctly reproduce the asymptotics. In Figure 12, we make this comparison, for $z = -10^{-8}$. When enough terms are added, the asymptotic formula reproduces the sequence with high precision. Note that the flat region at the very beginning of the sequence is only reproduced when we included many oscillations, as in Fourier analysis.

We will now move beyond the analytic predictions in Section 5.7, and check the boundary conditions in an instanton sector for which we have no analytic control over the asymptotics of the topological string amplitudes. We will focus on the $X^1$ instanton sector in the conifold frame defined by the A-periods $(X^1, P_0)$. As $X^1$ is an element of the set of A-periods defining the frame, the second conjecture formulated in section 5.7 predicts that the associated instanton amplitude should be of the simple form (312), (313).

We first consider the case of the quintic. Unfortunately, $X^1$ is never the dominant instanton sector in the range between the MUM point and the conifold point. This is because, as we have seen from Figure 6 (left) and Figure 7 (right), both $X^0$ and $X^1$ appear as Borel singularities in this region, and the $X^0$ instanton sector always dominates the $X^1$ instanton sector, as $X^0$ is

Table 6: For the quintic in the conifold frame defined by the A-periods $(X^1, P_0)$ at $z = (1 - 10^{-6})\mu$, we compare the Stokes discontinuity (underlined are stabilized digits) across the positive real axis where the Borel singular point $\aleph X^1$ is located, and the value of the simple one-instanton amplitude (312) with $\tau_k = \delta_{k,1}/(2\pi)$ of instanton action $\mathcal{A} = \aleph X^1$, evaluated with $\mathcal{F}^{(0)}$ up to order $g = 61$. The ratio of the two results is constant for a suitable range of $g_s$ and it is close to $n_{0,1} = 2875$.

| $g_s$ | $\mathrm{disc}_0 \mathcal{F}^{(0)}$ | $\mathcal{F}^{(1)}_{\mathcal{A}}$ | $\mathrm{disc}_0 \mathcal{F}^{(0)}/\mathcal{F}^{(1)}_{\mathcal{A}}$ |
|---|---|---|---|
| 1/2 | $\underline{2.0727336107126} \times 10^{-40}\,\mathrm{i}$ | $7.2095082301714 \times 10^{-43}\,\mathrm{i}$ | 2874.99999 |
| 1 | $\underline{1.5789470403404} \times 10^{-18}\,\mathrm{i}$ | $5.4919897055452 \times 10^{-22}\,\mathrm{i}$ | 2874.99999 |
| 3/2 | $\underline{2.6261206}071978 \times 10^{-11}\,\mathrm{i}$ | $9.1343325745150 \times 10^{-15}\,\mathrm{i}$ | 2874.99999 |
| 2 | $\underline{9.8839645}377145 \times 10^{-8}\,\mathrm{i}$ | $3.4379000230359 \times 10^{-11}\,\mathrm{i}$ | 2875.00057 |
| 5/2 | $\underline{1.31813}44321030 \times 10^{-5}\,\mathrm{i}$ | $4.5848197859366 \times 10^{-9}\,\mathrm{i}$ | 2874.99725 |

smaller than $X^1$ in absolute values here, cf. Figure 3. As the $X^0$ instanton amplitude is nontrivial in the conifold frame, its effect on the asymptotics of perturbative free energies cannot be easily subtracted. It is therefore difficult to extract information on the $X^1$ instanton sector from the large order asymptotics of the perturbative free energies of the quintic.

Instead, we compute the Stokes discontinuity across the positive real axis, on which $X^1$ is located. By (343), the result will be proportional to the sum over all Borel resummed instanton amplitudes associated to actions lying on the real axis. In addition to $X^1$, $P^0 \in \mathbb{R}$ close to the conifold point. However, unlike the contribution of the $X^0$ instanton sector, the contribution of the $P^0$ sector (and the associated multi-instanton sectors) can readily be subtracted from the asymptotics. To compute the Stokes discontinuity associated to the $X^1$ instanton sector, we therefore consider the Borel transforms of the sum of the regularized free energies $\mathcal{F}^{\mathrm{reg}}_g(X^1, P_0)$ defined in (349). As we only have access to finitely many orders in the perturbation series, the Borel transforms we work with numerically do not exhibit logarithmic singularities. It turns out however that the poles of the Padé approximant suffice to obtain the discontinuity of the Borel resummation to high numerical precision. The Laplace transform integral (154) can either be performed numerically along a path slightly above/below the positive real axis, or the discontinuity (156) can be computed directly as a sum over the residues of the integrand (the difference of the upper/lower semi-circle contributions from the (+), (−) integration path respectively yield the full residue at each pole). Either way, the result should be proportional to the Borel resummed $X^1$ instanton amplitude, with proportionality constant the associated Stokes constant. We give the result of the ratio of the discontinuity and the instanton amplitude $\mathcal{F}^{(1)}_{\mathcal{A}}$ given by (312) associated to the instanton action $\mathcal{A} = \aleph X^1$, evaluated at the special boundary condition $\tau_k = \delta_{1,k}/(2\pi)$, in Table 6. We find that this ratio is constant for a suitable range of $g_s$ (dependent on the order to which the perturbative series is available), and it agrees with

$$n_{0,1} = 2875. \tag{363}$$

Note that this is the same Stokes constant associated to the $X^1$ instanton sector in the MUM frame, cf. (330).

Another Calabi–Yau manifold which is of particular interest is $X_{2,2,2,2}(1^8)$, which has the property that near the conifold point, $\aleph X^1$ is smaller than $\aleph X^0$ in magnitude, so that $\pm \aleph X^1$ are the dominant Borel singularities for the regularized conifold free energy $\mathcal{F}^{(0)\mathrm{reg}}(X^1, P_0)$ defined in (349). For instance at $z = (1 - 10^{-6})\mu$,

$$\aleph X^0 = 44.1619\ldots\mathrm{i}, \qquad \aleph X^1 = 35.6383\ldots \tag{364}$$

We can then apply the technique of large order asymptotics. We find that

$$\mathcal{F}^{(0)\,\mathrm{reg}}_g(X^1, P_0) \sim \frac{\mathsf{S}_1}{\pi} \frac{\Gamma(2g-1)}{\mathcal{A}_1^{2g-1}} \left( \mathcal{F}^{(1)}_0(X^1, P_0) + \mathcal{F}^{(1)}_1(X^1, P_0) \frac{\mathcal{A}_1}{2g-2} \right), \tag{365}$$

where

$$\mathsf{S}_1 = n_{0,1} = 512, \quad \mathcal{A}_1 = \aleph X^1, \quad \mathcal{F}^{(1)}_0(X^1, P_0) = \frac{\mathcal{A}_1}{2\pi}, \quad \mathcal{F}^{(1)}_1(X^1, P_0) = \frac{1}{2\pi}. \tag{366}$$

This implies that the one-instanton amplitude associated to the instanton action $\mathcal{A}_1$ is indeed of the type (312), (313). For instance, to find $\mathsf{S}_1 \mathcal{F}^{(1)}_0(X^1, P_0)/\pi$, we can study the sequence

$$s^{(1)}_g = \mathcal{F}^{(0)\,\mathrm{reg}}_g(X^1, P_0) \frac{\mathcal{A}_1^{2g-1}}{\Gamma(2g-1)}, \tag{367}$$

which asymptotes to $\mathsf{S}_1 \mathcal{F}^{(1)}_0(X^1, P_0)/\pi$ in large $g$ and the convergence can be made faster using the Richardson transformation. Similarly, once $\mathsf{S}_1 F^{(1)}_0(X^1, P_0)/\pi$ is found, we can study the sequence

$$s^{(2)}_g = \left( \mathcal{F}^{(0)\,\mathrm{reg}}_g(X^1, P_0) \frac{\mathcal{A}_1^{2g-1}}{\Gamma(2g-1)} - \frac{\mathsf{S}_1 \mathcal{F}^{(1)}_0(X^1, P_0)}{\pi} \right) \frac{2g-2}{\mathcal{A}_1}, \tag{368}$$

which asymptotes to $\mathsf{S}_1 \mathcal{F}^{(1)}_1(X^1, P_0)/\pi$ at large $g$. Finally, we can use the sequence

$$s^{(3)}_g = \left( \mathcal{F}^{(0)\,\mathrm{reg}}_g(X^1, P_0) \frac{\mathcal{A}_1^{2g-1}}{\Gamma(2g-1)} - \frac{\mathsf{S}_1 \mathcal{F}^{(1)}_0(X^1, P_0)}{\pi} - \frac{\mathsf{S}_1 \mathcal{F}^{(1)}_1(X^1, P_0)}{\pi} \frac{\mathcal{A}_1}{2g-2} \right) \frac{(2g-2)(2g-3)}{\mathcal{A}_1^2}, \tag{369}$$

to explore a possible third term $\mathsf{S}_1 \mathcal{F}^{(1)}_2(X^1, P_0)/(\pi)$. Here we always use

$$\mathsf{S}_1 = n_{0,1} = 512. \tag{370}$$

The sequences of $s^{(1)}_g, s^{(2)}_g, s^{(3)}_g$ are illustrated in Figs. 13. It is clear that $\mathcal{F}^{(1)}_{0,1}(X^1, P_0)$ fit well and $F^{(1)}_2(X^1, P_0)$ should be zero.

As in the case of the quintic, another test we can perform is to compute the Stokes discontinuity. The Stokes discontinuity of the perturbative free energy across the positive real axis should be

$$\mathrm{disc}_0 \mathcal{F}^{(0)\,\mathrm{reg}}(X^1, P_0) = \frac{\mathsf{S}_1}{2\pi} \left( \frac{\mathcal{A}_1}{g_s} + 1 \right) e^{-\mathcal{A}_1/g_s}. \tag{371}$$

This is also verified by numerical calculation as shown in Table 7.

## 6.3 Checking one-instanton amplitudes against asymptotics and Stokes discontinuities

In this section, we provide strong evidence for the solution of the nontrivial one-instanton amplitude (278) from the holomorphic anomaly equations. We will focus on two instanton sectors: the $P_0$ instanton sector in the MUM frame defined by the A-periods $(X^0, X^1)$, and the $X^0$ instanton sector in the conifold frame defined by the A-periods $(X^1, P_0)$.

The asymptotic behavior of $\mathcal{F}_g = \mathcal{F}_g(X^0_*, X^1_*)(z)$ at large $g$ is governed by instanton amplitudes via the relation

$$\mathcal{F}_g \sim \sum_\omega \frac{\mathsf{S}_\omega}{2\pi} \frac{\Gamma(2g-b_\omega)}{\mathcal{A}_\omega^{2g-b_\omega}} \left( \mathcal{F}^{(\omega)}_0 + \frac{\mathcal{F}^{(\omega)}_1 \mathcal{A}_\omega}{2g-b_\omega-1} + \frac{\mathcal{F}^{(\omega)}_1 \mathcal{A}_\omega^2}{(2g-b_\omega-1)(2g-b_\omega-2)} + \dots \right), \tag{372}$$

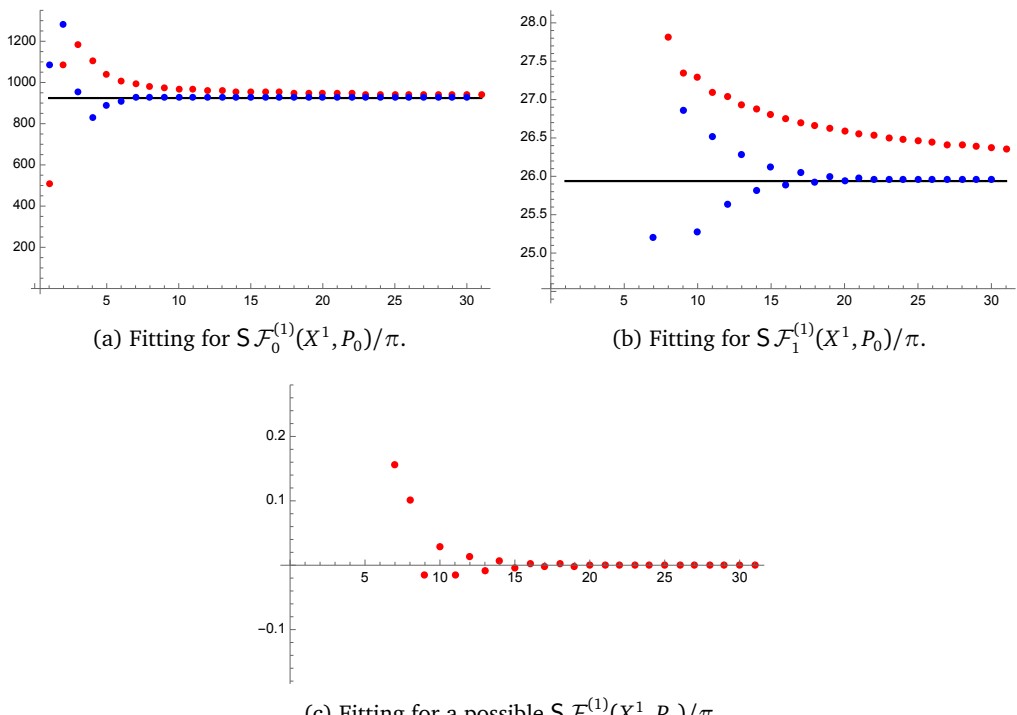

(a) Fitting for $\mathsf{S}\mathcal{F}_0^{(1)}(X^1, P_0)/\pi$.

(b) Fitting for $\mathsf{S}\mathcal{F}_1^{(1)}(X^1, P_0)/\pi$.

(c) Fitting for a possible $\mathsf{S}\mathcal{F}_2^{(1)}(X^1, P_0)/\pi$.

Figure 13: Comparison between auxiliary sequences (367),(368), (369) (red dots) which asymptote to $\mathsf{S}_1\mathcal{F}_{0,1,2}^{(1)}(X^1, P_0)/\pi$ with $\mathsf{S}_1 = n_{0,1}$, their Richardson transforms (1st-order, blue dots), and the prediction (366) (black lines) for the example of the $X_{2,2,2,2}(1^8)$ model at $z = (1 - 10^{-6})\mu$.

according to (320). The RHS of this relation depends on the Stokes constants $\mathsf{S}_\omega$, the constants $b_\omega$, the instanton actions $\mathcal{A}_\omega$, and the associated instanton amplitudes $\mathcal{F}_g^{(\omega)}$. As noted above, we will also label, when convenient, an instanton sector $\omega$ by the associated action $\mathcal{A}_\omega$.

The instanton sectors which dominate the asymptotics of $\mathcal{F}_g$ can be read off the Borel plane plots studied in section 6.1. When the asymptotics is dominated by a single instanton sector,[24] the corresponding instanton action can be extracted from the asymptotics by considering the auxiliary sequence

$$s_g^{(\mathcal{A})} = 2g\sqrt{\frac{\mathcal{F}_g}{\mathcal{F}_{g+1}}} \quad \xrightarrow{g \gg 1} \quad \mathcal{A}. \tag{373}$$

We can accelerate the convergence of the sequence by using Richardson transformations.

As argued for in section 5, consistency of our ansatz requires $b_\omega = 1$. This condition can also be verified numerically: if the asymptotics is dominated by the instanton action $\mathcal{A}$, then the associated constant $b$ will be the limit of the sequence

$$s_g^{(b)} = -\frac{1}{2}\left(\frac{\mathcal{A}^2}{2g}\frac{\mathcal{F}_{g+1}}{\mathcal{F}_g} - 2g - 1\right) \quad \xrightarrow{g \gg 1} \quad b. \tag{374}$$

Most significantly, the one-instanton amplitudes can be extracted from the asymptotics and compared to our theoretical prediction. Again assuming that the asymptotics is dominated by

---

[24]Recall that due to the occurrence of only even powers of $g_s$ in the asymptotic series (1), all instanton sectors occur in pairs $\pm\mathcal{A}$. In the following, we will refer to such pairs as a "single" instanton sector.

Table 7: For the model $X_{2,2,2,2}(1^8)$ in the conifold frame defined by the A-periods $(X^1, P_0)$ at $z = (1 - 10^{-6})\mu$, we compare the Stokes discontinuity (underlined are stablised digits) across the positive real axis where the Borel singular point $\aleph X^1$ is located, with $\mathcal{F}^{(0)}$ evaluated up to order $g = 32$, and the value of the simple one-instanton amplitude (312) with $\tau_k = \delta_{k,1}/(2\pi)$ of instanton action $\mathcal{A} = \aleph X^1$. The ratio of the two results is constant for a suitable range of $g_s$ and it is close to $n_{0,1} = 512$.

| $g_s$ | $\mathrm{disc}_0 \mathcal{F}^{(0)}$ | $\mathcal{F}^{(1)}_{\mathcal{A}}$ | $\mathrm{disc}_0 \mathcal{F}^{(0)} / \mathcal{F}^{(1)}_{\mathcal{A}}$ |
|---|---|---|---|
| 1/2 | $\underline{6.532093616} \times 10^{-28}\,\mathrm{i}$ | $1.275799523 \times 10^{-30}\,\mathrm{i}$ | 512.00000 |
| 1 | $\underline{9.942777405} \times 10^{-13}\,\mathrm{i}$ | $1.941948685 \times 10^{-15}\,\mathrm{i}$ | 512.00000 |
| 3/2 | $\underline{9.693393658} \times 10^{-8}\,\mathrm{i}$ | $1.893240419 \times 10^{-10}\,\mathrm{i}$ | 512.00014 |
| 2 | $\underline{2.798526539} \times 10^{-5}\,\mathrm{i}$ | $5.465895145 \times 10^{-8}\,\mathrm{i}$ | 511.99784 |
| 5/2 | $\underline{8.007846}050 \times 10^{-4}\,\mathrm{i}$ | $1.563991489 \times 10^{-6}\,\mathrm{i}$ | 512.01340 |

the instanton action $\mathcal{A}$, we have

$$\mathcal{F}_g \sim \frac{\mathsf{S}_{\mathcal{A}}}{2\pi} \frac{\Gamma(2g-1)}{\mathcal{A}^{2g-1}} \mathcal{F}_0^{(\mathcal{A})} + \frac{\mathsf{S}_{-\mathcal{A}}}{2\pi} \frac{\Gamma(2g-1)}{(-\mathcal{A})^{2g-1}} \mathcal{F}_0^{(-\mathcal{A})}. \tag{375}$$

Assuming symmetry under $\mathcal{A} \to -\mathcal{A}$, such that $\mathsf{S}_{\mathcal{A}} = \mathsf{S}_{-\mathcal{A}}$ and $\mathcal{F}_0^{(\mathcal{A})} = -\mathcal{F}_0^{(-\mathcal{A})}$, this implies

$$s_{\mathcal{A},g}^0 = \frac{\mathcal{A}^{2g-1}}{\Gamma(2g-1)} \mathcal{F}_g \xrightarrow{g \gg 1} \frac{\mathsf{S}_{\mathcal{A}}}{\pi} \mathcal{F}_0^{(\mathcal{A})}. \tag{376}$$

Given a theoretical prediction for $\mathcal{F}_0^{(\mathcal{A})}$, we can similarly obtain a numerical prediction for the one-loop contribution to the one-instanton amplitude via

$$s_{\mathcal{A},g}^1 = \left( \frac{\mathcal{A}^{2g-1}}{\Gamma(2g-1)} \mathcal{F}_g - \frac{\mathsf{S}_{\mathcal{A}}}{\pi} \mathcal{F}_0^{(\mathcal{A})} \right) \frac{2g-2}{\mathcal{A}} \xrightarrow{g \gg 1} \frac{\mathsf{S}_{\mathcal{A}}}{\pi} \mathcal{F}_1^{(\mathcal{A})}. \tag{377}$$

These asymptotic estimates for the loop coefficients $\mathcal{F}_g^{(\mathcal{A})}$ of the one-instanton amplitude can be compared to our theoretical predictions, extracted from the exact formula (278). The first few loop orders are extracted from the general formula in (284). These formulae depend on the two constants $c^0$ and $c^1$ defined in (229). In the one-modulus case, we can evaluate all derivatives in terms of the free energies $\mathcal{F}_0(t)$ and $\mathcal{F}_1(t)$ as

$$\begin{aligned}
\tau_{00} &= 2\mathcal{F}_0(t) - 2t\partial_t \mathcal{F}_0(t) + t^2 \partial_t^2 \mathcal{F}_0(t), \\
\tau_{01} &= \partial_t \mathcal{F}_0(t) - t\partial_t^2 \mathcal{F}_0(t), \\
\tau_{11} &= \partial_t^2 \mathcal{F}_0(t),
\end{aligned} \tag{378}$$

and

$$\begin{aligned}
C_{000} &= -\frac{t^3}{X^0} \partial_t^3 \mathcal{F}_0(t), \qquad C_{001} = \frac{t^2}{X^0} \partial_t^3 \mathcal{F}_0(t), \\
C_{011} &= -\frac{t}{X^0} \partial_t^3 \mathcal{F}_0(t), \qquad C_{111} = \frac{1}{X^0} \partial_t^3 \mathcal{F}_0(t),
\end{aligned} \tag{379}$$

as well as

$$\frac{\partial \mathcal{F}_1(t)}{\partial X^0} = -\frac{t}{X^0} \partial_t \mathcal{F}_1(t), \qquad \frac{\mathcal{F}_1(t)}{X^1} = \frac{1}{X^0} \partial_t \mathcal{F}_1(t). \tag{380}$$

When the contributions from two instanton sectors are comparable, we can still extract numerical predictions under the condition that the contribution of one of the instanton sectors is known completely. We will see instances of this in examples, to which we now turn.

Table 8: The asymptotic estimate of the instanton action obtained via the sequence $s_g^{(\mathcal{A})}$ compared to the periods $P_0$ and $X^0$. The computation is based on $\mathcal{F}_g(X^0, X^1)$ evaluated up to genus 64 for the example of the quintic.

| $z$ | Asymptotic estimate of action | $\aleph P_0$ | $\aleph X^0$ |
|---|---|---|---|
| $10^{-6}\mu$ | $39.4784191203291289117078\,\mathrm{i}$ | $-1138.03925609468142282947$ | $-39.4784191203291289125828\,\mathrm{i}$ |
| $10^{-4}\mu$ | $39.47856920606554978139\,\mathrm{i}$ | $-494.0326608439227865458$ | $-39.47856920606554978519\,\mathrm{i}$ |
| $10^{-2}\mu$ | $39.4936233785076876\,\mathrm{i}$ | $-141.223891994711932$ | $-39.4936233785077847\,\mathrm{i}$ |
| $1/8\mu$ | $39.67553248\,\mathrm{i}$ | $-42.99972954$ | $-39.67553551\,\mathrm{i}$ |
| $1/7\mu$ | $39.704120\,\mathrm{i}$ | $-39.321466$ | $-39.705033\,\mathrm{i}$ |
| $1/6\mu$ | $44.\,\mathrm{i}$ | $-35.$ | $-40.\,\mathrm{i}$ |
| $1/5\mu$ | $30.73$ | $-30.65$ | $-39.80\,\mathrm{i}$ |
| $1/3\mu$ | $19.0714633228$ | $-19.0714633440$ | $-40.0447429295\,\mathrm{i}$ |
| $1/2\mu$ | $11.1503736933130532$ | $-11.150373693313239$ | $-40.390151835645641\,\mathrm{i}$ |
| $5/6\mu$ | $2.65608862910239270$ | $-2.65608862910239633$ | $-41.3365927932450749\,\mathrm{i}$ |
| $23/24\mu$ | $0.6030509245517648306$ | $-0.6030509245517648990$ | $-41.924087413819182
92\,\mathrm{i}$ |

We will begin by studying the asymptotics of the topological string amplitudes $\mathcal{F}_g(X^0, X^1)$ in the MUM frame. We will perform this analysis for the quintic.

We numerically evaluate the $\mathcal{F}_g(X^0, X^1)$, at real values of $z$ in between the MUM point and the conifold point,

$$0 < z < \mu = 5^{-5}, \tag{381}$$

and obtain a sequence of values of $\mathcal{F}_g(X^0, X^1)$. We then substitute these values into the sequence $s_g^{(\mathcal{A})}$ defined in (373) and perform the number of Richardson transformations which best stabilizes the sequence (i.e. which leads to the largest number of coincident digits in the last two entries of the Richardson transformed sequence). These values are given in Table 8 for a range of values for $z$. From our study of the Borel plane, see Figure 4, we expect the dominant instanton action to transition from $\aleph X^0$ close to the MUM point to $\aleph P_0$ close to the conifold point. These two periods are also listed in Table 8 for each value of $z$. We find that the respective actions are reproduced to high precision by the asymptotics close to the associated singular point. The method fails when the two actions are of comparable size.

We next determine asymptotic estimates for $b$ at points in moduli space at which the asymptotics is dominated by one instanton action. The results are collected in Table 9, and not surprisingly confirm the theoretical value $b = 1$.

Finally, we turn to the asymptotic predictions for $\mathcal{F}_0^{(1)}$ and $\mathcal{F}_1^{(1)}$. Close to the MUM point,

Table 9: The asymptotic estimate of the the constant $b$ via the sequence $s_g^{(b)}$ evaluated for $\mathcal{A} = \aleph P_0$, $\mathcal{A} = \aleph X^0$ as appropriate. The computation is based on $\mathcal{F}_g(X^0, X^1)$ evaluated up to genus 64 for the example of the quintic.

| $z$ | Asymptotic estimate of $b$ |
|---|---|
| $10^{-6}\mu$ | $1.0000000000000000376$ |
| $10^{-4}\mu$ | $1.0000000000000000376$ |
| $10^{-2}\mu$ | $1.0000000000000157$ |
| $1/8\mu$ | $1.0000042$ |
| $1/3\mu$ | $1.00000000537$ |
| $1/2\mu$ | $1.0000000000001046$ |
| $5/6\mu$ | $1.0000000000001197$ |
| $23/24\mu$ | $1.000000000000001512$ |

Table 10: Comparison, in the frame $(X^0, X^1)$, between the asymptotic estimate for the normalized genus 0 one-instanton amplitude $\frac{S_\mu}{\pi}\mathcal{F}_0^{(\mu)}$ and the prediction, using $S_\mu = 1$, for the example of the quintic.

| $z$ | Asymptotic estimate I | Asymptotic estimate II | Prediction |
|---|---|---|---|
| $\mu/8$ | $1. \times 10^9$ | $-2.0685 \times 10^{-8}$ | $-2.0618 \times 10^{-8}$ |
| $\mu/7$ | $10000.$ | $-4.659137 \times 10^{-8}$ | $-4.658992 \times 10^{-8}$ |
| $\mu/6$ | $0.01$ | $-1.163074023 \times 10^{-7}$ | $-1.163074007 \times 10^{-7}$ |
| $\mu/5$ | $-3.335 \times 10^{-7}$ | $-3.313309985143 \times 10^{-7}$ | $-3.313309985104 \times 10^{-7}$ |
| $\mu/3$ | $-5.1510310321251 \times 10^{-6}$ | $-5.151031032069825626 \times 10^{-6}$ | $-5.151031032069825187 \times 10^{-6}$ |
| $\mu/2$ | $-0.000038081205262381317350$ | $-0.000038081205262381317350$ | $-0.000038081205262381316984$ |
| $5\mu/6$ | $-0.000374000001694825755160$ | $-0.000374000001694825755160$ | $-0.000374000001694825754743$ |
| $23\mu/24$ | $-0.0004997585182539551567396$ | $-0.0004997585182539551567396$ | $-0.0004997585182539551566954$ |

the asymptotics will be governed by the analytically derived instanton amplitudes (322) associated to the instanton action $\mathcal{A}_{\text{MUM}} = \aleph X^0$. Recall that these were imposed as boundary conditions in our computations. Close to the conifold point, we predict that the instanton amplitudes (284) associated to the instanton action $\mathcal{A}_\mu = \aleph P_0$, i.e. evaluated with

$$c^0 = \aleph, \qquad c^1 = 0, \tag{382}$$

should govern the asymptotics. In the second column of Table 10 and of Table 11, under the heading "Asymptotic estimate I", we have listed the asymptotic estimates for $\frac{S_\mu}{\pi}\mathcal{F}_0^{(\mu)}$ and $\frac{S_\mu}{\pi}\mathcal{F}_1^{(\mu)}$ respectively, based on the sequences $s^0_{\mathcal{A}_\mu, g}$ and $s^1_{\mathcal{A}_\mu, g}$ introduced above.

The estimates match the prediction based on (382), given in the fourth column of Table 10 and of Table 11, convincingly close to the conifold point, but break down around $z \sim \frac{\mu}{6}$. At this value of $z$, $|\mathcal{A}_\mu| < |\mathcal{A}_{\text{MUM}}|$, i.e. the conifold action is still dominant, see Table 8. Given that the action enters in the asymptotics to the power of $1-2g$, the breakdown of our estimate already at this value of $z$ implies that the instanton amplitudes associated to $\mathcal{A}_{\text{MUM}}$ must be appreciably larger than the amplitude associated to $\mathcal{A}_\mu$. And indeed, at genus $g = 64$, we compute

$$\left|\frac{\Gamma(2 \times 64 - 1)}{\mathcal{A}_\mu^{2\times64-1}}\right| \sim 2 \times 10^{15}, \qquad \left|\frac{\Gamma(2 \times 64 - 1)}{\mathcal{A}_{\text{MUM}}^{2\times64-1}}\right| \sim 8 \times 10^7, \tag{383}$$

while

$$|\mathcal{F}_0^{(\mu)}| \sim 10^{-7}, \qquad |\mathcal{F}_0^{(\text{MUM})}| \sim 40, \tag{384}$$

and

$$|\mathcal{F}_1^{(\mu)}| \sim 10^{-7}, \qquad |\mathcal{F}_1^{(\text{MUM})}| \sim 1. \tag{385}$$

Table 11: Comparison, in the frame $(X^0, X^1)$, between the asymptotic estimate for the normalized genus 1 one-instanton amplitude $\frac{S_\mu}{\pi}\mathcal{F}_1^{(\mu)}$ and the prediction, using $S_\mu = 1$, for the example of the quintic.

| $z$ | Asymptotic estimate I | Asymptotic estimate II | Prediction |
|---|---|---|---|
| $\mu/8$ | $-4. \times 10^9$ | $-1.6724 \times 10^{-8}$ | $-1.6650 \times 10^{-8}$ |
| $\mu/7$ | $-50000.$ | $-3.206806 \times 10^{-8}$ | $-3.206579 \times 10^{-8}$ |
| $\mu/6$ | $-0.05$ | $-6.405055587 \times 10^{-8}$ | $-6.405055488 \times 10^{-8}$ |
| $\mu/5$ | $-1.3 \times 10^{-7}$ | $-1.280693556310 \times 10^{-7}$ | $-1.280693556226 \times 10^{-7}$ |
| $\mu/3$ | $5.88102454425421 \times 10^{-7}$ | $5.88102454418872040 \times 10^{-7}$ | $5.88102454418871349 \times 10^{-7}$ |
| $\mu/2$ | $0.0000255348240454449402553$ | $0.0000255348240454449402553$ | $0.0000255348240454449402293$ |
| $5\mu/6$ | $0.0012143581076950720071777$ | $0.0012143581076950720071777$ | $0.0012143581076950720070550$ |
| $23\mu/24$ | $0.0060936988242359349007$ | $0.0060936988242359349007$ | $0.0060936988242359349297$ |

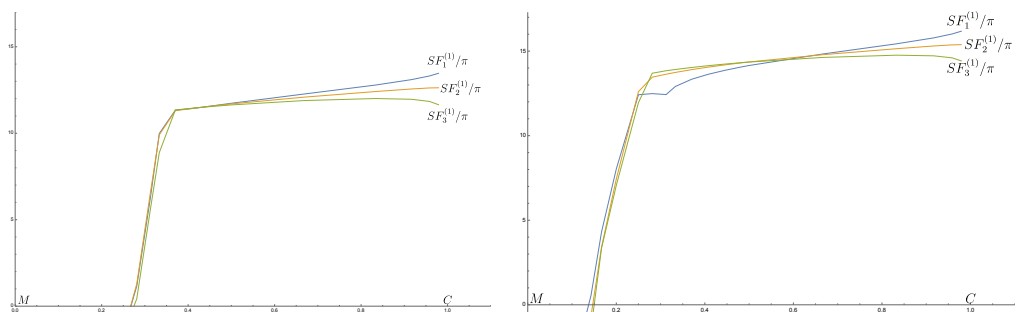

Figure 14: The significant digits of the asymptotic estimates for 1 instanton genus $1, 2, 3$ contributions without (left) and with (right) constant map subtraction for the quintic $X_5(1^5)$ to $g = 64$.

Hence, beyond $z \sim \frac{\mu}{5}$, the contribution of the two instanton sectors associated to $\mathcal{A}_\mu$ and $\mathcal{A}_{\text{MUM}}$ to the asymptotics become first comparable and then dominated by the latter. However, as we know the leading asymptotics of $\mathcal{F}_g(X^0, X^1)$ close to the MUM point exactly – it is given by (322) – we can subtract it from $\mathcal{F}_g(X^0, X^1)$ and test the prediction based on (382) also in this region. The results of this computation, $s^0_{\mathcal{A}_\mu, g}$ $s^1_{\mathcal{A}_\mu, g}$ evaluated on $\mathcal{F}_g(X^0, X^1)$ with the leading asymptotics subtracted, are given in the third column, under the heading "Asymptotics II" of Table 10 and Table 11 respectively and show convincing agreement with the predictions.

Very close to the MUM point (e.g. at the values $10^{-2}\mu$, $10^{-4}\mu$, $10^{-6}\mu$ considered in Table 8 and Table 9), our numerical precision is no longer sufficient to resolve the subleading contribution of the $\mathcal{A}_\mu$ instanton sector to the asymptotics.

The numerical study of asymptotic estimates is illustrated graphically in Figure 14, which also compares the asymptotic estimates for $\mathcal{F}_2^{(\mu)}$ and $\mathcal{F}_3^{(\mu)}$ to the theoretical predictions. Indeed, as we have predictions for the instanton amplitudes $\mathcal{F}_g^{(\mu)}$ at arbitrary loop order, we can obtain asymptotic predictions for $\mathcal{F}_n^{(\mu)}$ from a sequence $s^n_{\mathcal{A}_\mu, g}$ in which the contributions from the lower loop coefficients to the asymptotics of $\mathcal{F}_g(X^0, X^1)$ have been subtracted. With increasing $n$, we must compute the topological string amplitudes to increasingly high genus and evaluate them with ever increasing numerical precision. Below, we will compute asymptotic estimates of the one-instanton amplitudes up to loop order 8 for $\mathcal{F}_g(X^1, P_0)$.

We now change frames and perform the analogous asymptotic analysis for the topological string amplitudes $\mathcal{F}_g(X^1, P_0)$ in the conifold frame. Here, the leading asymptotics close to the conifold point is known exactly – it is given by (337). We can hence test the prediction of (284) for the instanton sector associated to the instanton action $\mathcal{A} = \aleph X^0$, i.e. for

$$c_0 = 0, \quad c_1 = \aleph. \tag{386}$$

Table 12: Comparison, in the frame $(X^1, P_0)$, between the asymptotic estimate for $\frac{S_{\text{MUM}}}{\pi} \mathcal{F}_0^{(\mathcal{A}_{\text{MUM}})}$ and the prediction, using $S_{\text{MUM}} = -\chi$, for the example of the quintic.

| $z$ | Asymptotic estimate I | Asymptotic estimate II | Prediction |
|---|---|---|---|
| $10^{-6}\mu$ | $-382.257657577866488\,\mathrm{i}$ | $-382.257657577866488\,\mathrm{i}$ | $-382.257657577867968\,\mathrm{i}$ |
| $10^{-4}\mu$ | $-364.260537318513919\,\mathrm{i}$ | $-364.260537318513919\,\mathrm{i}$ | $-364.260537318518819\,\mathrm{i}$ |
| $10^{-2}\mu$ | $-312.00085776255\,\mathrm{i}$ | $-312.00085776255\,\mathrm{i}$ | $-312.00085777409\,\mathrm{i}$ |
| $\mu/8$ | $-401.997458\,\mathrm{i}$ | $-402.2\,\mathrm{i}$ | $-401.997221\,\mathrm{i}$ |
| $\mu/7$ | $300\,\mathrm{i}$ | $-229.4055067\,\mathrm{i}$ | $-229.4055313\,\mathrm{i}$ |
| $\mu/6$ | $2 \times 10^7\,\mathrm{i}$ | $-221.78747\,\mathrm{i}$ | $-221.78865\,\mathrm{i}$ |
| $\mu/5$ | $2 \times 10^{13}\,\mathrm{i}$ | $-212.105904\,\mathrm{i}$ | $-212.106060\,\mathrm{i}$ |

Table 13: Comparison, in the frame $(X^1, P_0)$, between the asymptotic estimate for $\frac{S_{\text{MUM}}}{\pi} \mathcal{F}_1^{(\mathcal{A}_{\text{MUM}})}$ and the prediction, using $S_{\text{MUM}} = -\chi$, for the example of the quintic.

| $z$ | Asymptotic estimate I | Asymptotic estimate II | Prediction |
|---|---|---|---|
| $10^{-6}\mu$ | $-19.96043925387713$ | $-19.96043925387713$ | $-19.96043925387822$ |
| $10^{-4}\mu$ | $-32.52506815749229$ | $-32.52506815749229$ | $-32.52506815749648$ |
| $10^{-2}\mu$ | $-49.74119660140$ | $-49.74119660140$ | $-49.74119660508$ |
| $\mu/8$ | $10.132210$ | $10.33$ | $10.132118$ |
| $\mu/7$ | $-1000$ | $-53.5985829$ | $-53.5985968$ |
| $\mu/6$ | $-6 \times 10^7$ | $-53.1246552$ | $-53.1246694$ |
| $\mu/5$ | $-5 \times 10^{13}$ | $-52.3718764$ | $-52.3718908$ |

According to the Borel plane plots on the left in Figure 6 and Figure 8, this should capture the leading asymptotics near the MUM point and the subleading asymptotics near the conifold point. In the second column of Table 12 and Table 13, under the heading "Asymptotics I", we have listed the asymptotic estimates for $\frac{S_{\text{MUM}}}{\pi} \mathcal{F}_0^{(\mathcal{A}_{\text{MUM}})}$ and $\frac{S_{\text{MUM}}}{\pi} \mathcal{F}_1^{(\mathcal{A}_{\text{MUM}})}$ respectively, based on the sequences $s^0_{\mathcal{A}_{\text{MUM}},g}$ and $s^1_{\mathcal{A}_{\text{MUM}},g}$. The estimates match the prediction based on (386), given in the fourth column of the tables, all the way up to $z = \mu/8$. By $z = \mu/7$, the instanton action $\mathcal{A}_\mu$ has inched closer to the origin of the Borel plane than $\mathcal{A}_{\text{MUM}}$. The prediction (386) now applies to the subleading asymptotics. Subtracting the leading asymptotics yields the asymptotic estimates listed in the third column of Table 12 and Table 13, under the heading "Asymptotics II" – these show good agreement with the predictions for all points $z$ in the moduli space studied. Note that unlike the case for $\mathcal{F}_g(X^0, X^1)$, the breakdown of the asymptotic estimate for the instanton amplitudes ("Asymptotics I" in the tables) coincides with the point in moduli space at which $\mathcal{A}_\mu$ moves past $\mathcal{A}_{\text{MUM}}$ to become the closest action to the origin of the Borel plane. The difference between the two cases is that for $\mathcal{F}_g(X^1, P_0)$, the size of the instanton actions in the two sectors is comparable, and cannot compensate for the difference in the contribution from the actions to the asymptotics (cf. discussion around (383)).

As pointed out above, we can push the comparison between asymptotic estimates for the one-instanton amplitudes and the theoretical prediction based on (278) to higher loop order. In Figure 15, we perform this comparison for $\mathcal{F}^{(0)}(X^1, P_0)$ at $z = 10^{-2}\mu$ up to loop order 8, and find good agreement.

Finally, we check the entire one-instanton amplitude formula using the Stokes discontinuity

Table 14: Comparison, in the frame $(P_0, P_1)$, between the asymptotic estimate for $\frac{S_{\text{MUM}}}{\pi} \mathcal{F}_0^{(\mathcal{A}_{\text{MUM}})}$ and the prediction using $S_{\text{MUM}} = -\chi$ for the first three values of $z$, and between the asymptotic estimate for $\frac{S_\mu}{\pi} \mathcal{F}_0^{(\mu)}$ and the prediction using $S_\mu = 1$ for the last four values of $z$, for the example of the quintic.

| $z$ | Asymptotic estimate | Prediction |
|---|---|---|
| $10^{-6}\mu$ | $2.578293673861729631 - 350.320544587085132123\,i$ | $2.578293673861729616 - 350.320544587085132654\,i$ |
| $10^{-4}\mu$ | $4.283968180390488729 - 314.909621199366985163\,i$ | $4.283968180390488721 - 314.909621199366985961\,i$ |
| $10^{-2}\mu$ | $6.26156388754039 - 241.46605914584608\,i$ | $6.26156388754057 - 241.46605914585064\,i$ |
| $\mu/3$ | $-0.966171619903455385$ | $-0.966171619903454465$ |
| $\mu/2$ | $-0.5648845303304413622559$ | $-0.5648845303304413621804$ |
| $5\mu/6$ | $-0.13455902187980453773105$ | $-0.13455902187980453771306$ |
| $23\mu/24$ | $-0.0305509167360954756 2773$ | $-0.0305509167360954756 2365$ |

Table 15: Comparison, in the frame $(P_0, P_1)$, between the asymptotic estimate for $\frac{S_{\text{MUM}}}{\pi} \mathcal{F}_1^{(\mathcal{A}_{\text{MUM}})}$ and the prediction using $S_{\text{MUM}} = -\chi$ for the first three values of $z$, and between the asymptotic estimate for $\frac{S_\mu}{\pi} \mathcal{F}_1^{(\mu)}$ and the prediction using $S_\mu = 1$ for the last four values of $z$, for the example of the quintic.

| $z$ | Asymptotic estimate | Prediction |
|---|---|---|
| $10^{-6}\mu$ | $-22.118606937252992852 + 0.154244032879358745\,i$ | $-22.118606937252992805 + 0.154244032879358748\,i$ |
| $10^{-4}\mu$ | $-25.715437369360251839 - 0.575488280479718081\,i$ | $-25.715437369360252154 - 0.575488280479718072\,i$ |
| $10^{-2}\mu$ | $-23.58594545231497 - 2.10986661318608\,i$ | $-23.58594545232135 - 2.10986661318582\,i$ |
| $\mu/3$ | $0.050660591821169677$ | $0.050660591821168886$ |
| $\mu/2$ | $0.050660591821168885 72605$ | $0.050660591821168885 72194$ |
| $5\mu/6$ | $0.050660591821168885 72605$ | $0.050660591821168885 72194$ |
| $23\mu/24$ | $0.050660591821168885 72605$ | $0.050660591821168885 72194$ |

relation (343). The Stokes discontinuity of $\mathcal{F}^{(0)}(X^1, P_0)$ across the negative imaginary axis should be proportional to the Borel resummation of the one-instanton amplitude with action $\aleph X^0$. At $z = 1/100\mu$ and with $g_s = -i$, the Stokes discontinuity, with $\mathcal{F}^{(0)}(X^1, P_0)$ expanded up to order $g = 61$, evaluates to

$$\text{disc}_{\pi/2} \mathcal{F}^{(0)}(X^1, P_0)(z = 1/100, g_s = -i) = \underline{1.85787242132104924}598 \times 10^{-16}, \qquad (387)$$

where stable digits are underlined. The Borel resummation of $\mathcal{F}^{(1)}(X^1, P_0)$ is

$$s_- \mathcal{F}^{(1)}(X^1, P_0)(z = 1/100, g_s = -i) = \underline{1.85787242132104914 39127620868310}874 \times 10^{-16}$$
$$- i\,\underline{2.144138518276324}130 \times 10^{-31}, \qquad (388)$$

where stable digits are underlined. Both the real and the imaginary part of these two results agree up to $10^{-31}$, which is the order of magnitude of the second instanton amplitude with action $2\aleph X^0$.

As a final example, we consider the asymptotics of the topological string amplitudes $\mathcal{F}_g(P_0, P_1)$. Unlike the two previous cases, we do not have access to exact asymptotics in this frame anywhere on moduli space. Hence, we are restricted to studying leading asymptotics. The predictions in Table 14 and Table 15 are for $\frac{S_{\text{MUM}}}{\pi} F_g^{(\mathcal{A}_{\text{MUM}})}$, $g = 0, 1$ for the first three values of $z$ close to the MUM point, and for $\frac{S_\mu}{\pi} \mathcal{F}_g^{(\mu)}$, $g = 0, 1$, for the remaining four values of $z$. We find convincing agreement at all loop orders considered.

All of the analysis in this section so far has been for the quintic. We can of course equally well study other one-parameter models. A comparison of asymptotic estimates and theoretical prediction for one-instanton amplitudes of $\mathcal{F}_g(X^0, X^1)$ is performed for the dectic $X_{10}(1^3, 2, 5)$ in Figure 16.

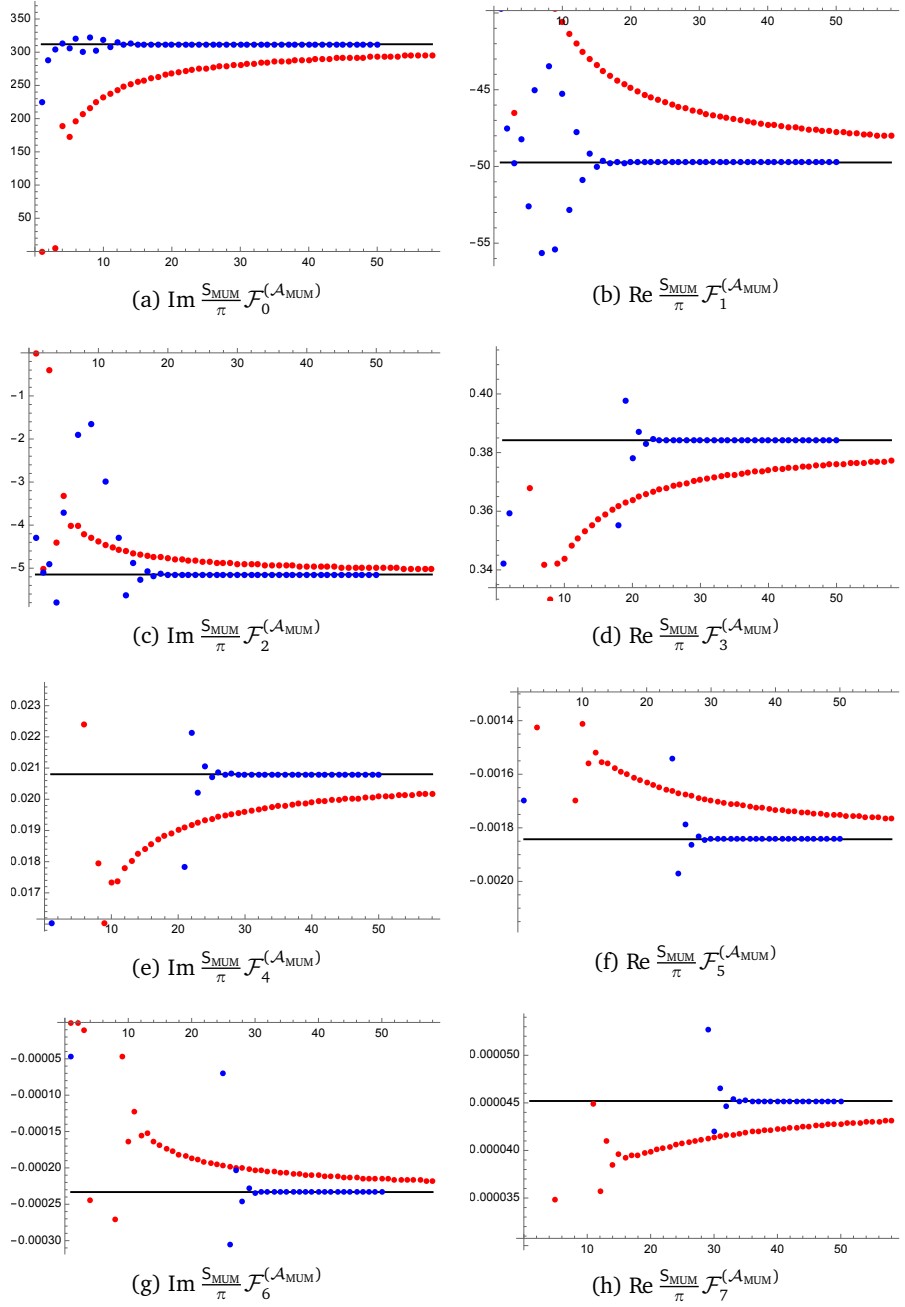

Figure 15: Comparison between auxiliary sequences (similar to (376), red dots) that asymptote to $\frac{S_{\text{MUM}}}{\pi}\mathcal{F}_g^{(\text{MUM})}$ ($g = 0, 1, \ldots, 7$) constructed from evaluation of $\mathcal{F}_g(X^1, P_0)$ up to $g = 61$ with $S_{\text{MUM}} = -\chi$, their Richardson transformations (8-th order, blue dots), and the prediction from the 1-instanton formula (278) (black line) for the $X^0$ instanton sector in the case of the quintic.

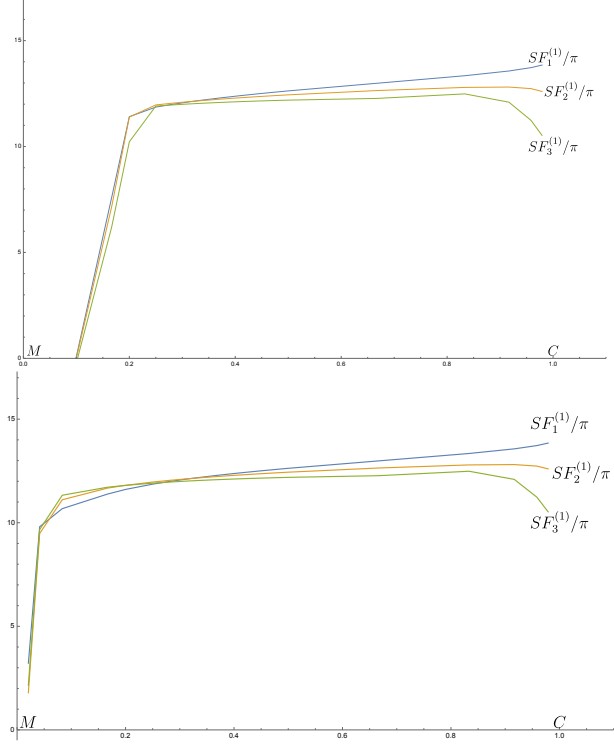

Figure 16: The significant digits of the asymptotic estimates for the one-instanton genus $1, 2, 3$ contributions without (left) and with (right) constant map subtraction for the dectic $X_{10}(1^3, 2, 5)$ to $g = 50$.

## 7 Conclusions

The asymptotic nature of the perturbative series in topological string theory indicates that there are additional non-perturbative sectors which can be decoded from perturbation theory. These sectors are characterized by their Borel singularities or instanton actions, by asymptotic series or instanton amplitudes, and by Stokes constants. In this paper, we have found general results for the instanton amplitudes for arbitrary compact Calabi–Yau manifolds, as well as particular results on the structure of Borel singularities, providing a far-reaching generalization of previous results in [20, 21, 23, 94], which focused on the one-modulus, local case. More concretely, we have found that the Borel singularities of perturbative topological string free energies in a generic compact Calabi-Yau threefold are integral periods of the form

$$\mathcal{A} = \ell(c^J P_J + d_J X^J), \tag{389}$$

similar to the findings made in the local cases [20, 21, 26]. Here $X^J, P_J$ are integral A- and B-periods respectively whose choice determines a symplectic frame, $\{c^J, d_J\}$ are coprime integers, and $\ell = 1, 2, \ldots$. We checked this result with the examples of hypergeometric one parameter Calabi-Yau threefolds listed in Table 1. We have found elegant closed-form expressions for the instanton amplitudes, i.e. the non-perturbative corrections to free energies, in the form of exponentially small trans-series, by solving the trans-series extension of the holomorphic anomaly equations. They are encoded in terms of non-perturbative partition functions given in (309), related to the free energies through the usual exponential-logarithm relation. The non-perturbative partition functions can be evaluated in the holomorphic limit associated to a chosen symplectic frame. The results depend on the type of the Borel singularity the instanton amplitude is evaluated at. If the Borel singularity is an A-period, in the sense that all

the coefficients $c^J$ vanish, the holomorphic limit reduces simply to (312), (313); otherwise, the holomorphic limit is obtained with the rules (310), (311). In the case of one-instanton sectors with $\ell = 1$, the two types of holomorphic limit of the instanton amplitude are respectively (179) and (278).

Our results for the amplitudes show that, in the holomorphic limit, they are simple functionals of the perturbative free energies, as showcased by (278) in the case of one-instanton amplitude, and by the replacement rules (310),(311) in the multi-instanton amplitudes. They also have experimental implications, since they determine the large genus asymptotics of the topological string free energies. We have verified that this is the case in many one-parameter compact Calabi–Yau 3-folds, including the famous quintic Calabi–Yau manifold – see Section 6.

The structure of the instanton amplitudes is intriguing from the physics point of view. These amplitudes involve shifts of the background – specified by the coordinates $X^I$ of the big moduli space – by integer multiples of the string coupling constant. This phenomenon was observed in [23] in the local case, but as we have mentioned in this paper, its appearance in the compact case is somewhat unexpected. It suggests a quantization of the big moduli space coordinates in units of the string coupling constant, as in large $N$ dualities.

Another surprising aspect of the non-perturbative sectors that we have described in this paper is that the corresponding instanton actions are closed string periods, i.e. masses of even/odd-dimensional D-branes in the A/B model, respectively. It has been sometimes suggested (see e.g. [97]) that non-perturbative corrections in the A/B topological string should be due to Lagrangian/holomorphic D-branes, respectively. These corrections do not seem to appear in the resurgent structure uncovered in this paper and in the previous works [20, 21, 23]. We should however emphasize that the resurgent structure associated to the perturbative series does not necessarily cover all the relevant non-perturbative sectors in a physical theory. Consider for example the non-linear $O(3)$ sigma model. There are instanton amplitudes in this model that capture the dependence of observables on the topological $\theta$ angle, and that cannot be detected from perturbation theory alone (this has been made completely explicit in recent studies [98, 99]). Therefore, it is conceivable that the non-perturbative sectors that we have described in this paper should be thought of as "renormalon" sectors of topological string theory, controlling the factorial divergence of the perturbative series due to integration over moduli space. There could be in addition purely "instanton" sectors, perhaps due to Lagrangian/holomorphic branes as suggested in [97], and undetectable in the topological string perturbative series. This possibility remains for the moment rather uncertain in the compact case, since in the absence of a concrete non-perturbative definition of the theory, it is difficult to talk about non-perturbative sectors, other than the ones obtained from the perturbative series.

Perhaps the main challenge is to find methods, other than numerical, to determine the remaining ingredients of the resurgent structure, namely the location of Borel singularities and the values of the Stokes constants. We believe that these ingredients are deeply related to stability structures and BPS invariants, and we have indeed shown in this paper that genus zero Gopakumar–Vafa invariants are realized as Stokes constants. A complete and more precise picture however is still lacking. Perhaps an extension of our non-perturbative analysis to degeneration points other than the MUM and conifold points studied above (those that occur at $z = \infty$ in hypergeometric models) could shed further light on these matters.

## Acknowledgments

We would like to thank Michael Borinsky, David Sauzin, Ricardo Schiappa and Samson Shatashvili for useful comments and discussions.

**Funding information** A.K. and M.M. would like to thank the IHES and the Physics Department of the École Normale Supérieure (Paris) respectively for hospitality during the inception of this paper. The work of M.M. has been supported in part by the ERC-SyG project "Recursive and Exact New Quantum Theory" (ReNewQuantum), which received funding from the European Research Council (ERC) under the European Union's Horizon 2020 research and innovation program, grant agreement No. 810573. J.G. is supported by the Startup Funding no. 4007022316 and no. 4060692201/011 of the Southeast University. A.K.K.P. acknowledges support under ANR grant ANR-21-CE31-0021.

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
