# Peer review of "Non-perturbative topological string theory on compact Calabi-Yau 3-folds"

_SciPost Physics, doi:SciPost Phys. 16, 079 (2024)_

## Round 1 · Referee Report · Anonymous (Referee 1) · 2023-12-11

Strengths

  • interesting results
  • useful review of the subject
  • well written

Weaknesses

  • too brief summary of the results of the paper

Report

In this paper the authors reveal non-perturbative structure of topological string amplitudes on arbitrary, compact Calabi-Yau manifolds. Their approach is based on the theory of resurgence and finding trans-series solutions of holomorphic anomaly equations. The results of this paper are far-reaching generalization of the results of the work cited as [23], where such trans-series solutions were found for a local Calabi-Yau with one modulus. Apart from finding explicit form of trans-series solutions, the authors also find relations between Stokes constants and integral Gopakumar-Vafa invariants, as well as the relations of amplitudes for Calabi-Yau three-folds to properties of Calabi-Yau four-folds. All these results are new and interesting and deserve publication. The paper is nicely written and large fraction of its volume is a review of various previous results – while it may not be necessary for experts in the field, it is certainly valuable for those who are not familiar with recent developments in this area. I have however one more critical remark – with such a longer paper, I find the introduction and the “Conclusions” section too brief and too general. It would be of advantage for readers if in one of these sections they could find a bit more detailed summary of the most important results, at least with references to most important formulas, pictures or tables from the bulk of the paper. I would encourage the authors to include such a more detailed summary of their results.

Requested changes

Including more detailed summary of the most important results, at least with references to most important formulas, pictures or tables from the bulk of the paper.

  • validity: high
  • significance: high
  • originality: high
  • clarity: high
  • formatting: excellent
  • grammar: perfect

Author:  Jie Gu  on 2023-12-12  [id 4183]

(in reply to Report 1 on 2023-12-11)

Thank you very much for the comments! We will expand the introduction/conclusion per your suggestion.

---

## Round 1 · Referee Report · Anonymous (Referee 2) · 2024-1-4

Strengths

1-strong results
2-detailed computations
3-convincing interpretation

Weaknesses

long and difficult to read and digest without detailed knowledge of previous results and many computational techniques, seemingly cumbersome notation

Report

This is a mind-blowing computational tour-de-force exploration of non-perturbative structures in topological string theory on compact Calabi-Yau 3-folds. The analysis proceeds along the lines developed by Schiappa, Marino et al. in local models (trans-series solutions to holomorphic anomaly equations), while the numerics depends on Klemm's high-genus perturbative data (determined by boundary conditions and modularity of BPS states).

The paper is fairly long and contains many formulas that depend on a variety of conventions for various frames and limits. While I have been able to verify most of what I had the courage to check, I am left with the impression that the presentation is not quite optimized and would like to encourage further streamlining in future installments. Specifically, section 2 contains a substantial number of misprints (in index position etc.) that the authors should identify and correct before publication.
  • validity: top
  • significance: high
  • originality: high
  • clarity: good
  • formatting: excellent
  • grammar: excellent

Author:  Jie Gu  on 2024-01-09  [id 4230]

(in reply to Report 2 on 2024-01-04)

Thank you very much for the comments! We will try to correct as many misprints in section two as possible. We also plan to expand the introduction/conclusion, including references to the most important formulas, to help the audience get a better handle of the most important results.

---

## Round 2 · Referee Report · Anonymous · 2024-2-7

Strengths

see initial review

Weaknesses

see initial review

Report

Following corrections by authors, I recommend publication in SciPos.

---

## Round 2 · Referee Report · Anonymous · 2024-2-8

Report

The authors made the changes that I suggested and the paper can be published.

---

## Round 2 · List of Changes

1. Typos corrections in section two, especially regarding the indices in formulas, per the request of one referee.
2. Expansion of conclusion and references to important formulas added, per the request of another referee.

---

## Editorial Decision

published